# Present, past, and future of the oxygen minimum zone in the northern Indian Ocean

Tim Rixen[1], Greg Cowie[2], Birgit Gaye[3], Joaquim Goes[4], Helga do Rosário Gomes[4], Raleigh R. Hood[5], Zouhair Lachkar[6], Henrike Schmidt[7], Joachim Segschneider[8], Arvind Singh[9]

Leibniz Centre for Tropical Marine Research (ZMT), Fahrenheitstr. 6, 28359 Bremen, Germany (tim.rixen@leibniz-zmt.de)

University of Edinburgh, School of Geosciences, James Hutton Road, Edinburgh EH9 3FE, Scotland (glcowie@ed.ac.uk)

Institute for Geology, Universität Hamburg, Bundesstraße 55, 20146 Hamburg, Germany (birgit.gaye@uni-hamburg.de)

Marine Biology, Department of Marine Biology and Paleoenvironment Lamont Doherty Earth Observatory at Columbia University, 61 Route 9W, Palisades, New York, 10964 (jig@ldeo.columbia.edu), (helga@ldeo.columbia.edu)

Horn Point Laboratory, University of Maryland Center for Environmental Science, P.O. Box 775, Cambridge, MD  21613 (rhood@umces.edu)

Center for Prototype Climate Modeling (CPCM), NYU Abu Dhabi (zouhair.lachkar@nyu.edu)

GEOMAR Helmholtz-Zentrum fuer Ozeanforschung Kiel, Duesternbrooker Weg 20, 24105 Kiel, Germany (hschmidt@geomar.de)

Institute of Geosciences, Christian-Albrechts-Universität Kiel (CAU), Ludewig-Meyn-Straße 10. 24118 Kiel, Germany (joachim.segschneider@ifg.uni-kiel.de)

Geosciences Division Physical Research Laboratory (PRL) Navrangpura, Ahmedabad 380 009, India (arvinds@prl.res.in)

*Correspondence to*: Tim Rixen (tim.rixen@leibniz-zmt.de)

**Abstract.**

Decreasing concentrations of dissolved oxygen in the ocean are considered as one

of the main threats to marine ecosystems as they jeopardize the growth of higher

organisms and alter the marine nitrogen cycle, which is strongly bound to the carbon

cycle and climate. While higher organisms in general start to suffer from oxygen

concentrations < ~63 µM (hypoxia), the marine nitrogen cycle responds to oxygen

concentration below a threshold of about 20 µM (microbial hypoxia), whereas anoxic

processes dominate the nitrogen cycle at oxygen concentrations of < ~0.05 µM

(functional anoxia). The Arabian Sea and the Bay of Bengal are home to

approximately 21% of the total volume of ocean waters revealing microbial hypoxia.

While in the Arabian Sea this oxygen minimum zone (OMZ) is also functionally

anoxic, the Bay of Bengal OMZ seems to be on the verge of becoming so. Even

though there are a few isolated reports on the occurrence of anoxia prior to 1960,

anoxic events have so far not been reported from the open northern Indian Ocean

(i.e. other than on shelves) during the last 60 years. Maintenance of functional anoxia

in the Arabian Sea OMZ with oxygen concentrations ranging between > 0 and ~0.05

µM is highly extraordinary considering that the monsoon reverses the surface ocean

circulation twice a year and turns vast areas of the Arabian Sea from an oligotrophic

oceanic desert into one of the most productive regions of the oceans within a few

20  weeks. This implies stable balances between the physical oxygen supply and the

biological oxygen consumption, which includes negative feedback mechanisms

reducing the biological oxygen consumption at decreasing oxygen concentrations. A

lower biological oxygen consumption is also assumed to be responsible for a less

intense OMZ in the Bay of Bengal. According to numerical model results, a

decreasing physical oxygen supply via the inflow of water masses from the south

intensified the Arabian Sea OMZ during the last 6000 years, whereas a reduced

oxygen supply via the inflow of Persian Gulf Water from the north intensifies the OMZ

today in response to global warming. The first is supported by data derived from the

sedimentary record, and that latter concurs with observations of decreasing oxygen

concentrations and a spreading of functional anoxia during the last decades in the

Arabian Sea. In the Arabian Sea this seems to have initiated a regime shift within the

pelagic ecosystem structure, and the trend also affects benthic ecosystems.

Consequences for biogeochemical cycles are unknown, which, in addition to the poor

representation of mesoscale features, reduces the reliability of predictions of the

future OMZ development in the northern Indian Ocean.

## 1. Introduction

The rise of atmospheric oxygen to nearly present-day levels was a precondition for the evolution of complex life forms and accompanied the appearance of algae and planktonic cyanobacteria at about 800 to 500 million years ago (Brocks et al., 2017; Canfield, 2014 ; Lenton et al., 2011; Lyons et al., 2014; Sánchez-Baracaldo, 2015). Numerically, planktonic cyanobacteria are still the most abundant plankton clade in the ocean and exert a strong control on the energy transfer into the marine biosphere by transforming nitrogen gas ($N_2$) into ammonium ($NH_4^+$) (nitrogen fixation, Fig. 1, Falkowski et al., 2004). Since algae, which in addition to cyanobacteria comprise marine primary producers, cannot fix nitrogen gas, they rely on the supply of fixed nitrogen ($NH_4^+$, $NO_2^-$, $NO_3^-$) for the production of organic matter. Energy yielded by the respiration of organic matter produced by cyanobacteria and algae sustains heterotrophic life in the ocean, while chemoautotrophic organisms oxidize degradation products such as methane and ammonium to gain energy for running their metabolisms (Dalsgaard et al., 2003; Kuypers et al., 2001; Middelburg, 2011). In the absence of elementary oxygen, oxygen bound to sulfur (e.g. sulfate) or nitrogen (nitrate and nitrite), can also be utilized to oxidize organic matter and its degradation products. Since nitrate and nitrite are (like ammonium) accessible to algae, their use as oxidizing agents reduces the availability of fixed nitrogen in the ocean. Accordingly, decreasing oxygen concentrations could exert a negative feedback on marine primary production by lowering the availability of fixed nitrogen for algal production (Canfield et al., 2019; McElroy, 1983).

The transition from anaerobic to aerobic systems occurs in steps, whereas in the absence of elementary oxygen (anoxia) methane is oxidized to $CO_2$ by the reduction of sulfate to hydrogen sulfide. Such fully anoxic conditions emerge only rarely in the ocean, but appear to be common in microenvironments within particles via which organic matter, which is produced in the sunlit surface ocean, is exported into the deep sea (e.g. Bianchi et al., 2018; Naqvi et al., 2000; Weeks et al., 2002). However, at low levels of dissolved oxygen, aerobic and anaerobic processes occur simultaneously and compete against each other. Chemoautotrophic microbes use the available elementary oxygen to oxidize ammonium to nitrite and further to nitrate (nitrification: $NH_4^+$ -> $NO_2^-$ –> $NO_3^-$), while microbes carrying out anaerobic processes transform ammonium as well as nitrate and nitrite into $N_2$ (Fig. 1). Among these anaerobic processes heterotrophic denitrification and the chemoautotrophic anaerobic oxidation of ammonium (anammox) are the most relevant. Denitrification reduces nitrate in a sequence of several steps via nitrite to $N_2$ ($NO_3^-$ -> $NO_2^-$ -> $N_2$),

whereas anammox bacteria utilize nitrite to oxidize ammonium ($NO_2^- + NH_4^+ -> N_2$).

Thus, nitrite plays an import role in the competition between these anaerobic and

aerobic processes because, independent of its formation via nitrification and

denitrification, it can either be oxidized to nitrate or reduced to $N_2$. Concentrations of

dissolved oxygen strongly influence the fate of nitrite and thereby exert a control on

the availability of fixed nitrogen in the ocean (Bristow et al., 2016; Gaye et al., 2013).

According to experiments and *in situ* observations, anammox sets in when oxygen

concentrations drop below ~20 µM, while denitrification occurs at oxygen

concentrations of approximately < 6 µM (Fig. 2, Bristow et al., 2016; Dalsgaard et al.,

2014; Kalvelage et al., 2011). Consequently, anammox competes with nitrification for

nitrite at oxygen level between 6 and 10 µM, and additionally with denitrification at

oxygen level < 6 µM. Decreasing oxygen concentrations favor anammox and

denitrification while, in addition to the influence of oxygen, the quality of the supplied

organic matter appears to also control the relative importance of denitrification vs.

anammox for the reduction of nitrite to $N_2$ (Babbin et al., 2014; Bristow et al., 2016;

Ward et al., 2009). However, since denitrification and anammox ultimately produce

$N_2$ at the expense of fixed nitrogen, the term denitrification is used as a synonym for

both processes in the following discussion if anammox is not specifically mentioned.

Hypoxia, which means 'low oxygen concentrations', describes oxygen concentrations

below which higher organisms start to suffer from the lack of oxygen (Ekau et al.,

2010; Vaquer-Sunyer et al., 2008).  Accordingly, an oxygen concentration of 60 – 63

µM is commonly applied as upper limit of hypoxia in fisheries and ecology. Since

such high threshold values do not reflect oxygen-dependent changes in the nitrogen

cycle, it is suggested to subdivide hypoxia into microbial hypoxia and functional

anoxia (Fig. 2). Functional anoxia was defined already in other works (Canfield et al.,

2019; Thamdrup et al., 2012) and covers oxygen levels below which denitrification

dominates the nitrogen cycle. Microbial hypoxia is suggested here as the range at

which decreasing oxygen levels progressively offset the oxygen-inhibition of

denitrification. Since this starts with the occurrence of anammox, we consider 20 µM

as the upper threshold of microbial hypoxia whereas anoxia (zero oxygen) terminates

hypoxia and therewith also functional anoxia and microbial hypoxia. Because oxygen

detection limits of classical Winkler titration (~1 µM), seabird sensors (~0.09 µM) and

the newly developed switchable trace oxygen sensors (STOX, ~0.01 µM) are too

high to prove anoxia (Thamdrup et al., 2012; Ulloa et al., 2012) the appearance of

hydrogen sulfide is generally considered as an indicator of anoxia.

Since hypoxic and anoxic conditions repress and prevent, respectively, the growth of

all life forms that depend on oxygen, their occurrence is often associated with mass

mortality of fish and invertebrates (e.g. Weeks et al., 2002). Thus, the spatial
expansion of hypoxia, which seems to be an increasingly common feature in coastal
waters, is called the "spreading of dead zones" (Altieri et al., 2017; Diaz et al., 2008).
This term ignores the life of anaerobic microbes, which dominated the world's
biosphere long before oxygen-breathing organisms evolved, but expresses the threat
of oxygen-depletion to many commercially important marine species. Eutrophication
and global warming mainly cause the spread of dead zones whereas eutrophication
increases oxygen consumption by enhancing the production of organic matter, while
global warming decreases the oxygen supply due to a reduced solubility of oxygen in
warmer waters. Since decreasing concentrations of dissolved oxygen have also been
widely observed in the open ocean (i.e. beyond coastal systems) during the last 50
120    years, deoxygenation of the ocean is considered as one of the main threats to
pelagic ecosystems (Breitburg et al., 2018; Keeling et al., 2009; Schmidtko et al.,
2017; Stramma et al., 2010a; Stramma et al., 2008; Stramma et al., 2010b).

Based on data obtained from the World Ocean Atlas, the total volume of waters
characterized by oxygen concentrations < 20 µM in the global ocean is
approximately $15 \times 10^{15}$ m$^3$, of which 21% ($3.13 \times 10^{15}$ m$^3$) is located in the northern
Indian Ocean (Fig. 3c, Acharya et al., 2016; Garcia et al., 2010). The largest
proportion of this oxygen-poor water body is in the Arabian Sea ($2.5 \times 10^{15}$ m$^3$) and
only a small fraction in the Bay of Bengal ($0.6\ 10^{15}$ m$^3$). In comparison to the Bay of
Bengal OMZ, with a mean concentration of dissolved oxygen of 14.51 µM, the
Arabian Sea OMZ is more intense, as indicated by mean concentration of dissolved
oxygen of 10.45 µM (Acharya & Panigrahi, 2016). Notably, in regions where these
OMZs impinge on continental margins, sediments and benthic communities are
exposed to semi-permanent bottom-water hypoxia. The Arabian Sea and the Bay of
Bengal together are currently home to ~59% of the Earth's marine sediments
exposed to hypoxia (Helly et al., 2004).

Denitrification, within sediments (benthic denitrification) and under hypoxic conditions
in the water column, is by far the largest sink of nitrate in the ocean (Gruber, 2004).
However, estimates of benthic and water column denitrification rates are still fraught
with large uncertainties on global as well as on regional scales. Global scale
estimates of benthic and water-column denitrification range between 65 and 300 Tg
N year$^{-1}$ and 39 and 270 Tg N year$^{-1}$, respectively (Eugster et al., 2012; Gruber,
2004; Somes et al., 2013). Although denitrification in the Arabian Sea has been
much more intensively studied than in the Bay of Bengal, estimates of benthic (1 to
6.8 Tg N year$^{-1}$) and water column (1 to 33 Tg N year$^{-1}$) denitrification in the Arabian
still reveal a wide range (Bange et al., 2000; Bristow et al., 2017; Deuser et al., 1978;

Gaye et al., 2013; Howell et al., 1997; Naqvi et al., 1982; Somasundar et al., 1990).

However, these estimates imply that on average the Arabian Sea contributes

approximately 2 % and 11 % to the global mean benthic and water column

denitrification, respectively, although published data indicate that the estimated

benthic denitrification rates might be too low. According to these more recent data,

the benthic denitrification at the Pakistan continental margin amounts to up to 10.5

152     Tg N year$^{-1}$ (Schwartz et al., 2009; Somes et al., 2013), which exceeds the former

budget (6.8 Tg N year$^{-1}$) of the entire Arabian Sea sediments (Bange et al., 2000). In

line with a severe depletion of nitrate in bottom waters on the Indian shelf (Naqvi et

al., 2010) this implies that benthic denitrification rates contribute > 2% to the global

mean benthic denitrification rate. This further emphasizes the role of the northern

Indian Ocean OMZ for the marine nitrogen cycle, which was considered as one of

the least understood OMZs in the world's ocean (Schmidt et al., 2020; Segschneider

et al., 2018). The aim of this paper is to provide a short background on the

development of OMZs and recent trends in the OMZ of the Indian Ocean as well as

to discuss drivers, ecosystem responses and the future development of this OMZ.

**2. Background**

**2.1 Oxygen Minimum Zones (OMZ)**

The first large ocean-going oceanographic expeditions discovered OMZs in Pacific,

Atlantic, and Indian Ocean between the end of the 19[th] and the first third of the 20[th]

century (Sewell et al., 1948 and references therein). Their occurrence was explained

by a sluggish horizontal renewal of water within the OMZ and the consumption of

oxygen during the respiration of organic matter exported from the sunlit surface

ocean (Dietrich, 1936; Seiwell, 1937). Sverdrup (1938) presented the first model

showing that oxygen concentrations within the OMZ represent the balance between

biological oxygen consumption and oxygen supply. Primary production and fluxes of

oxygen across the air-sea interface are the sources of dissolved oxygen. Vertical

mixing and subduction of oxygen-enriched surface waters during the deep and mode

water formation at high latitudes are in turn the main processes ventilating the interior

of the ocean (McCartney, 1977; Sverdrup, 1938). Accordingly, Broecker and Peng

(1982) introduced a model in which upwelling of oxygen-enriched deep water and

vertical mixing serve to ventilate the OMZ from the bottom and the top, respectively,

whereas the respiration of exported organic matter decrease oxygen concentrations

in the OMZ (Fig. 4). The OMZ in general emerges at water-depths between ca. 100

and 1000 m because this is the depth range at which the majority of the exported

organic matter is respired (Suess, 1980). Due to the respiration of exported organic

matter, oxygen concentrations decrease during the spreading of water masses in the

ocean and pronounced OMZs occur in these so-called 'shadow zones', where aged

water masses accumulate (Karstensen et al., 2008).

Since oxygen concentrations result from the balance between oxygen consumption

and supply, which in turn also influence these processes, a dynamic evolved such

that an enhanced oxygen supply does not necessarily result in increased oxygen

concentrations. Although, in principle, an enhanced physical oxygen supply

increases oxygen concentrations, an enhanced oxygen supply caused by increased

oxygen consumption can be associated with lower oxygen concentrations. The

underlying mechanism can be illustrated with a simple conceptual steady-state box

model (Fig. 4). Within this model, the OMZ is a box with a constant volume and

oxygen concentrations at steady-state due to an oxygen consumption that equals the

oxygen supply. Vertical mixing links the oxygen concentration in the OMZ to those at

the sea surface and therewith to the flux of oxygen across the air-sea interface. The

air-sea flux of oxygen is proportional to the oxygen partial pressure difference

between the ocean and the atmosphere and increases in response to decreasing

oxygen concentrations in the ocean. Accordingly, increased oxygen consumption can

enhance the flux of oxygen from the atmosphere into the ocean by decreasing

oxygen concentrations in the OMZ. Thereby, an enhanced oxygen supply balances

an increased oxygen consumption and re-establishes a steady-state that was

disturbed by the increase of the oxygen consumption. Steady-state oxygen

concentrations resulting from such a perturbation are inversely proportional to the

oxygen consumption/supply. They rise or fall if oxygen consumption/supply decrease

or increase, respectively.

The main processes controlling the physical oxygen supply include temperature- and

salinity-related changes of the solubility of oxygen in water, which are in turn related

to factors such as variation in ocean circulation and water movements associated

with mesoscale eddies. For instance, changes in the ocean's circulation can increase

oxygen concentrations by enhancing the lateral inflow of oxygen-enriched water

masses into the OMZ. The increased oxygen supply is in turn balanced by an

enhanced outflow of oxygen along with water, which compensates the inflow.

Secondly the enhanced oxygen concentrations lower the supply of oxygen from the

atmosphere by lowering the air-sea flux of oxygen as discussed before. Thereby, a

new steady state is established, but at a higher oxygen level. The impacts of

mesoscale eddies are more difficult to describe as they develop from baroclinic and

barotrophic instabilities related to the shear of horizontal currents, and create a

patchiness of environmental conditions with complex and non-linear impacts on the

OMZ and the marine carbon cycle (McWilliams, J.C., 2008, McGillicuddy Jr, D.J.,
2016, Fassbender et al., 2018). In order to elucidate their role for the development of
OMZs, multidisciplinary approaches involving *in situ* observations, remote sensing,
and modeling are required (McGillicuddy, 2016). Results obtained from numerical
model studies in the Indian Ocean (Lachkar et al., 2019; Lachkar et al., 2016;
McCreary Jr et al., 2013; Resplandy et al., 2012) will be introduced in a separate
section. Nevertheless, oxygen concentrations as observed in the OMZ of the Arabian
Sea and Bay of Bengal can be seen as steady-state concentration. This implies that
higher oxygen concentrations in the Bay of Bengal OMZ relative to that of the
Arabian Sea reflect a lower oxygen consumption or a higher physical oxygen supply.
In order to test these two options, we will first of all provide a general overview about
processes controlling oxygen consumption and the physical oxygen supply in the
Arabian Sea and Bay of Bengal.
**2.2 OMZ and upwelling**
In the Atlantic and Pacific Oceans, hypoxic OMZs are associated with major eastern
boundary current upwelling systems. They are among the most productive regions in
the ocean and account for circa 20% of the global fish catch (Carr, 2001; Pauly et al.,
1995). The high productivity emphasizes the role of oxygen consumption for the
development of OMZ. In the Indian Ocean, the geographic setting prevents the
development of a strong eastern boundary current upwelling system but a major
monsoon-driven seasonal upwelling system emerges in the western Arabian Sea off
the Arabian Peninsula (Fig. 3a). Initially described by Schott (1935), this upwelling
system was subject to intense studies including the International Indian Ocean
Expedition (IIOE) between 1959 and 1965 and the Joint Global Ocean Flux Study
(JGOFS) in the Arabian Sea with its field phase between 1994 and 1997 (e.g. Bauer
et al., 1991; Brock et al., 1991; Bruce, 1974; Currie et al., 1973; Sastry et al., 1972;
Wyrtki, 1973).
High monsoonal upwelling-driven productivity sustains a high export of organic
matter into the deep sea (Haake et al., 1993; Rixen et al., 1996) but the impact of
upwelling is not restricted to coastal areas because upwelling–driven blooms advect
offshore and increase the organic carbon export in the central Arabian Sea (Rixen et
al., 2006a). Nevertheless, against expectations, the OMZ is most intense in the
eastern Arabian Sea and not in western Arabian Sea where the upwelling-driven
productivity is highest (Fig. 4c, Antoine et al., 1996; Naqvi, 1991). The seasonal
monsoon-driven reversal of the surface circulation in combination (Fig. 5) with the a
strong inflow of oxygen-enriched Indian Ocean Central Water (ICW) into the western

Arabian Sea is assumed to cause this eastwards displacement of the OMZ in the
Arabian Sea (Rixen et al., 2005; Sen Gupta et al., 1984; Swallow, 1984). The ICW
originates by convective mixing as Subantarctic Mode Water (SAMW) in the southern
Indian Ocean (McCartney, 1977; Sverdrup et al., 1942) and enters the western
Arabian Sea along with Timor Sea Water and the Subtropical Subsurface Water via
the Somali Current (Schott et al., 2001; Stramma et al., 1996; You, 1997). The inflow
of the ICW is part of the meridional overturning circulation of the upper Indian Ocean,
which includes a cross-equatorial cell, driven by downwelling in the south and
upwelling in the north (Fig. 6, Rahaman et al., 2020).

In addition to the strong upwelling off the Arabian Peninsula, a weaker summer
monsoon-driven upwelling develops along the Indian southwest coast (Fig. 3a,
Schott, 1935; Sharma, 1978; Shetye et al., 1990). Here an undercurrent emerges,
which compensates the poleward flowing West Indian Coastal Current (Fig 5a) and
carries the ICW northwards into the eastern Arabian Sea (Schmidt et al., 2020;
Shenoy et al., 2020; Shetye et al., 1990). Despite the inflow of ICW, the OMZ
expands southwards during the summer monsoon, and retreats northwards during
the winter monsoon in the eastern Arabian Sea (Shenoy et al., 2020). In contrast to
these movements, the OMZ retreats eastwards in summer and expands westwards
in winter in the western Arabian Sea (Rixen et al., 2014). This opposing behavior
creates a seesaw billowing of the OMZ as a consequence of the seasonal reversal of
the surface ocean circulation - anticlockwise during the winter monsoon and
clockwise during summer monsoon.

Influences of the seasonally reversing surface ocean circulation on the volume and
intensity of the OMZ were studied by Acharya and Panigrahi (2016). These authors
analyzed the Word Ocean Atlas 2013 (WOA13) and data compiled by the Global
Ocean Data Analysis Project. Thereby, they took methodical biases into account,
corrected the WOA13 data, and applied the 20-µM threshold to define the OMZ.
Based on the corrected data set, these authors determined the areal extension and
depth of the OMZ and calculated the mean oxygen concentration within the OMZ.
According to this data, the OMZ reveals its lowest areal extension in summer (Fig.
7a) due to the inflow of ICW, which favors the eastward retreat of the OMZ in the
western Arabian Sea and attenuates the influence of the clockwise circulation and
the associated southwards expansion of the OMZ in the eastern Arabian Sea. Since
the shrinking spatial extension of the OMZ is associated with a deepening, the OMZ
reveals its largest volume in summer, which in turn is accompanied by the lowest
concentration of oxygen within the OMZ (Fig. 7b). The deepening of the OMZ is
favored by the negative wind stress curl and the associated downwelling in the

central Arabian Sea, and is coincident with the confluence of ICW that enters the
Arabian Sea in west and east (Brock et al., 1991; Rao et al., 1989).

The low oxygen concentration reflects in turn impacts of the enhanced upwelling-
driven productivity and the associated increased oxygen consumption in the OMZ,
which apparently exceeds the increased physical oxygen supply via the inflow of
ICW. According to the conceptual box model, such a situation represents a steady-
state characterized by high oxygen consumption and air-sea oxygen supply that
establishes in response to the monsoon-driven increase of the biological production.
Thereby the enhanced supply of oxygen via the inflow of ICW acts as an additional
oxygen sources, which lowers the impact of the high oxygen consumption on the air-
sea oxygen supply. In order to test whether the monsoon establishes steady states
and controls therewith the intensity of the OMZ via its impact on the biological
production, satellite-derived primary production values were obtained from ocean
primary production website (http://www.science.oregonstate.edu/ocean.productivity/).
The data covered the period between 2002 and 2019 and were averaged seasonally
for the Arabian Sea north of 10°. In general, the obtained mean primary production
rates were inversely linked to the OMZ oxygen concentrations (Fig. 7b, 8a), which
supports the assumption that monsoon-driven changes in the biological carbon
production and export exert a strong control on the intensity of the OMZ on a
seasonal time scale. However, the fall appears to be an exception. During this
season the OMZ oxygen concentration was lower as expected from the primary
production rate (Fig. 8a) indicating a physical oxygen supply, which was too low to
establish a steady-state. One explanation could be the strongly reduced inflow of
ICW in combination with a low air-sea oxygen supply in response to the warming and
the resulting enhanced stratification of surface waters after the upwelling season in
fall.

Similar to the Arabian Sea, upwelling-favorable winds also occur in the Bay of Bengal
during the summer monsoon (Hood et al., 2017; Shetye et al., 1988). However, high
freshwater fluxes, from both river runoff and precipitation, form a buoyant low-salinity
surface layer that isolates nutrient-enriched subsurface water and increases
stratification (Kumar et al., 1996). This weakens upwelling and vertical mixing, and
thus nutrient supply into the sunlit surface ocean (Rixen et al., 2006b). Hence,
productivity in the Bay of Bengal is lower than in the Arabian Sea and reveals only a
weakly pronounced seasonality (Fig. 3a,b). Nevertheless, sediment trap studies have
shown that, despite a lower productivity, organic carbon fluxes into the deep Bay of
Bengal are almost as high as those in the central and eastern Arabian Sea, due to a
ballast effect associated with high loadings of lithogenic mineral material (Rao et al.,

1994; Rixen et al., 2019b). Ballast minerals, supplied from land via rivers or as dust,
protect organic matter against bacterial attacks by adsorbing organic molecules to
atomic lattices (Armstrong et al., 2002) and accelerate the sinking speed of particles
(Haake et al., 1990; Hamm, 2002; Ramaswamy et al., 1991). Enhanced sinking
speeds reduce respiration in shallower waters and thereby increase the flux of
organic matter to deeper waters (Banse, 1990; Ittekkot, 1993). A stronger ballast
effect, in addition to the lower primary production, was assumed to lower the oxygen
consumption in the Bay of Bengal in comparison to that in Arabian Sea OMZ (Al
Azhar et al., 2017; Rao et al., 1994). On the other hand, in contrast to the Arabian
Sea, a buoyant low-salinity layer reduces the oxygen supply via vertical mixing.
Furthermore, instead of oxygen-enriched ICW, oxygen-poor Arabian Sea Water flows
into the Bay of Bengal OMZ, as indicated by a broad salinity maximum that occurs
below the low-salinity surface layer, to a water depth of approximately 1000 m (Rao
et al., 1994 and references thereine). Hence, higher oxygen concentrations in the
Bay of Bengal relative to the Arabian Sea can be seen as a consequence of the
lower oxygen consumption, which is balanced by a lower oxygen supply.

**2.3 Recent trends in the Bay of Bengal and the Arabian Sea**

In contrast to the monsoon, which seems to influence the intensity of the OMZ mainly
by its impact on the productivity and resulting biological oxygen consumption, global
warming is assumed to intensify OMZs through its influence on the solubility of
oxygen in water and the resulting physical oxygen supply. However, there are
indications that at least the Arabian Sea OMZ was more intense in the past, prior to
the Indian Ocean Expedition (IIOE, 1959 - 1965), than thereafter. For instance,
Carruthers et al (1959) described mass mortality of fish along the Arabian and Indian
coast as well as in the central Arabian Sea at around 62.5°E and 9 °N. The
development of harmful algae blooms were discussed but oxygen depletion was
preferred as the more likely mechanism explaining these mass mortalities
(Carruthers et al., 1959). This view was supported by a report on the occurrence of
hydrogen sulfide from the north-eastern Arabian Sea and off Oman at Ras-al-Hadd
(Ivanenkov  et al., 1961). These were the only reports on the occurrence of hydrogen
sulfide in Arabian Sea (Naqvi et al., 2000; Swallow, 1984) except for Naqvi et al.
(2000) who discovered an anoxic event that developed along the western Indian
coast off Mumbai in the late summer of 1999. Such strong events do not evolve
every year (Gupta et al., 2016; Sudheesh et al., 2016), but their appearance shows
that the spreading of 'dead zones' in coastal regions does not spare the Indian shelf
(Altieri et al., 2017; Diaz & Rosenberg, 2008; Diaz et al., 2019).

In contrast to these shelf processes, global syntheses of OMZs (beyond continental shelves) reveal only a weak decrease of dissolved oxygen concentrations in the OMZs of the Arabian Sea and the Bay of Bengal in comparison to OMZs of the South Atlantic Ocean and the Pacific Ocean (Ito et al., 2017; Naqvi, 2019; Schmidtko et al., 2017; Stramma et al., 2008). The detailed analysis of all oxygen data available from the central Arabian Sea by Banse et al. (2014) ascribes this to opposing regional trends within the Arabian Sea. The authors analyzed oxygen data, which were measured between 1959 and 2004 in the Arabian Sea and in the depth range between 100 and 500 m. Biases caused by different analytical procedures were taken into account and oxygen data were compiled for sub-regions within the Arabian Sea. The results showed that oxygen concentrations increased in the southern part of the Arabian Sea and declined in the central Arabian Sea. Follow-up studies also reported decreasing oxygen concentrations in the western and northern Arabian Sea (Piontkovski et al., 2015; Queste et al., 2018). In the northern Arabian Sea, dissolved oxygen concentrations in the surface mixed layer largely reflect the trend seen in the OMZ, as indicated by a compilation of dissolved oxygen data covering the period from the 1960s to 2010 (Gomes et al., 2014).

Since STOX data for the Arabian Sea were unavailable prior to 2007, the data compiled by Banse et al. (2014) do not resolve any changes in the oxygen concentrations below 0.09 µM (Fig. 2). Such low oxygen concentrations have recently been measured in the Arabian Sea and in the Bay of Bengal as well as in OMZ of the eastern Pacific Ocean (Bristow et al., 2017; Jensen et al., 2011; Thamdrup et al., 2012). In the latter study, it was shown that nitrite accumulates in the water column at oxygen concentrations of < 0.05 µM. In the Arabian Sea OMZ, the accumulation of nitrite was first described during the John Murray expedition of 1933 - 34 (Gilson, 1937). It is called the secondary nitrite maximum (SNM) and assumed to indicate active denitrification (Naqvi, 1991). The role of the SNM as indicator of active denitrification is further supported by profiles of stable isotope ratios of nitrogen in nitrate ($\delta^{15}N_{NO3}$) and nitrate ($NO_3^-$) concentration profiles (Fig. 9, Gaye et al., 2013; Rixen et al., 2014). Since denitrification increases $\delta^{15}N_{NO3}$ in the water column due to the preferential uptake of the lighter $^{14}NO_3^-$ (Cline et al., 1975; Mariotti et al., 1981), low nitrate concentrations correspond to high $\delta^{15}N_{NO3}$ within the SNM. Assuming a similar response of the nitrogen cycle to low oxygen concentration in the Arabian Sea as in the eastern Pacific Ocean suggests that the Arabian Sea SNM is characterized by oxygen concentrations of about ~0.05 µM, and in turn that denitrification dominates the nitrogen cycle at such low oxygen concentrations. Nevertheless, an isotope tracer study indicated that the re-oxidation of nitrite to

nitrate reduced the formation of $N_2$ by 50 to 60% (Gaye et al., 2013), which implies an active competition between aerobic and anaerobic processes even at such low oxygen concentrations. However, since anaerobic denitrification dominates the competition, as indicated by the accumulation of nitrite, the depletion of nitrate and the $\delta^{15}N_{NO3}$ maxima, an oxygen concentration of ~0.05 μM is seen as a threshold below which functional anoxia occurs (Fig. 2).

Naqvi (1991) was the first to use the SNM to map the spatial extent of functional anoxia in the Arabian Sea. His analysis was based on data obtained during the International Indian Ocean Expeditions between 1959 and 1965 and cruises thereafter but did not include data from Ivanenkov and Rozanov (1961) and JGOFS. The data of Ivanenkov and Rozanov (1961) indicate a more intense OMZ, including the occurrence of hydrogen sulfide as mentioned earlier and a larger extent of the SNM than calculated by Naqvi (1991), although there are doubts regarding the reliability of the older data (Sen Gupta & Naqvi, 1984). Comparison of JGOFS data, collected in 1994/95, with those compiled by Naqvi (1991), shows that the SNM has expanded south- and westwards (Rixen et al., 2014). This implies, in accordance with decreasing oxygen concentrations, an expansion of the OMZ in the Arabian Sea, which might have started in the early 1990s. However, the reliability of this trend has been questioned by Naqvi (2019). While it is acknowledged that the data are too sparse to have unquestionable confidence in this trend, it is difficult to assess the doubts raised by Naqvi (2019), as these are based on data derived from sediment cores that do not cover the most recent past.

In the Bay of Bengal, a pronounced SNM has not yet been detected, although a recent study presented data from 7 stations in the northern Bay of Bengal and revealed oxygen concentrations which partly drop below 0.05 ~μM at four sites (Bristow et al., 2017). Incubation experiments were carried but with, one exception, these failed to prove denitrification. During the one exception, a denitrification rate of 0.9 nmol L$^{-1}$ day$^{-1}$ was measured, which falls much below denitrification rates of > 20 nmol L$^{-1}$ day$^{-1}$ as measured by Ward et al.(2009) in the Arabian Sea. However, these results indicate that the Bay of Bengal OMZ is on the verge of functional anoxia, with re-oxidation of nitrite to nitrate as yet preventing significant denitrification (Bristow et al., 2017). However, outbreaks of hydrogen sulfide as seen in the upwelling systems off Peru (Schunck et al., 2013) and Namibia (Weeks et al., 2002) have so far not been reported in the northern Indian Ocean during the last 50 years, other than in bottom waters on the Indian shelf as mentioned earlier. This implies that the physical oxygen supply and the biological oxygen consumption maintained hypoxic conditions

439 and prevented persistent anoxia in the Arabian Sea and functional anoxia in the Bay
of Bengal OMZ.

## 3. Export production and its controlling effect on the intensity of the OMZ

### 3.1 Interplay between the intensity of the OMZ and export production

Maintenance of functional anoxia in the Arabian Sea OMZ with oxygen concentrations ranging between > 0 and ~0.05 µM is highly extraordinary considering that the monsoon-driven seasonality reverses the surface ocean currents and turns the Arabian Sea from an oligotrophic oceanic desert into one of the most productive regions in the world's ocean within weeks. This calls for feedback mechanisms counteracting impacts of the monsoon on the intensity of the OMZ. Since there is growing evidence that concentrations of dissolved oxygen influence the respiration of organic matter in the water column (Aumont et al., 2015; Cavan et al., 2017; Laufkötter et al., 2017; Thamdrup et al., 2012; Van Mooy et al., 2002), it has been assumed that an oxygen–related feedback mechanism stabilizes the Arabian Sea OMZ (Rixen et al., 2019a). This feedback mechanism is evident if one considers the sunlit surface mixed layer and the seasonal thermocline, which serves as the habitat of the pelagic ecosystems (Fig. 10).

The seasonal thermocline is the subsurface layer from which water is introduced into the euphotic zone via physical processes such as upwelling and vertical mixing on a seasonal timescale. Nutrients supplied by these mechanisms largely sustain the productivity of pelagic ecosystems and the associated export production (Eppley et al., 1979). Hence, the seasonal thermocline is the main nutrient reservoir of pelagic ecosystems, and to fulfill this role the vast majority of the exported organic matter must be respired within the seasonal thermocline. Accordingly, the seasonal thermocline represents the main zone of respiration and, similar to soils on land, accommodates the nutrient recycling machinery of the pelagic ecosystem. Nutrient losses from the seasonal thermocline, via particle fluxes into the deep sea, denitrification, and lateral advection, must be compensated by nutrient inputs in order to maintain the productivity (Rixen et al., 2019a). Nitrogen fixation, river discharges, and atmospheric deposition can be important nutrient sources, but in the Arabian Sea lateral inflow of water masses from the south via the cross-equatorial cell are the main source balancing nutrient losses from the seasonal thermocline (Bange et al., 2000; Gaye et al., 2013). Accordingly, a significant negative impact of denitrification on primary and export production and the associated oxygen consumption appear to be unlikely on seasonal to centennial time scales in the Arabian Sea.

The SNM, which occurs at water depths between 200 and 400 m in the central
Arabian Sea and as deep as 500 m in the eastern Arabian Sea, divides the seasonal
thermocline into an aerobic part at water depths between ~40 and 200 m and an
anaerobic part down to the base of the SNM (Fig. 10). The depth of the seasonal
thermocline of approximately 300 – 400 m corresponds to the depth range of
vertically migrating zooplankton as observed during the large summer bloom in the
Arabian Sea (Smith, 2001), and roughly matches the water depth range from where
subsurface water is introduced via upwelling into the euphotic zone in the western
Arabian Sea (Brock et al., 1992; Rixen et al., 2000). Furthermore, nitrate
concentrations, which decrease within the SNM, remain above 10 µM (Fig. 9),
suggesting that supply of decomposable organic matter (rather than nitrate
availability) limits denitrification, as also suggested by other studies (Bristow et al.,
2016; Ward et al., 2009). A substrate limitation at a water depth of 400 to 500 m, and
the arrival of organic matter at sediment traps deployed in the deep sea at a water
depth of 3000 m, support the concept of export production that is divided between
free (reactive) and protected (low reactivity) organic matter (Armstrong et al., 2002).
This partition is based on the assumption that ballast-associated, protected organic
matter is preferentially exported to deeper waters as fast sinking particles, whereas
the slow sinking free organic matter is preferentially respired within the seasonal
thermocline.

Similar to the ballast-effect also decreasing oxygen concentrations could increase
the organic carbon flux into the deep sea. This is assumed to be caused by the
negative impact of low oxygen concentrations on the respiration rate (Aumont et al.,
2015; Laufkötter et al., 2017; Thamdrup et al., 2012; Van Mooy et al., 2002) and the
low zooplankton abundance in the OMZ (Cavan et al., 2017), which will further be
discussed in the following section. However, a reduced respiration due to decreasing
oxygen concentrations prevents further depletion of oxygen and the development of
anoxia in the SNM by lowering the oxygen consumption and increasing the export of
organic matter out of the seasonal thermocline. An increased export of carbon into
the deep sea could, in turn, favor a deepening of the OMZ by enhancing the oxygen
consumption at the base of the OMZ at water-depths between 1150 m and 1240 m.
This would imply that low OMZ oxygen concentrations are accompanied by a
deepening of the OMZ. Data compiled by Acharya and Panigrahi (2016) support this
view by showing the link between primary production and OMZ oxygen
concentrations as discussed before and furthermore a correlation between the
seasonal mean oxygen concentrations within the OMZ and the depth of the OMZ
(Fig. 8), even though the latter is also influenced by physical processes as also

discussed above. Accordingly, it seems that the influence of decreasing oxygen concentrations on respiration within the SNM mitigates impacts of monsoon-driven primary and export production on the intensity of the OMZ.

**3.12 Biological oxygen consumption**

The role of an oxygen-dependent, biologically driven feedback mechanism, counteracting impacts of the monsoon on the intensity of the OMZ, depends on the local oxygen consumption. If this is too low, remote processes instead of the local monsoon-driven carbon export rates need to be considered as forces controlling the intensity of the OMZ. The apparent oxygen utilization (AOU), in addition to mixing analyses, provide information that helps to estimate the role of the local oxygen consumption for maintaining the intensity of the OMZ in the Arabian Sea. The AOU represents the oxygen deficit caused by biological oxygen consumption and is calculated by subtracting the measured oxygen concentration from the temperature- and salinity-dependent oxygen saturation concentration. This approach is based on the assumption that the water mass of interest was saturated with respect to oxygen during its formation and, since then, the respiration of exported organic matter consumed oxygen within the water mass. The OMZs in the northern Indian Ocean are melting pots collecting the influence of a variety of water masses with different origins and histories (e.g. Morcos et al., 2012; Schott & McCreary, 2001; You, 1997). Mixing analyses indicate that in the Arabian Sea the inflow of oxygen-rich Indian Ocean Deep Water affects the lower OMZ (water-depth > 500 m) as already discussed before (Fig. 3, Acharya & Panigrahi, 2016; Hupe et al., 2000; Rixen & Ittekkot, 2005). In addition to the formation of the Arabian Sea Water (ASW), the intrusion of ICW and water masses from the Persian Gulf and Red Sea strongly influenced the upper OMZ. A mixing analyses based on data measured during JGOFS in 1994/95 reveals that oxygen-deficits inherited from ICW contribute approximately 25% to the AOU determined in the Arabian Sea OMZ (Rixen & Ittekkot, 2005). Accordingly, the respiration of organic matter produced in the Arabian Sea largely causes the low oxygen consumption in the Arabian Sea OMZ, which further supports the hypothesis that monsoon-driven productivity, including the oxygen-dependent biologically driven feedback mechanism, controls and stabilizes the OMZ in the Arabian Sea.

However, in the Arabian Sea, mean satellite-derived export production rates were too low to sustain a high biological oxygen consumption considering a residence time of water within the OMZ of 10 years (Rixen & Ittekkot, 2005). The mismatch reflects uncertainties caused by the poorly constrained residence time of water within the

OMZ and export production rates. Even though residence times of water within the
Arabian Sea and Bay of Bengal OMZ of 10 and 12 years seems to be well accepted
(Bristow et al., 2017; Olson et al., 1993), there also are estimates ranging from 1 to
51 years for the Arabian Sea OMZ (Naqvi et al., 1993; Sen Gupta & Naqvi, 1984). In
the Arabian Sea, satellite-derived export production rates vary by a factor of 10
(Rixen et al., 2019b) and an even larger variability can be seen on the global scale.
Global export production rates derived from satellite-data, numerical models,
extrapolated from sediment trap data and based on estimates of the biological
oxygen demand vary approximately between 1.8 and 27.5 Pg C yr⁻. In general
estimates based on the biological oxygen demand, in line with inverse modeling
studies, call for higher export production rates (Burd et al., 2010; del Giorgio et al.,
2002; Schlitzer, 2000; Schlitzer, 2002), whereas satellite and sediment trap data, as
well as numerical model studies, suggest lower export production rates (Emerson,
2014). However, a longer residence time, in addition to a higher carbon export could
explain the AOU determined in the Arabian Sea (Rixen & Ittekkot, 2005).
**3.3 Implications**
Understanding of the processes controlling oxygen consumption within OMZs, such
as export production and the residence time of water, is still fraught with large
uncertainties. Nevertheless, it seems that there are oxygen-dependent, biologically
driven feedback mechanisms countering impacts of the monsoon on the intensity
of the OMZ. This could explain the absence of persistent anoxia and functional
anoxia in the Arabian Sea and Bay of Bengal OMZs. However, the recent expansion
of the OMZ in the Arabian Sea, and the first indication of denitrification in the Bay of
Bengal OMZ, indicate that there are other processes overriding effects of these
feedback mechanisms.

**4. The role of mesoscale eddy activity as a driver of OMZ ventilation**
Previous studies have also shown that, in addition to upwelling and vertical mixing,
eddies influence biological production as well as OMZ, through upward and
downwards movements of water (Boyd et al., 2019; Chelton et al., 2011; Gruber et
al., 2011; Omand et al., 2015; Oschlies et al., 1998; Resplandy et al., 2019).
Furthermore, eddies also affect the transport of biological tracers, through eddy
stirring that enhances patchiness in the ocean, and through eddy trapping that can
maintain properties of the trapped fluid over relatively long time periods (Chelton et
al., 2011; d'Ovidio et al., 2013; Mahadevan, 2016). In particular, stirring of oxygen by
eddies along isopycnal surfaces has been suggested to modulate the intensity and

distribution of low-oxygen waters in the ocean (Fig. 5, Gnanadesikan et al., 2013; Gnanadesikan et al., 2012). In the eastern tropical Atlantic and Pacific Ocean, recent work has highlighted the role of eddies in enhancing ocean mixing in regions of sluggish circulation, thus contributing to the ventilation of the OMZ located there (Bettencourt et al., 2015; Brandt et al., 2015; Gnanadesikan et al., 2013). In this context, long-term changes in oxygen have been linked to changes in the intensity of eddy activity. For instance, Brandt et al (2010) have shown that a reduction in filamentation and the strength of alternating zonal jets associated with mesoscale eddies between the periods 1972-1985 and 1999-2008 in the tropical north Atlantic has contributed to a reduction in the ventilation of the OMZ located there. Eddy trapping in turn maintains properties of the trapped fluid over relatively long time periods which favors the development of localized low-oxygen environments as seen e.g. in the open North Atlantic Ocean and off Peru in the Pacific Ocean (Bourbonnais et al., 2015; d'Ovidio et al., 2013; Fiedler et al., 2016; Karstensen et al., 2017; Schütte et al., 2016).

**4.1. Effects of eddies on the Arabian Sea OMZ**

In the Arabian Sea, numerical model studies have shown that eddies play an important role in the transport of nutrients and oxygen (Lachkar et al., 2016; McCreary Jr et al., 2013; Resplandy et al., 2012; Resplandy et al., 2011). For instance, Resplandy et al., (2011) emphasized the role of mesoscale eddies in spreading nutrients vertically and horizontally in the Arabian Sea (Fig. 11). Furthermore, mesoscale eddies and filaments were shown to dominate, on an annual timescale, the vertical supply of oxygen to the OMZ in the Arabian Sea (Resplandy et al., 2012). This study also showed that eddy-driven horizontal advection substantially contributes to the lateral transport of ventilated waters into the central and northern Arabian Sea. In a process study aiming to explore the dynamics of the Indian Ocean OMZs, McCreary et al (2013) also highlighted the important role of vertical eddy mixing in the ventilation of the western Arabian Sea in addition to the inflow of ICW. Their work further stresses the importance of this mechanism in the eastward shift of the upper OMZ relative to the region of highest productivity located along the western part of the Arabian Sea, as mentioned before.

Using a suite of regional model simulations with increasing horizontal resolution, Lachkar et al (2016) found that isopycnal eddy transport of oxygen to the Arabian Sea OMZ strongly limits the extent of its suboxic core. Within the model, an enhanced vertical mixing enhances the oxygen supply into the upper OMZ and the upward transport of nitrate into sunlit surface ocean. The latter enhanced the

621 biological production and the resulting export of organic matter, which in turn
increased the oxygen consumption at depth. The consequence is an expansion of
the volume of the OMZ. Finally, eddies have also been shown to control the transport
and the spreading of the Persian Gulf Water (PGW) into the Gulf of Oman (Queste et
al., 2018; Vic et al., 2015). These dense waters, relatively rich in $O_2$, subduct in the
northern Arabian Sea and strongly contribute to the ventilation of the upper OMZ
there (Lachkar et al., 2019; Schmidt et al., 2020). The projected future warming of
the Persian Gulf can cause a reduction in the sinking of PGW in the Gulf of Oman,
and hence a drop in oxygen at depths between 200 and 300 m in the northern
Arabian Sea.

**4.2. Eddies and the ventilation of the Bay of Bengal**
In the Bay of Bengal, previous studies have highlighted the role of eddy pumping of
nutrients in enhancing biological productivity during all seasons (Kumar et al., 2007;
Prasanna Kumar et al., 2004; Singh et al., 2015). Eddies have also been shown to
affect the ventilation of the Bay of Bengal and subsequently the intensity of its OMZ.
For instance, Sarma et al (2018a) showed that while cyclonic eddies inject nutrients
into the euphotic zone, thus enhancing productivity and oxygen consumption at
depth, anticyclonic eddies supply oxygen to the subsurface layer and hence weaken
the OMZ. Sarma and Baskhar (2018b) focused on anticyclonic eddies sampled by
bio-Argo floats between 2012 and 2016 in the Bay of Bengal. These anticyclonic
eddies seem to be formed in the eastern side of the basin and propagate westward.
They ventilate the layer between 150 and 300 m and weaken the OMZ. The frequent
episodic injection of oxygen, likely by mesoscale eddies, could be mechanism that
enhances the oxygen supply by lowering impacts of the strong stratification on
vertical mixing. However, in the Bay of Bengal, strong interannual variations in the
intensity of the eddy activity have been reported (Chen et al., 2012). These are
expected to cause strong variations in the subsurface ventilation that may eventually
lead to an intensification of the OMZ (Johnson et al., 2019).

**4.3 Implications**
So far, numerical model studies have shown that the variability of eddy activity in
space and time can modulate the intensity of OMZs between different regions and
across time, thus contributing to the observed variability of dissolved oxygen. The
fact that eddies affect both the supply of oxygen (through ventilation) and its
consumption (through biological productivity) in a non-trivial manner can explain the

fundamental difficulty to adequately parameterize the effects of eddies on dissolved
oxygen in coarse resolution models. An additional potential source of error in the
currently used parameterizations is their underlying assumption that the eddy-driven
isopycnal tracer mixing (i.e., diffusive part of eddy mixing) and the adiabatic
isopycnal flattening (i.e., advective effect of eddies) occur at similar rates (Griffies,
1998). Yet, recent studies (e.g. Gnanadesikan et al., 2013) suggest that the two can
be substantially different. In the Arabian Sea, Lachkar et al (2016) showed that the
eddy driven transport of oxygen is mostly driven by enhanced mixing along the
isopycnal surfaces, with very little change in the slope of the isopycnals. The
postulated decrease in the physical oxygen supply caused by the inflow of warmer
and hence more oxygen-depleted PGW agreed with the observed decreasing
oxygen concentrations and expansion of the SNM in the Arabian Sea. In the Bay of
Bengal, a response to global warming is more difficult to establish due to strong
interannual variations in the intensity of the eddy activity.

**5. Holocene records**

**5.1. Sediment $\delta^{15}$N records of OMZ strength**

On millennial and even longer timescales, sedimentary records have been used to
trace changes in the OMZ intensities. The $\delta^{15}$N values of particulate nitrogen in
sediments are often used as tracers of OMZ intensify because they reflect major
shifts in the pool of fixed nitrogen due denitrification, as discussed before (Altabet et
al., 1995; Altabet et al., 1999; Brandes et al., 1998; Ganeshram et al., 1995). Locally,
eolian and riverine nitrogen supply affect $\delta^{15}$N values (Kendall et al., 2007; Voss et
al., 2006), but in the Indian Ocean sedimentary $\delta^{15}$N reflects the balance between
denitrification vs. nitrogen fixation. Deep water nitrate has an average $\delta^{15}$N value of
~5‰ (Sigman et al., 2005) but, due to denitrification, the $\delta^{15}$N of nitrate in Arabian
Sea increase to values > 17 ‰ (Fig. 9). Convective mixing, eddy pumping, and
especially upwelling, move nitrate-deficient water masses from the OMZ to the
surface, so that nitrate with high $\delta^{15}$N values is transported into the euphotic zone.
After assimilation into biomass by phytoplankton, $^{15}$N-enriched particulate matter
sinks through the water column to the seafloor where the signal of denitrification, and
hence OMZ intensity, is preserved in sediments (Altabet et al., 1995; Gaye-Haake et
al., 2005; Naqvi et al., 1998; Suthhof et al., 2001). Early diagenesis may raise
sedimentary $\delta^{15}$N values by 2 - 5 ‰, and the diagenetic effect increases with water
depth (Altabet, 2006; Tesdal et al., 2013). Nevertheless, the relative changes of $\delta^{15}$N
in deep-sea sediments record variations in the OMZ intensity while records from the

continental slopes are subjected to negligible diagenetic enrichments so that they
retain the signal of the nitrogen source (Altabet et al., 1999; Gaye et al., 2018).

**5.2. OMZ Fluctuations during the Holocene**

A sediment core from the northern Bay of Bengal (Contreras-Rosales et al., 2016)
indicates that the highest $\delta^{15}$N values (and thus the lowest OMZ oxygen
concentrations) recorded in the core prevailed during the Holocene and the last
glacial maximum, with a $\delta^{15}$N range between 4.4 and 5.0 ‰ (i.e., in the range of the
average value of deep ocean waters, see above). Therefore, denitrification in the
past 21,000 years can be ruled out in the Bay of Bengal from a paleoceanographic
perspective (Contreras-Rosales et al., 2016). The $\delta^{15}$N values at the core top (4.6 ‰)
were similar to values in sediment trap materials (3.7 - 4.5 ‰), and were explained
by a mixture of nutrients or suspended matter from the Ganges-Brahmaputra-
Meghan river system with nitrate from subsurface water (Contreras-Rosales et al.,
2016; Gaye-Haake et al., 2005; Unger et al., 2006). Enhanced $\delta^{15}$N values in the
early Holocene to 6000 years BP (BP = before present, where present means 1950)
coincide with a stronger monsoon and were attributed to enhanced supply of nitrate
from the subsurface, which has elevated $\delta^{15}$N compared to the depleted values of the
riverine end-member (Sarkar et al., 2009). However, to our knowledge there is only
one published sediment record from the Bay of Bengal spanning the entire Holocene
(Contreras-Rosales et al., 2016) so that we know nothing about the spatial variability
within the basin.

In contrast to the Bay of Bengal, denitrification in the Arabian Sea has prevailed
during the warm interstadials of the Pleistocene and during the entire Holocene, as
can be discerned from sedimentary $\delta^{15}$N values > 6 ‰, with maxima of > 11 ‰
(Agnihotri et al., 2003; Higginson et al., 2004; Kessarkar et al., 2018; Möbius et al.,
2011; Pichevin et al., 2007). Productivity increased with the onset of the Holocene as
the summer monsoon strengthened and monsoonal upwelling off Somalia and Oman
commenced and became a permanent feature of the Holocene Arabian Sea (Böning
et al., 2009; Gaye et al., 2018). A rise of $\delta^{15}$N by at least 2 ‰ shows that an onset of
upwelling immediately strengthened the OMZ and led to denitrification across the
entire basin in the beginning of the Holocene (Böll et al., 2015; Gaye et al., 2018).

Furthermore, southward retreat of Antarctic Sea Ice is assumed to have reduced
ventilation of the Arabian Sea OMZ through its influences on the formation of the
oxygen-enriched ICW and associated meridional overturning circulation of the upper
Indian Ocean (Fig. 6 Böning & Bard, 2009; Naidu et al., 2010). A decline in $\delta^{15}$N by
about 1 ‰ is found in the early Holocene until 6000 years BP in high-resolution

sediment cores from the western, northern and eastern Arabian Sea, and indicates
that the OMZ weakened and became less persistent during this period (Fig. 6a).
More vigorous upwelling, discernible from benthic foraminifera, may have led to a
better ventilation of the basin by ICW from the south during this period, by reducing
the residence time of OMZ waters (Das et al., 2017).

After 6000 years BP, increasing $\delta^{15}N$ values indicate a strengthening of the OMZ
across the entire basin (Fig 12 b). It is assumed that a weaker ventilation is
responsible for decreasing oxygen concentrations and it could be due to reduced
inflow of ICW, as it was blocked by the enhanced inflow of PGW and RSW since the
sea level high stand at 6000 years BP (Pichevin et al., 2007). Furthermore, a
southward shift of the West Indian Coastal Current and the associate poleward
undercurrent lowered the inflow of ICW and the thereby the ventilation of the OMZ in
the eastern Arabian Sea (Mahesh et al., 2014). The reduced inflow of ICW generally
might have not only reduced ventilation but also prolonged the residence time of
water within the OMZ (Böning & Bard, 2009; Pichevin et al., 2007).

**5.3 Holocene model simulations**
In order to give an additional, model-based estimate of the OMZ evolution in the
Indian Ocean, transient model simulations over the Holocene were performed with
the global atmosphere-ocean Kiel Climate Model (KCM, Park et al., 2009) and the
marine biogeochemistry model PISCES (Aumont et al., 2003).

In a first step, KCM was forced with transient orbital parameters and greenhouse gas
concentrations from 9500 years BP to present. In a second step, the PISCES model
was forced with the ocean physical fields from the above KCM experiment in so-
called off-line mode. This model set-up comprised a ventilation age tracer of the
water masses (see Segschneider et al. (2018) for a more detailed description of the
model components and experiment set-up). While the oceanic $2^o \times 2^o$ grid in this set-
up was refined to a meridional resolution of $0.5^o$ near the equator to allow a better
representation of equatorial waves, the long integrations (9500 model years) required
a coarse model resolution that is far from eddy resolving. The ballast scheme for the
export of POC also neglected the lithogenic ballast effect, which is important in the
Arabian Sea and Bay of Bengal a discussed before.

From these model experiments, temperature and oxygen fields have been analyzed
and compared to sedimentary records mainly in the Arabian Sea. Here the model
results were subdivided into areas corresponding to the binned sediment core
regions specified in Gaye et al (2018) (Fig. 12 a; North: $62^oE-68^oE$, $20^oN-25^oN$; East:

68°E-75°E, 13°N-20°N; West: 54°E-60°E, 15°N-20°N; South: 48°E-55°E, 7°N-12°N).
The simulated oxygen concentrations in the Arabian Sea are generally somewhat too high at the surface due to a cold bias of the KCM, but the observed near-surface gradients of oxygen concentrations are very well matched. However, in the deeper layers the model overestimates oxygen concentrations (not shown, see supplementary figure A.1c in (Segschneider et al. (2018)). As a result, oxygen concentrations in the model Arabian Sea are nowhere low enough for denitrification to occur (denitrification sets in at 6 µM in the PISCES model, with a transition phase to full denitrification at lower oxygen concentrations). Moreover, no nitrogen isotopes are simulated in the current model version. Comparison to the $\delta^{15}$N data from the sediment cores is, therefore, restricted to a qualitative assessment.

The simulated oxygen concentrations (averaged between 200 m and 800 m depth) show the lowest concentrations in the northern Arabian Sea (initially around 80 µM in the early Holocene, yellow curve in Fig. 12c). The concentrations are 10 µM higher in the western Arabian Sea (blue line), and a further 5 µM higher in the eastern Arabian Sea (red line), while they are much higher in the southern Arabian Sea (starting at 155 µM, grey line). Oxygen concentrations are fairly constant over the first 2.5 thousand years, and then gradually decrease until the late Holocene. This decrease is strongest in the northern Arabian Sea (-20 µM) and quite similar in the western and eastern Arabian Sea (-10 µM). This is in qualitative agreement with the Holocene trends of $\delta^{15}$N data (Fig. 12b) that show highest $\delta^{15}$N values (indicating strong denitrification and thus low oxygen) for the shallow northern core, and lower $\delta^{15}$N for the western and eastern cores.

Simulated export production and water mass age in the Arabian Sea have been discussed for an earlier model experiment with the same model set-up (but accelerated forcing) by Gaye et al. (2018), and in more detail for the global OMZs including the Indian Ocean for the model experiment analyzed here by Segschneider et al. 2018. While simulated export production in the Arabian Sea is fairly constant throughout the Holocene (Fig. 7 in Gaye et al. 2018), ventilation age is increasing throughout the Holocene concurrent with decreasing oxygen concentrations (Fig. 15 in Segschneider et al., 2018). This implies that changes in the ocean circulation and the associated inflow of oxygen-enriched ICW largely influenced the OMZ during the Holocene after the onset of upwelling at the beginning of Holocene.

## 5.4 Implications

The $\delta^{15}N$ sedimentary records reveal the difference in the late Pleistocene and Holocene history of denitrification in the Arabian Sea and Bay of Bengal. Oxygen concentrations in the Bay of Bengal never declined below the threshold of denitrification, whereas denitrification prevailed in the Arabian Sea during the warm interstadials and the entire Holocene. A data-model comparison shows that the age of the OMZ water mass increased after 6000 years BP in both basins (not shown for BoB), coinciding with a strengthening of the OMZ and denitrification in the Arabian Sea. Based on the model results of constant export production and increasing water mass age, it is concluded that a reduced ventilation is responsible for decreasing oxygen concentration. The similar temporal evolution of observed OMZ intensity and modeled oxygen concentration in the Arabian Sea under orbital and greenhouse gas forcing thus indicates that the mid- to late Holocene OMZ intensification may be related to oceanic circulation rather than to local processes in the Northern Indian Ocean. The progressive oxygen loss over the Holocene may thus be the result of orbital and greenhouse gas forcing in a qualitatively similar way to the much stronger variations simulated for LGM to mid-Holocene changes (Bopp et al. 2017).

## 6. Model predictions

### 6.1 Global models

For future climate predictions we rely on earth system models (ESM). Although these models reproduce large-scale features and global trends, they suffer from considerable mismatches between measured and model oxygen concentrations in the ocean (Bopp et al., 2013; Cabré et al., 2015; Oschlies et al., 2018; Oschlies et al., 2008). In comparison to observational data, they underestimate oxygen losses significantly (e.g. Oschlies et al., 2018 and references therein), and simulated volumes of OMZs differ considerably. Unresolved physical oxygen supply mechanisms, poorly constrained biological oxygen consumption rates, and their hardly known responses to global change, cause these uncertainties (e.g., Oschlies et al., 2018; Segschneider et al., 2013). Furthermore, feedbacks caused by the strong coupling of the marine oxygen and nitrogen cycles complicate long-term predictions (Fu et al., 2018; Oschlies et al., 2019).

Especially in the Indian Ocean, global coupled biogeochemical ESMs struggle to represent the OMZs (Fig. 13, Oschlies et al., 2008). In most ESMs the east – west contrast between the Arabian Sea and Bay of Bengal is opposite to what observations show, with most global models producing lower oxygen concentrations

in the Bay of Bengal than in the Arabian Sea. Furthermore, half of the models cannot
predict hypoxic conditions in the Arabian Sea at all. A comparison of the thickness of
the hypoxic layer in the northern Indian Ocean shows a disagreement among all
models (Fig. 13). The maximum simulated volume ($8.2 \times 10^{15}$ m$^3$, CESM1-BGC) is
more than twice the hypoxic volume found from observations ($3.1 \times 10^{15}$ m$^3$,
WOA13). Moreover, this volume extends too far horizontally and does not cover the
thickness of the observed OMZ in the Arabian Sea (Fig.13).

To some degree, this problem may be attributed to the fact that ESMs are not tuned
for the northern Indian Ocean. In addition, global models generally have coarser
resolution to reduce computational costs, and thus are far from eddy resolving, as for
the KCM (results discussed in the previous section). Eddy transport is parameterized
in the ESMs, but these still fail to represent the OMZs in the northern Indian Ocean.
Even though the next generation of ESMs already targeted this problem, by providing
high-resolution options including mesoscale processes in models used in the 6[th]
coupled model intercomparison experiment (CMIP6), there are only moderate
improvements in subsurface oxygen representation (Kwiatkowski et al., 2020). The
CMIP6 models still tend to overestimate oxygen concentrations in the Arabian Sea
(Séférian et al., 2020).

**6.2 Future prediction**
The poor representation of the OMZs in the northern Indian ocean in ESMs reduces
the reliability of future predictions of potential changes in the OMZs related to natural
and anthropogenic forcing, and thus their ecological impacts and possible feedbacks
to climate change. Global models suggest a general decline of oxygen for the entire
ocean, but there is no clear trend visible in the Indian Ocean (Oschlies et al., 2017).
However, an older set of ESMs analyzed in Cocco et al. (2013) suggest a future
decrease in oxygen in the subtropical Indian Ocean in the upper mixed layer, and a
small increase in the western tropical Indian Ocean. This increasing oxygen
concentration is also seen in response to climate change in the RCP8.5 and RCP2.6
scenarios of the 5[th] coupled model intercomparison project (CMIP5, Bopp et al.,
2013). Specifically, Bopp et al. (2013) showed that a decrease in productivity is
consistently simulated across all CMIP5 models and scenarios in the tropical Indian
Ocean and that, by 2100, all models project an increase in the volume of waters
below an oxygen concentration of 80 µM, relative to 1990–1999. This response is
more consistent than that of the previous generation of ESMs, i.e., changes varying
from −26 to +16% over 1870 to 2099 under the SRES-A2 scenario (Cocco et al.,
2013).

However, for lower oxygen levels, there is less agreement among the CMIP5 models
and also compared to observations regarding the volume of the OMZ (Bopp et al.,
2013). Specifically, for the volume of waters below 50 µM, four models project an
expansion of 2 to 16% (both GFDL-ESMs, HadGEM2-ES and CESM1-BGC),
whereas two other models project a slight contraction of 2% (NorESM1-ME and MPI-
ESMMR). For the volume of waters below an oxygen concentration of 5 µM, only one
model (IPSL-CM5A-MR) is close to the volume estimated from observations, and
projects a large expansion of this volume (+30% in the 2090s). These results for low
oxygen waters (oxygen concentrations of 5 - 50 µM) agree with those of Cocco et al.
(2013), with large model–data and model–model discrepancies, and simulated
responses varying in sign for the evolution of these volumes under climate change
(Bopp et al., 2013).

Globally, the models agree on a negative oxygen trend driven by declining solubility
through global warming (Resplandy, 2018; Schmidtko et al., 2017) and reduced
ventilation by changes in oxygen transport by circulation (Bopp et al., 2017), and take
into consideration negative feedback caused by reduced tropical export production
due to increased stratification of the upper water columns (Fu et al., 2018; Palter et
al., 2018). Uncertainties and disagreements among the models arise from subtle
differences in timing and magnitude of these opposing trends (Bopp et al., 2017).
Waters with low oxygen saturation are particularly sensitive to impacts of climate
warming (Fu et al., 2018) as well as vertical diffusivity that is parameterized by the
mixing coefficient in the models (Duteil et al., 2012) and also mesoscale eddy
transport and the lateral mixing coefficient (Bahl et al., 2019; Lachkar et al., 2016).
Globally, reduced mixing across the mixed layer explains 75% of the reduced
subduction, but regionally more important are changes in wind patterns that cause
modulations in Ekman pumping and subduction (Couespel et al., 2019). Thus, future
trends in the northern Indian Ocean OMZs derived from the ESMs are highly
uncertain, with predicted potential increases or decreases in the volume of low
oxygen waters, depending on the model and the oxygen levels under consideration
(Bopp et al., 2013; Cocco et al., 2013).

**6.3 Implications**

The OMZ in the Indian Ocean is the one we know least about but it may also be the
OMZ with the most complex dynamics in terms of forcing and variability. As
discussed before, regional eddy resolving modelling studies have been able to
reproduce the OMZs and thus they have helped us to understand the interplay
between physical and biogeochemical drivers (Lachkar et al., 2019; McCreary Jr et

al., 2013; Resplandy et al., 2012; Resplandy et al., 2011). Global models still struggle
to reproduce the Indian Ocean OMZ. One explanation for this is the coarse resolution
of these models, i.e., they cannot resolve the mesoscale and submesoscale
processes that ventilate the subsurface waters and they underestimate coastal
upwelling during the monsoon seasons and, therefore, also primary production and
biological oxygen demand. As a result, the oxygen trend in the tropical Indian Ocean
remains unclear. However, in addition to poor representation of mesoscale and
submesoscale features in global models, large uncertainties stem also from largely
unknown biogeochemical and ecosystem responses to global physical changes.

**7. Ecosystem responses**

**7.1 Pelagic ecosystems**

Dissolved oxygen concentrations in seawater are crucial for the successful
development of many pelagic organisms, particularly marine animals (both
vertebrates and invertebrates) whose metabolism, life cycle performance, growth
capacity, reproductive success and longevity are intimately linked to oxygen
availability. Even though an oxygen concentration of < 60 - 63 µM is widely accepted
as a threshold below which higher organisms start to suffer from the lack of oxygen
the tolerance to decreasing oxygen, critical concentrations vary enormously among
species and even within the same species (Ekau et al., 2010; Vaquer-Sunyer &
Duarte, 2008). Differences can be very large even at various growth stages of
animals (Miller et al., 2002). Many fish larvae present in the pelagic realm are
incapable of further growth and development at oxygen values < 134 µM, while
organisms such as euphausiids can survive to 4.5 µM. Thus, a change in the
average or the range of dissolved oxygen concentrations in the water column could
have significant impacts on the survival of certain species and consequently the
species composition in the ecosystem. As compared to marine vertebrates and
invertebrates, the impacts of hypoxia on phytoplankton physiology and growth are
less known.

**7.1.1 *Noctiluca* blooms**

There is a large body of evidence on the effects of hypoxia on macro-organisms.
This includes reduced diel migration depths, vertical habitat compression and
shoaling distributions of fishery species and their prey (Breitburg et al., 2018).
However, few reports exist on the effects of hypoxia on phytoplankton, the primary
producers of the marine ecosystem. This is because it is generally understood that
biological consequences of reduced oxygen concentrations are likely to be most

notable for the 200 – 300 m layer, as these waters impinge on the euphotic zone (Stramma et al., 2010b). There have been several reports of coastal upwelling bringing up nutrient-enriched and hypoxic waters onto continental shelves stimulating production and increasing local biological oxygen demand (Stramma et al., 2010b). Death and decay of phytoplankton blooms (often harmful algae), of exceptional biomass, can completely strip nearshore waters of oxygen. But in the Arabian Sea, which is seeing one of the most extreme and dramatic changes tied to anthropogenic planetary warming, there has been a shift in the base of the ecosystem from predominantly autotrophy before 2000 (Garrison et al., 1998; Garrison et al., 2000; Smith et al., 1998a), to greater dependence on mixotrophy more recently (Gomes et al., 2014). A large-scale, ongoing study conducted by the National Institute of Oceanography, India, from 2003 onward, in support of India's ocean color program, first documented the appearance of extensive blooms of green mixotrophic dinoflagellate, *Noctiluca scintillans* (*Noctiluca*). *Noctiluca* is large (~1 mm in diameter) and capable of sustaining itself via photosynthesis from its green free-swimming endosymbionts, *Pedinomonas noctilucae* (Wang et al., 2016) and/or by ingestion of exogenous prey (Goes et al., 2016; Gomes et al., 2014; Gomes et al., 2009; Prakash et al., 2008; Prakash et al., 2017). Within a decade and a half, *Noctiluca,* has taken over the once diatom-dominated food chain of the Arabian Sea and the Sea of Oman, forming large green mats that can be observed from space with regular predictability (Fig. 16).

In their quest to find the ecological drivers of *Noctiluca*, Gomes et al. (2014) conducted on-deck dissolved oxygen amendment experiments, in the central and western Arabian Sea during the winter monsoons of 2009, 2010, and 2011 to provide the first conclusive evidence that the growth of green *Noctiluca* blooms was being facilitated by hypoxia. Results from this study showed that green *Noctiluca* is predisposed to hypoxic waters and is able to fix $CO_2$ more efficiently under hypoxic conditions than at higher concentrations of dissolved oxygen. In contrast, diatoms and other phytoplankton showed a > 50% decrease in $CO_2$ fixation rates under lower oxygen concentrations. While it appears that green *Noctiluca* thrives in hypoxic waters, it is unclear whether *Noctiluca* is capable of modulating its intracellular environment in order to maximize photosynthetic rates by its endosymbionts. However, the regular occurrence of *Noctiluca* seems to be in line with the decreasing oxygen concentrations in the Arabian Sea as discussed before.

Continuous long-term sampling at a coastal station (Bay of Bhandar Khayran), from 2001, and at another location offshore from 2005, both in the Sea of Oman, supplemented by field observations, provided a mesocosm-like situation to study the

ecophysiological characteristics that underpin *Noctiluca*'s recent success. The glider-based study of Piontkovski et al. (2017) and an earlier observational study (Goes & Gomes, 2016) showed that prior to appearing as surface blooms in late winter, *Noctiluca* are seen at deeper depths close to the oxycline, often as large, actively photosynthesizing subsurface blooms advantaged by the intrusion of hypoxic waters into the euphotic column and the higher concentrations of nutrients required for endosymbiont photosynthesis. In this region, *Noctiluca* blooms (Al-Azri et al., 2015; Al-Hashmi et al., 2015) are found in association with a large cyclonic eddy that facilitates the up-shoaling of low-oxygen, high-nutrient waters to the surface (Gomes et al., 2009; Harrison et al., 2017). As the water column warms and stabilizes, a requirement for dinoflagellates to proliferate, *Noctiluca* blooms, as mixotrophs with phagotrophic abilities, proliferate, now advantaged by the plentiful food of phytoplankton, associated bacteria and detritus. Altimetry data show furthermore that this semi-permanent cyclonic and mesoscale eddy is responsible for sustaining this bloom for a prolonged period along the coasts of Oman and Iran even until February (Gomes et al., 2009). Both cyclonic and anticyclonic eddies disperse *Noctiluca* eastwards into the central and eastern Arabian Sea, ultimately engulfing the entire northern Arabian Sea (Gomes et al., 2009).

A more recent study (Lotliker et al., 2018) refutes the connection between *Noctiluca*'s blooms and the spread of hypoxia. However, their conclusions were not backed by any experimental data and their $O_2$ data were from Bio-ARGO floats that were located south of where *Noctiluca* blooms occur. Sensor calibration was also a contentious issue, and Lotliker et al. (2018) provided only 6 calibration points for a dataset that spanned from Feb 2013 to April 2016. However, we are still uncertain if these large blooms will further intensify the Arabian Sea OMZ. Nonetheless, we are aware that *Noctiluca* is not a preferred food for most zooplankton, but is voraciously grazed upon by gelatinous tunicates such as salps and tunicates. During our field campaigns, we have seen large swarms of salps, known to be as efficient filter feeders, devouring *Noctiluca* (Gomes et al., 2014) and depositing large pellets. Salp pellets are known to be fast sinking, (up to 2700 m day−1), carbon-rich (up to 37% DW), contributing disproportionately to carbon flux compared with other zooplankton (Henschke et al., 2016; Martin et al., 2017).

### 7.1.2 Zooplankton migration

Vertical oxygen gradients of the OMZ set the limits to the horizontal and vertical distribution of zooplankton, affecting their distribution, diel vertical migration, and

ecological functions strongly (Saltzman et al., 1997; Wishner et al., 2008). Recent modeling studies (Aumont et al., 2018) estimate that about one third of the epipelagic biomass performs diurnal vertical migrations.

In general, most zooplankton taxa show minimum abundances in the core of the OMZ, and higher abundances in well-oxygenated waters above or beneath the OMZ (Böttger-Schnack, 1996; Saltzman & Wishner, 1997; Wishner et al., 1995). Certain zooplankton, however, have developed vertical migration strategies that enable them to pass through or even live within the OMZ (Gonzalez et al., 2002; Herring et al., 1998; Longhurst, 1967). The ability to do so has been linked to the presence of lactic dehydrogenase (LDH), an enzyme associated with anaerobic metabolism (Escribano, 2006; Gonzalez & Quiñones, 2002).

In the Arabian Sea, almost 85% of the epipelagic mesozooplankton biomass is found within the upper aerobic part of the seasonal thermocline or approximately in the upper 100 m. Below this region, in the anaerobic part of the seasonal thermocline, zooplankton concentrations decline sharply (Banse, 1994; Böttger-Schnack, 1996; Smith et al., 2005; Wishner et al., 1998). The most comprehensive study of this region, the US JGOFS (Smith et al., 1998b; Smith & Madhupratap, 2005), concluded the following *vis a vis* zooplankton distributions and the OMZ: 1) exclusion from the suboxic core of the OMZ of most zooplankton biomass 2) paradoxically, the occurrence of extremely high abundances of a few species of diel vertical migrators at depth during the daytime, well within the suboxic zone 3) organism-specific (and probably species-specific) distribution boundaries at the upper and lower edges of the OMZ, probably associated with particular oxyclines 4) very high biomass of diel vertical migrators that moved between the surface waters at night and the suboxic waters during the day with many of these animals spending the day at depths where the oxygen was less than 4.5 µM and 5) aggregation of mesozooplankton communities to surface layers, in locations where the OMZ was forced upwards due to physical processes and where they are susceptible to predators.

A comparison between a eutrophic, more oxygenated onshore station and an offshore station with a strong OMZ elucidated the influence of depth and oxygen concentration, as well as other factors on the copepod distribution in the Arabian Sea (Wishner et al., 2008). The extent and intensity of the oxycline at the lower boundary of the OMZ, and its spatial and temporal variability over the year of sampling, was an important factor affecting distributional patterns. Calanoid copepod species showed vertical zonation through the lower OMZ oxycline, but no apparent diel vertical migration for either calanoid or non-Calanoid copepods was observed at these midwater depths. Subzones of the OMZ, termed the OMZ Core, the Lower Oxycline,

and the Sub-Oxycline, had different copepod communities and ecological interactions. The Calanoid copepod community was most diverse in the most oxygenated environments (oxygen > 6.25 $\mu$M), but the rank order of abundance of species was similar in the Lower Oxycline and Sub-Oxycline. Some species were absent or much scarcer in the OMZ Core. It thus appears that the vertical zonation of copepod species through the lower OMZ oxycline is probably a complex interplay between physiological limitation by low oxygen, potential predator control, and potential food resources.

Only one species in the Arabian Sea, *Pleuromamma indica*, has displayed the ability to survive and thrive in hypoxic waters. This species is not only observed in large numbers in hypoxic waters (Goswami et al., 1992; Haq et al., 1973; Saraswathy et al., 1986; Vinogradov et al., 1962), but is also capable of migrating daily through the well-oxygenated surface layer (Saraswathy & Iyer, 1986). There are also indications that the increased abundance of *P. indica* in recent years is tied to the geographically more widespread oxygen depletion.

While a considerable body of information is available on the OMZ as a determinant of zooplankton distribution, less is known on the extent of effects of diel migration on oxygen depletion in OMZs of the world. Using measurements from shipboard acoustic Doppler current profilers (ADCPs) and a global biogeochemical model, Bianchi et al. (2013) found that by clustering in the upper margins of OMZs, vertical migrators accentuate organic matter breakdown in these waters, exacerbating the oxygen deficit. Aumont et al. (2018) used a fully coupled model to simulate the net impact of diurnal vertical migration on dissolved oxygen of the entire pelagic ecosystem on a global scale. Respiration and egestion by migratory organisms induce a modest decrease in oxygen between 150 and 500 m, which reaches about 5 µM averaged globally at 500 m, although less so in the Arabian Sea and the Bay of Bengal. Three distinct vertical layers could be distinguished over the global ocean: 1) Vertical migration generates a positive dissolved oxygen anomaly in the subsurface above 200 m, that can exceed 10 µM which is explained by less intense respiration in the seasonal thermocline 2) Further below, down to about 1,000 diel vertical migration produces a depletion in oxygen from respiration by migrators with greatest depletion at middle and high latitudes 3) Finally, in the bathypelagic domain (below 1,000 m), oxygen levels are increased by almost 2 µM as a result of a slightly lower oxygen consumption.

### 7.1.3. Implications

We are still uncertain if the recent emergence and persistence of *Noctiluca* blooms will further intensify the Arabian Sea's OMZ. Satellite-derived Chlorophyll-*a* trends (1980-2019) reveal an almost three-fold increase in phytoplankton biomass, with increases particularly in the northwestern and central Arabian Sea (Goes et al., 2020). With respect to repercussions for the food chain, we are aware that *Noctiluca* is not a preferred food for most zooplankton, but is voraciously grazed upon by gelatinous tunicates such as salps and tunicates. High gelatinous zooplankton biomass is often observed in regions of persistent low oxygen concentrations (Lucas et al., 2014) suggesting that the recent appearance of extensive blooms of *Noctiluca* reflect the intensification of the Arabian Sea OMZ. The earlier, comprehensive JGOFS studies of the 1990s, which investigated the vertical migration and distributions patterns Arabian Sea's OMZ, have not been repeated even on a moderate scale as acute piracy and political instability have hindered campaigns to the region. Thus, while modeling (Lachkar et al., 2019; Lachkar et al., 2018) and data compilation studies (Banse et al., 2014; Rixen et al., 2014) suggest the expansion of the OMZ in the Arabian Sea, as discussed earlier, little is known of its effect on zooplankton distribution and vertical migration.

### 7.2 Benthic ecosystems

### 7.2.1 Benthic communities

Hypoxia has major consequences at the sea floor, for benthic communities and for the biogeochemical processes they drive. Benthic communities and processes in the Bay of Bengal have thus far received less study than those of the Arabian Sea. It is however clear that oxygen exerts an important control on benthic communities across the margins of both basins (e.g, Ingole et al., 2010; Raman et al., 2015). There are grain-size related contrasts in communities across the shelves, but also clear oxygen-related patterns across the upper slope depth ranges where mid-water oxygen minima impinge on the sea floor (Fig. 9). In the Arabian Sea, the degree to which this oxygen effect is expressed varies between margins due to differing degrees of bottom-water ventilation. On the Pakistan margin, where ventilation and bottom-water oxygen levels are lowest, hypoxia-resistant foraminifera are the only fauna to persist at the core of the OMZ, and macro- and megafauna are totally absent (Gooday et al., 2009). By contrast, on the Indian margin, and even off Oman, where upwelling-driven productivity and delivery of organic matter to sediments are particularly high, macrofauna generally persist across the entire margin, albeit in

reduced numbers and diversity at the OMZ core (e.g., Ingole et al., 2010; Levin et al.,
2000).

Further, across the OMZ boundaries, clear "edge effects" have been observed; sharp
changes in community composition and faunal abundance linked to different oxygen
thresholds (e.g., Levin et al., 2009b). These have also been observed on other
hypoxia-impacted margins in the eastern Pacific and off SW Africa (e.g., Levin et al.,
1991), as well as in hypoxic basins such as the Baltic Sea, and at sites impacted by
excess organic matter input (e.g., Rosenberg, 2001). While there are some common
patterns, specific oxygen thresholds are difficult to constrain because of inter-margin
and inter-basin differences in faunal assemblages, which are also affected by local
differences in factors such as food availability and predator avoidance, as well as
inter-study differences in the availability and quality of bottom-water oxygen data.

**7.2.2 Benthic ecosystem function**

The strong but variable cross-OMZ gradients in bottom-water oxygen and benthic
communities translate to contrasts in benthic ecosystem function, which also varies
between margins. For example, the numbers, size and depth of faunal burrows, and
the extent of bioturbation and bio-irrigation, change across the OMZ boundaries
(e.g., Cowie et al., 2009a; Smith et al., 2000). In the extreme case, this leads to total
absence of bioturbation and bio-irrigation at the core of the OMZ off Pakistan, and
the resulting presence of annually laminated (varved) sediments, which are not
observed on the better-ventilated margins of the Arabian Sea or in the Bay of Bengal.
In the Arabian Sea, there are also clear oxygen-dependent differences in benthic
community organic matter processing, as have been revealed by tracer incubation
experiments. For example, a threshold oxygen concentration occurs, above which
macrofauna dominate short-term organic matter (OM) processing, and below which
meiofauna and bacteria dominate. This was illustrated on the Pakistan margin both
at sites that spanned the lower OMZ boundary and at a shelf-edge site that
underwent strong seasonal change in bottom-water oxygen levels, from fully
oxygenated (intermonsoon) to hypoxic (summer monsoon) (e.g., Andersson et al.,
2008; Pozzato et al., 2013; Woulds et al., 2009; Woulds et al., 2007).

Further, the "edge effect" seen in benthic community composition also has been
observed in faunal OM processing. At sites in the lower OMZ transition zone, the
polychaete *Linopherus sp.* showed clear morphological adaptation to low oxygen
levels, and overwhelmingly dominated both the benthic community and also the
uptake and processing of organic matter (Jeffreys et al., 2012). These results, and
those of other experiments (e.g., Hunter et al., 2012; White et al., 2019), illustrate

that faunal assemblage composition may represent an important factor determining the pattern of seafloor processing, but also the composition, bioavailability and fate of residual organic matter. It is certainly clear that faunal digestive processes are recorded in the composition of organic matter deposited across the margins (e.g., Jeffreys et al., 2009; Smallwood et al., 1999). In summary, oxygen-dependent cross-margin variability in benthic communities and ecosystem function (feeding, bioturbation and bio-irrigation etc) may be important contributors to the role that oxygen exposure plays in controlling organic carbon distribution and burial across Arabian Sea margins, although other factors, most notably hydrodynamic processes, are also important (e.g., Cowie, 2005; Cowie et al., 2009b; Koho et al., 2013; Kurian et al., 2018).

### 7.2.3 Sediment redox conditions and microbial processes

Alongside the contrasts in faunal communities, bioturbation and irrigation, there are cross-OMZ differences in sediment redox conditions and microbial processes. Again, these are expressed to varying degree on the different margins of the Arabian Sea (Cowie, 2005), and will be less apparent in the Bay of Bengal due to the less intense oxygen depletion at the OMZ core. In the Arabian Sea, sulfate reduction has generally been shown to be surprisingly limited in near-surface sediments (top ~50 cm) (e.g., Cowie, 2005; Law et al., 2009), and redox conditions overall to be only moderately reducing (e.g., Crusius et al., 1996) relative to rates and conditions observed on upwelling/OMZ margins in other basins. Nonetheless, Pakistan margin sediments, and possibly those on other Arabian Sea margins, are home to significant rates of denitrification and anammox (e.g., Schwartz et al., 2009; Sokoll et al., 2012) and authigenic phosphorous (P) burial (e.g., Filippelli et al., 2017; Kraal et al., 2012). These phenomena represent important sink terms in the N and P biogeochemical cycles, and, along with sediment-water nutrient fluxes that vary in direction, magnitude and N:P stoichiometry across the OMZ, serve as potential controls on pelagic nutrient inventories.

Finally, there is evidence that Pakistan margin sediments (and possibly OMZ sediments on other margins), sequester important amounts of "dark" (non-photosynthetic) carbon arising from anammox and possibly other chemoautrophic processes occurring in overlying waters or within the sediments (e.g., Cowie et al., 1999; Cowie et al., 2009b; Lengger et al., in press). It is a term that is currently underestimated or ignored in carbon budgets and biogeochemical models. On the Pakistan margin, there are also chemosynthetic bacterial mats associated with methane seeps (Himmler et al., 2018)

**7.2.4 Implications**

As mentioned above, the coastal hypoxia on the western Indian shelf can reach the
extreme of fully sulfidic conditions in nearshore bottom waters (e.g. Naqvi et al.,
2000). Apart from mortality of benthic (as well as pelagic) fauna under extreme
conditions, details of the effects of seasonal hypoxia on benthic communities in the
shelf and coastal waters of Arabian Sea and Bay of Bengal are not well documented.
Thus, while seasonal contrasts in benthic community organic matter processing were
reported on the Pakistan shelf (see above), it is not otherwise clear if or how benthic
communities have adapted to the recurring, possibly intensifying, hypoxia. What is
clear is that wholesale seasonal changes occur in benthic microbial processes and in
the magnitudes and directions of sediment-water nutrient fluxes (e.g., Pratihary et al.,
2014).

Potential benthic ecosystem and biogeochemical consequences of projected
intensification and expansion of hypoxia have been the subject of multiple reviews
(e.g., Levin et al., 2009a; Middelburg et al., 2009; Stramma et al., 2008).
Intensification of hypoxia within the Arabian Sea and Bay of Bengal OMZs would
predictably drive distributions in benthic communities, sediment characteristics and
biogeochemical processes towards those currently observed off Pakistan. This would
result in potentially expanded depth ranges devoid of macro- and megafauna (and
thus bioturbation and irrigation), but also shifts in the locations and composition of
"edge" populations associated with oxygen gradients at OMZ boundaries. Other
hypoxia-related phenomena might also impact on benthic ecosystems. These include
the increasing prevalence of *Noctiluca* and jellyfish and their potential impacts on
food webs and organic matter export to depth. Mass deposition of jelly fish on the
seafloor off Oman (Billett et al., 2006) have major impacts on seafloor communities
and processes (Sweetman et al., 2016).

It is not yet clear what the net effect of such changes would be on carbon burial, but
changes in faunal populations, and transition from hypoxic to fully anoxic conditions,
could have major impacts on benthic N and P cycling and sediment-water nutrient
fluxes (and N:P ratios), as has been observed with expanding hypoxia in the Baltic
(Jilbert et al., 2011; Karlson et al., 2007). Intensification of existing seasonal coastal
hypoxic zones, or shoaling of upper OMZ boundaries (currently close to shelf edge
depth) into shelf waters, could have particularly pronounced impacts on benthic (and
pelagic) fauna – with direct implications in terms of food security for large human
populations - and on biogeochemical processes.

Intensification or increased duration of coastal hypoxia could lead to increasing
occurrence of mass mortality or to reduced ability of faunal populations to recover

between hypoxic events. It would also result in expanded areas of reducing sediments and potential changes to carbon sequestration, N and P cycling and $N_2O$ emissions (Middelburg & Levin, 2009). Further, the magnitudes and the dramatic intermonsoon/monsoon (oxic/hypoxic) changes in benthic processes and nutrient fluxes seen at sites on the western Indian shelf (Pratihary et al., 2014), imply that expanded or intensified hypoxia could, through benthic-pelagic coupling, have major influences on nutrient inventories and processes occurring in shallow overlying waters.

**8. Conclusion**

The Arabian Sea and the Bay of Bengal are home to ~59% of the Earth's marine sediments exposed to severe oxygen depletion and approximately 21% of the total volume of oxygen-depleted waters. However, the Arabian Sea OMZ is larger, more intense and reveals functional anoxia in its upper part, whereas the smaller and less intense Bay of Bengal OMZ seems to be on the verge of becoming functionally anoxic. Since oxygen concentrations within this range can presently only be measured by STOX sensors, and are below detecting limits of standards methods, our understanding of the response of the nitrogen cycle to such low oxygen concentrations is based on only a few measurements and suffers from the lack of data.

Although there are a few reports on the occurrence of anoxia prior to the first large international Indian Ocean Expedition (IIOE), fully anoxic (sulfidic) events have so far not been reported from the open northern Indian Ocean (i.e. beyond coastal waters) during the last 60 years. Maintenance of functional anoxia in the Arabian Sea OMZ is highly extraordinary considering the impact of the monsoon-driven seasonality on the surface ocean circulation and the productivity in the Arabian Sea. Stable balances between physical oxygen supply and biological oxygen consumption including feedback mechanisms caused by the negative influence of decreasing oxygen concentrations on the biological oxygen consumption, seem to have prevented the occurrence of persistent anoxic conditions in the Arabian Sea OMZ and functional anoxia in the Bay of Bengal OMZ. A reduced biological oxygen consumption due to lower productivity and stronger ballast effect are in line with a less intense Bay of Bengal OMZ. The reduced oxygen consumption is largely driven by river discharges, which supply huge amounts of ballast minerals and lower the nutrient supply from subsurface waters into the sunlit surface ocean by enhancing the stratification in the surface ocean. However, there is still very little known about the interannual variability of the Indian Ocean OMZs, as there are limited long-term observational

data and the influence of the remote forcing processes that drive this variability (e.g.,
IOD and ENSO) is not fully understood.

Results obtained from the global atmosphere-ocean Kiel Climate Model and eddy
resolving regional models indicate that a decreasing inflow of oxygen-enriched water
masses from the south (ICW) intensified the Arabian Sea OMZ during the last 6000
1273 years, whereas a decreasing oxygen concentration within inflowing Persian Gulf
Water intensifies the OMZ in response to global warming. These trends significantly
affect benthic and pelagic ecosystems. The regular occurrence of *Noctiluca* is an
example of a new phenomenon that is assumed to herald a regime shift within the
pelagic ecosystem of the Arabian Sea in response to declining concentrations of
dissolved oxygen. Comprehensive studies investigating possible repercussions on
the OMZ through e.g. impacts on the export production and vertical migration and
distributions of zooplankton are missing. Accordingly, these recent changes augment
the problems that arise when trying to represent the Indian Ocean OMZ in models,
and thus in predicting the impact of the changing monsoon system on productivity
and OMZ development under global change scenarios. This holds true for the CMIP5
models and is hardly improved in the new CMIP6 models.

**9 Author contribution**

The paper was written jointly by all co-authors whereas Tim Rixen coordinated the
writing processes and co-authors focused on specific sections as listed in the
following: Sections 1 – 3 (Tim Rixen), section 4 (Zouhair Lachkar), section 5 (Birgit
Gaye and Joachim Segschneider), section 6 (Henrike Schmidt and Raleigh R.
Hood), section 7.1 (Joaquim Goes, Helga do Rosário Gomes and Arvind Singh), and
section 7.3 (Greg Cowie).

**10 Competing interests**

The authors declare that they have no conflict of interest.

**Acknowledgment**

We would like to thank the many scientists, technicians, officers and their crews of
the numerous research vessels as well as all colleagues and the various national
funding agencies that made this work possible. P. Wessels and W.H.F Smith are
acknowledged for providing the generic mapping tools (GMT).

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

**Figure Captions:**

**Figure 1**: Simplified schematic view of the nitrogen cycle

**Figure 2:** Schematic illustration showing the occurrence of microbial processes at varying oxygen levels, the subdivision of hypoxia into microbial hypoxia and functional anoxia as well as in red, detection limits of methods used to measure concentration of dissolved oxygen in sea water. Broken lines indicate processes, which occur but do not control the fate of nitrite (reduction to $N_2$ versus oxidation to nitrate).

**Figure 3**: (a, b) Monthly mean primary production rates (Behrenfeld et al., 1997) covering the periods between 2002 and 2014. (c) Minimum oxygen concentration in the water column of the Indian Ocean. Oxygen concentrations > 20 µM are indicated by the white color. The data was obtained from the World Ocean Atlas 2013 (Boyer et al., 2013). The black line indicates the extent of the secondary nitrate maximum (SNM) in 1997 (Rixen et al., 2014). The maps were produced with Generic Mapping Tool.

**Figure 4:** Oxygen profile which were obtained during the RV Meteor cruise M74/1b (Station 450) in 2007 at around 64°E and 15°N in the Arabian Sea (see Rixen et al. (2014) for further details). Blue arrows indicate processes controlling the oxygen concentrations in the OMZ such as the respiration of organic matter, fluxes of oxygen across the air-sea interface, upwelling and lateral inputs of oxygen-enriched waters, as well as the outflow of OMZ water balancing the inflow of water.

**Figure 5:** (a, b) Monthly mean sea surface temperature in the Indian Ocean (Smith et al., 2008) and the surface ocean circulation simplified and redrawn from Schott and McCreary (2001). The arrows indicate the South Equatorial Current (SEC), South Monsoon Current (SMC), Sri Lanka Dome (SD), East Indian Coastal Current (EICC), South Java Current (SJC), Indonesian Through Flow (ITF), Somali Current (SC), Great Whirl (GW), Ras al Had Jet (RHJ), West Indian Coastal Current (WICC), North Monsoon Current (NMC). The maps were produced with Generic Mapping Tool.

**Figure 6:** Conceptual illustration of the meridional overturning circulation the Indian Ocen according to Rahaman et al. (2020) and Bange et al. (2000), including the supply of oxygen via air-sea fluxes and upwelling of oxygen-enriched deep waters. The oxygen profiles were obtained during the RV Meteor cruise M74/1b and the RV

Sonne cruise SO270 in 2007 and 2019, respectively at 15°N and 15°S and approximately the same longitude (~ 64°E). ICW means Indian Ocean Central Water.

**Figure 7:** (a) The seasonal mean areal extension and the depth of the Arabian Sea OMZ. (b) Seasonal mean satellite-derived primary production which were obtained from ocean primary production website in August 2020 (http://www.science.oregonstate.edu/ocean.productivity/) and the seasonal mean oxygen concentration within the Arabian Sea OMZ. The satellite-data covered the period between 2002 and 2019 and were averaged seasonally for the Arabian Sea north of 10°. The OMZ data are obtained from Table 5 in Acharya and Panigrahi (2016).

**Figure 8:** (a) The seasonal mean satellite-derived primary production versus the seasonal mean oxygen concentration within the Arabian Sea OMZ and (b) the seasonal mean oxygen concentration versus the OMZ thickness. The data are the same as shown in figure 7.

**Figure 9:** Vertical profiles of nitrite, nitrate, and dissolved oxygen (a) as well as $\delta^{15}N$ of nitrate (b) measured during the RV Meteor cruise M74/1b in 2007. The figure was obtained from Rixen et al. (2014).

**Figure 10:** Fluxes of protected and free particulate organic carbon versus water depth (black line). The fluxes were calculated according to the equation introduced by Armstrong et al. (2002) and data measured by a sediment trap in the central Arabian Sea. The black circle shows the long-term mean organic carbon fluxes measured by sediment traps in the central Arabian Sea. The blue and broken black lines indicate concentrations of dissolved oxygen and nitrite measured during the cruise M74 in 2007 in the central Arabian Sea (Station 450). Rixen et al. (2014) provide further information about the sediment trap study and the RV Meteor cruise M74. The broken horizontal lines mark the depth of the OMZ, the surface mixed layer, and the seasonal thermocline. The latter is subdivided into an aerobic upper and a lower anaerobic part. SNM means 'Secondary Nitrite Maximum'.

**Figure 11**: Schematic sketch illustrating the role of mesoscale eddies in ventilating the Arabian Sea OMZ (a) by enhancing isopycnal mixing and water stirring (based on results from Lachkar et al (2016). (b) In the absence of eddies (e.g., in coarse-resolution global models) the along-isopycnal ventilation is reduced and the OMZ is larger and more intense, leading to potentially stronger denitrification and more compressed marine habitats.

**Figure 12:** (a) Locations of high resolution cores (circles) and areas of model simulations (boxes). (b) Increasing $\delta^{15}N$ values in high resolution cores from the Arabian Sea (note inverted scale) show increasing denitrification since about 6000 – 8000 years BP; data from the northern (yellow; light brown), eastern (red), western (blue) and southwestern (black) Arabian Sea. Sediment cores: SO90-63KA (Burdanowitz et al., 2019), RC27-23 (Altabet et al., 2002), NIOP-905P (Ivanochko et al., 2005), SK148-55 (Kessarkar et al., 2018), MD04-2876 (Pichevin et al., 2007) parallel with (c) sinking oxygen concentrations in biogeochemical model simulations driven by the Kiel Climate/PISCES Model in the northern (yellow), eastern (red), western (blue) and southern Arabian Sea (dark grey). See text for definition of regions. Model results are 20 yr running means.

**Figure 13:** Thickness of the OMZ (oxygen concentration < 20 $\mu M$) in 10 ESM from the 5[th] coupled model intercomparison project (CMIP5; Taylor et al., 2012) and in observations from oxygen climatologies of the World Ocean Atlas 2013 (Garcia et al., 2013; bottom right). The model data cover the period from 1900-1999 and are taken from the 'historical' experiment. For more information on the models see Cabré et al. 2015 (Table A1). The maps were produced with MATLALB.

**Figure 14:** (a) NOAA Suomi-VIIRS derived Chl *a* concentrations on 6[th] of Feb. 2018 showing *Noctiluca* blooms in the Sea of Oman in association with a cyclonic eddy. For projecting the Chl a concentrations the google earth low-resolution land elevation map was used © google earth (b) *Noctiluca* blooms along the coast of Muscat on 6[th] Feb. 2018.

**Figure 15**: A summary of water-column conditions, sediment properties, benthic communities and processes influencing carbon cycling across the OMZ on the Indus margin of the Arabian Sea (modified from Cowie and Levin (2009a) and reprinted with the permission of Elsevier). Water-column dissolved oxygen (DO) concentration profiles are shown for intermonsoon (April - May) and late-to-postmonsoon (September-October) periods. Organic carbon ($C_{org}$) concentrations (weight percent) are for surficial (0 - 2 cm) sediments. Vertical shaded zone indicates OMZ boundaries as defined by DO ≤ 0.5 ml/l. Shaded depth ranges denote the OMZ core (~250 - 750 m, near-uniform DO of ≤ 0.1 ml/l), a lower OMZ transition zone (~750-1300 m) in which DO and the numbers of and activity of macrofauna increase with station depth, and a seasonally hypoxic zone (~100-250 m) in which the upper OMZ

boundary shoals during the summer monsoon season. Faunal classes are as defined by Gooday et al (2009).

Figure 1

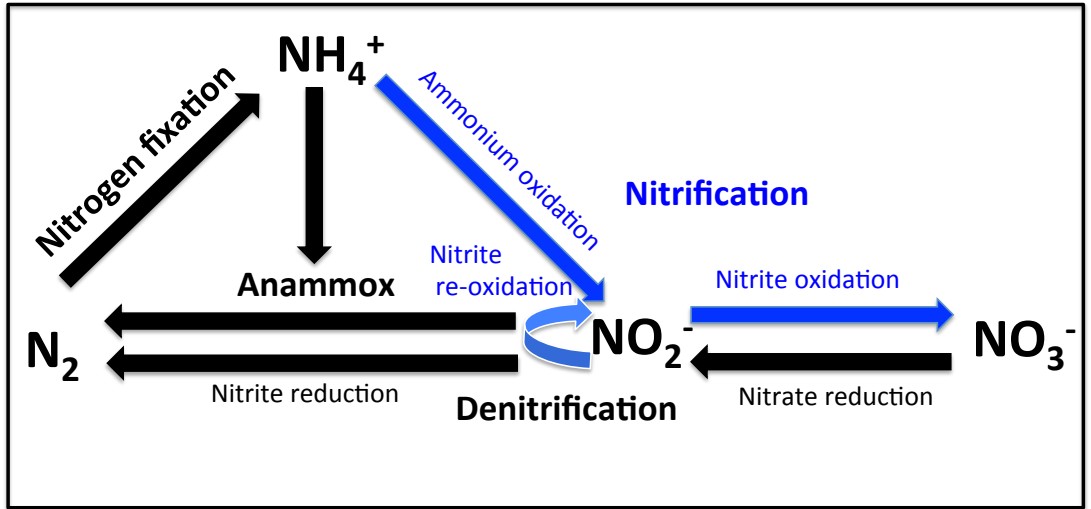

Figure 2

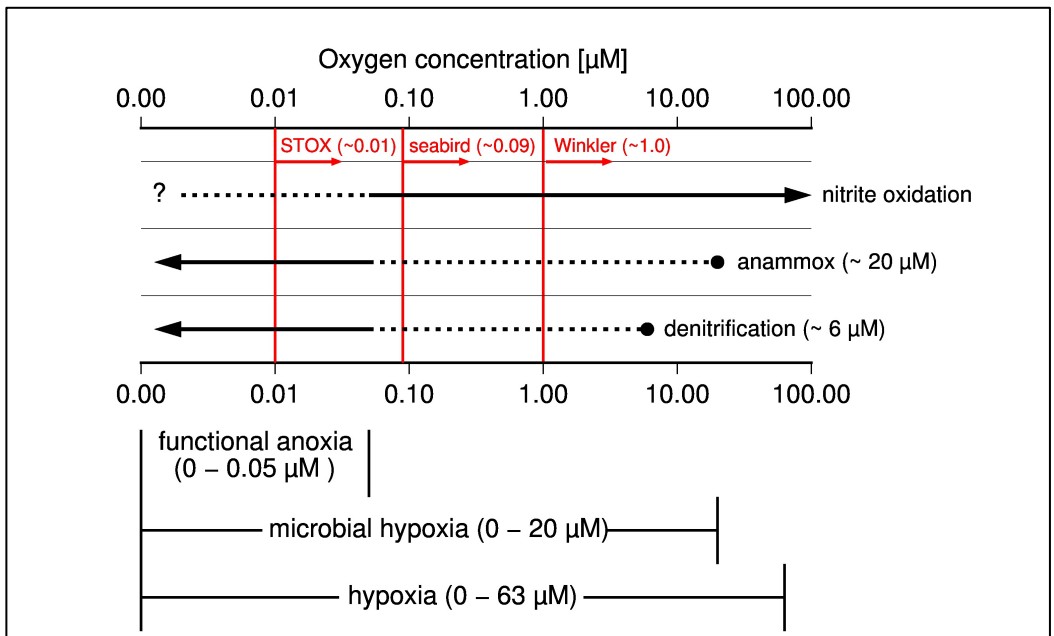

Figure 3

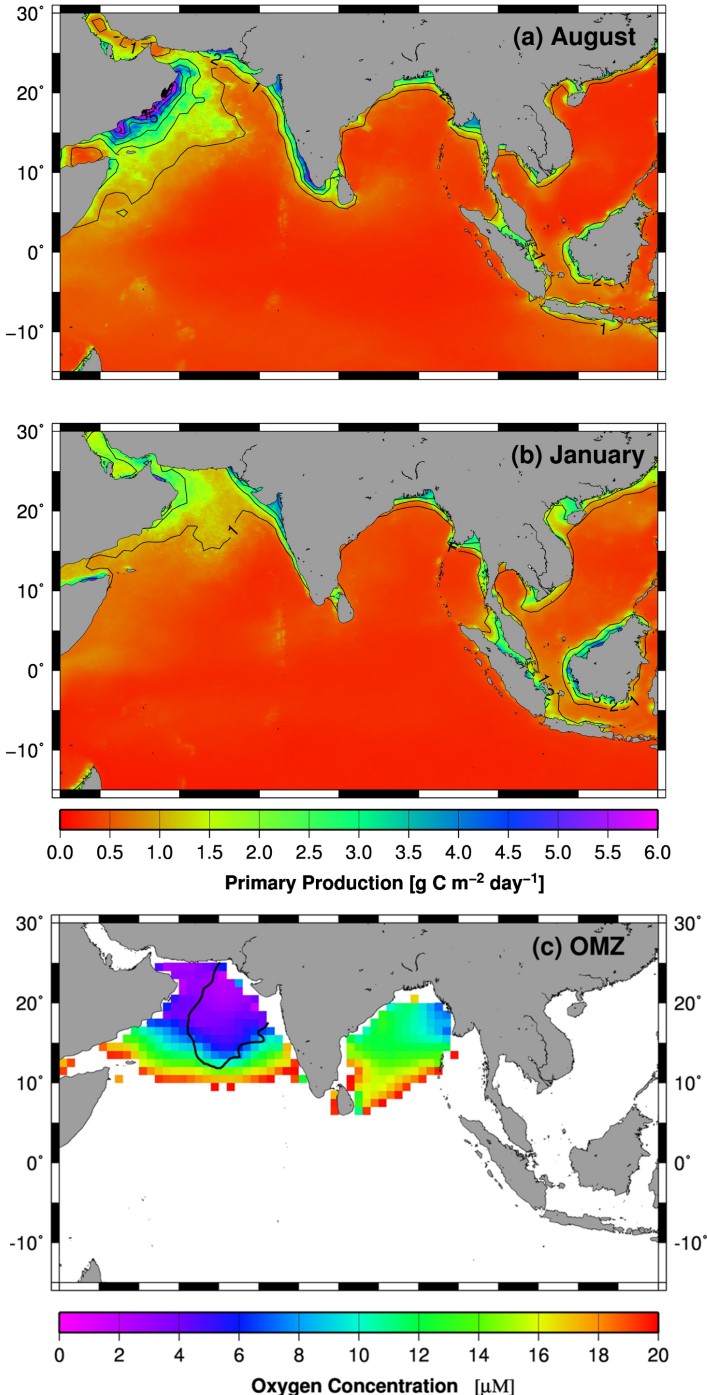

Figure 4

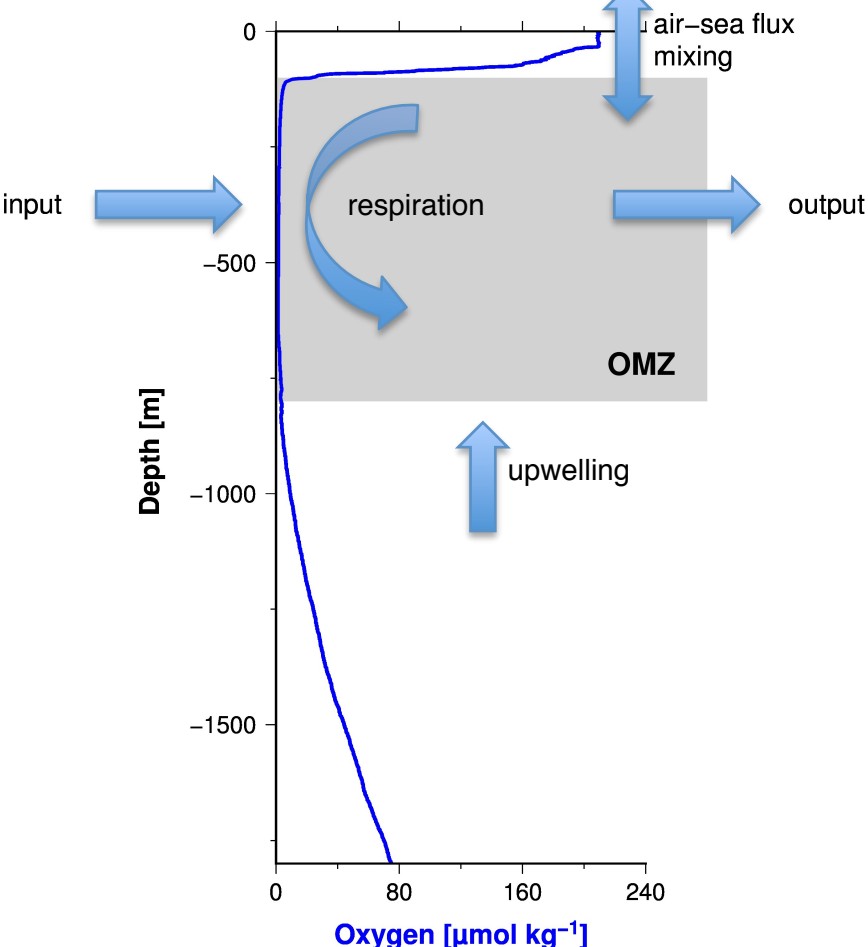

Figure 5

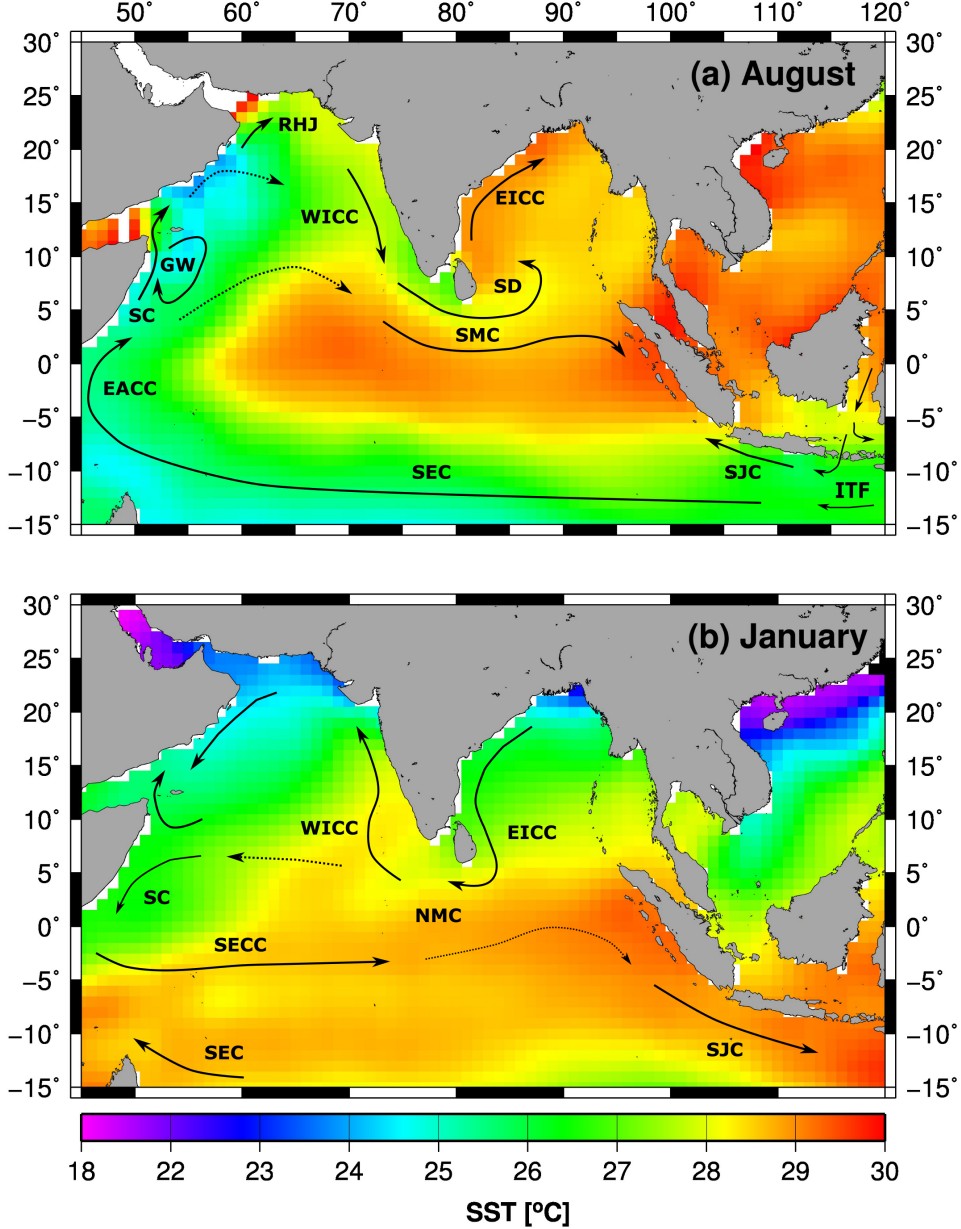

Figure 6

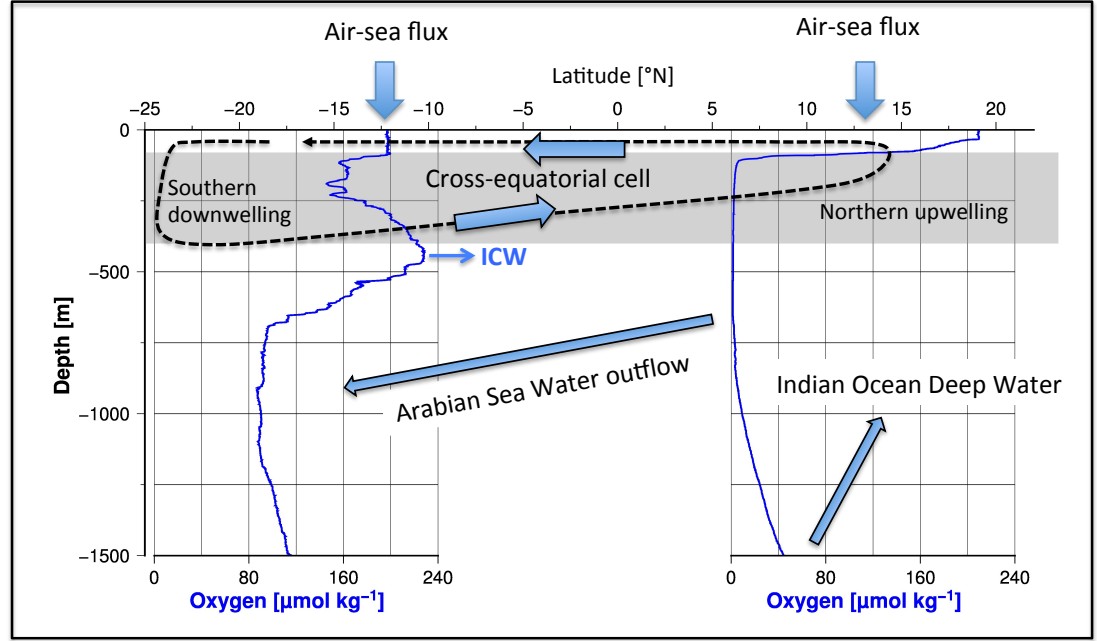

Figure 7

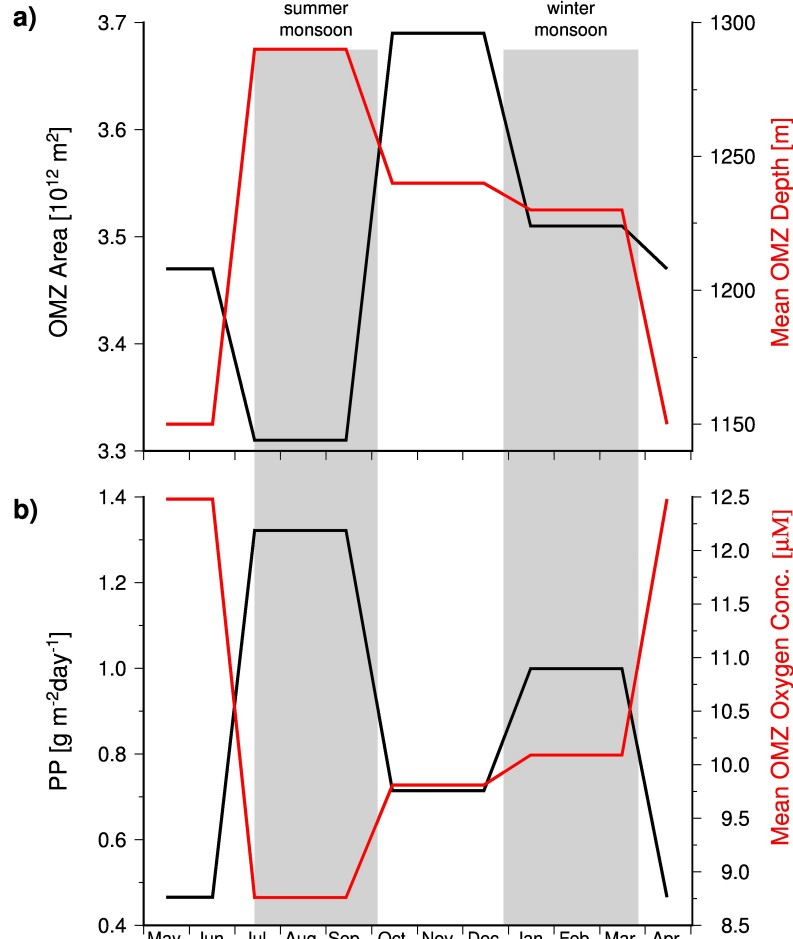

Figure 8

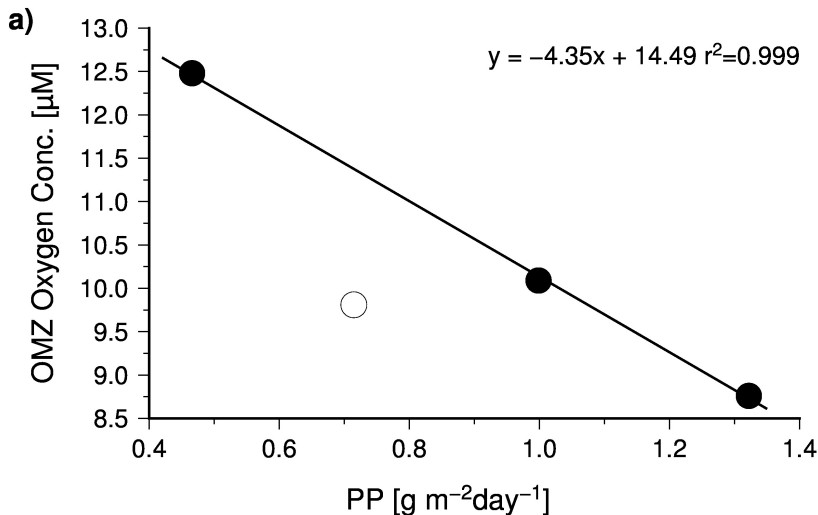

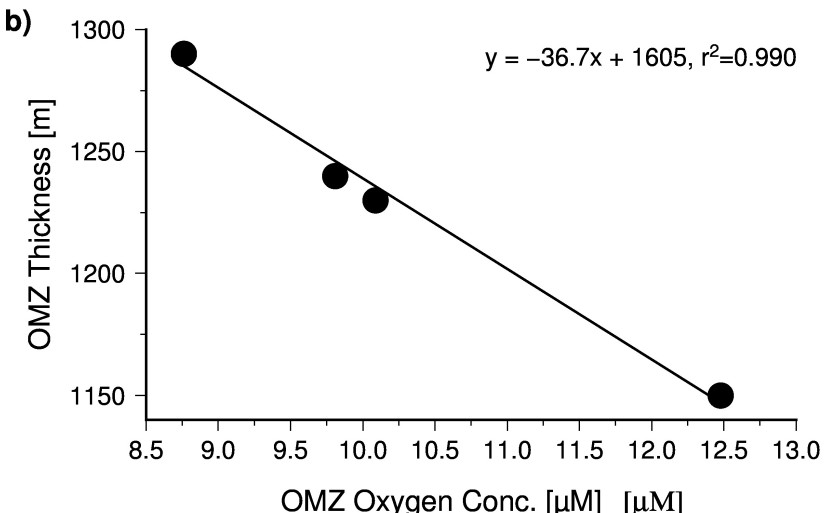

Figure 9

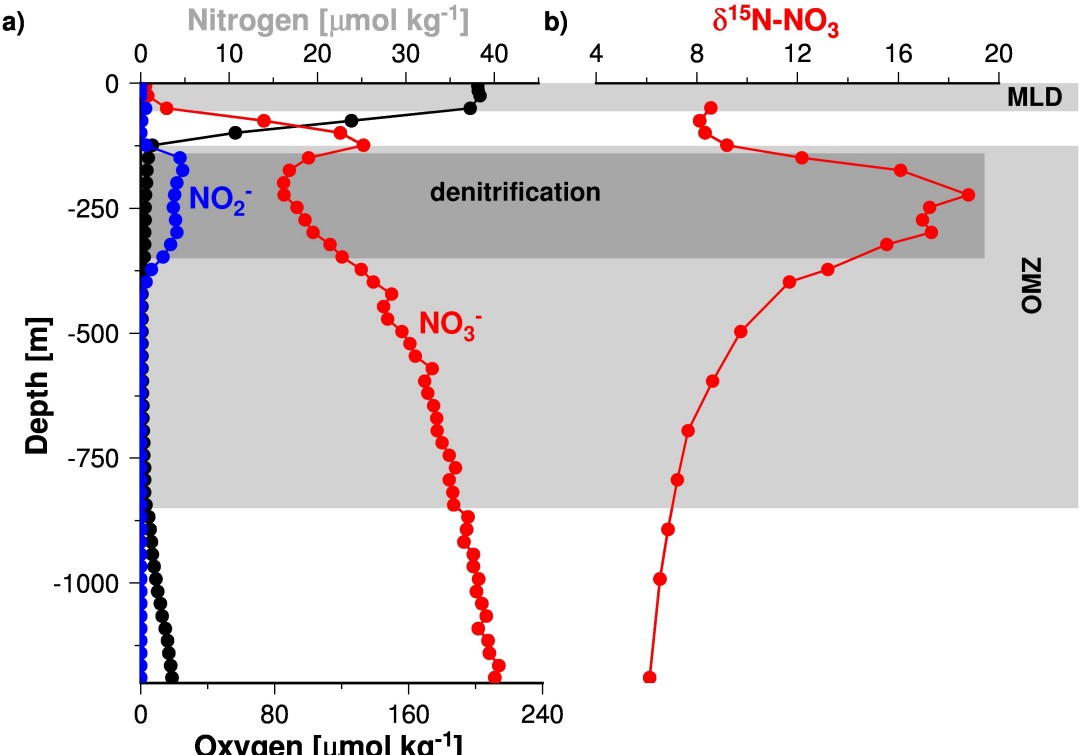

Figure 10

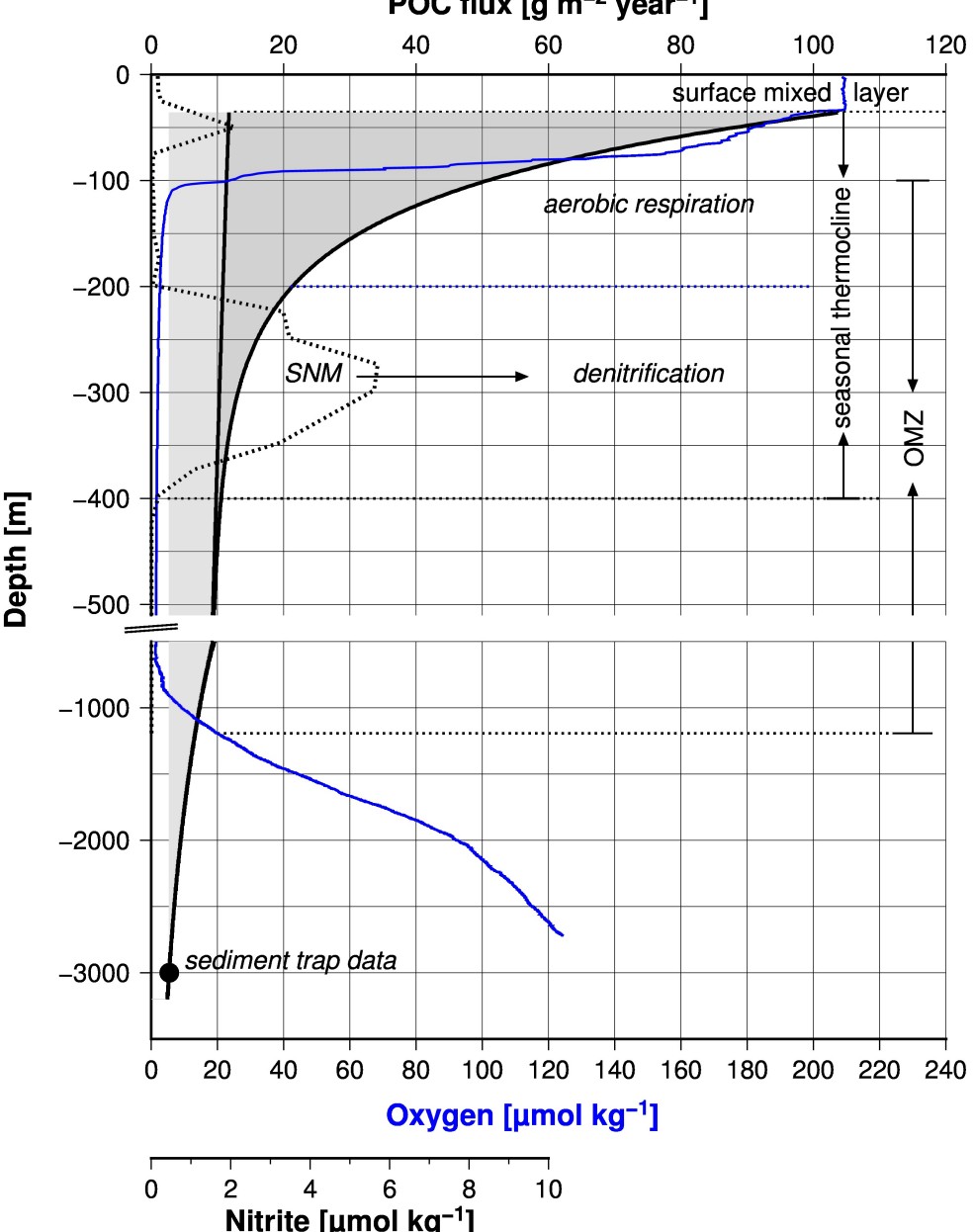

Figure 11

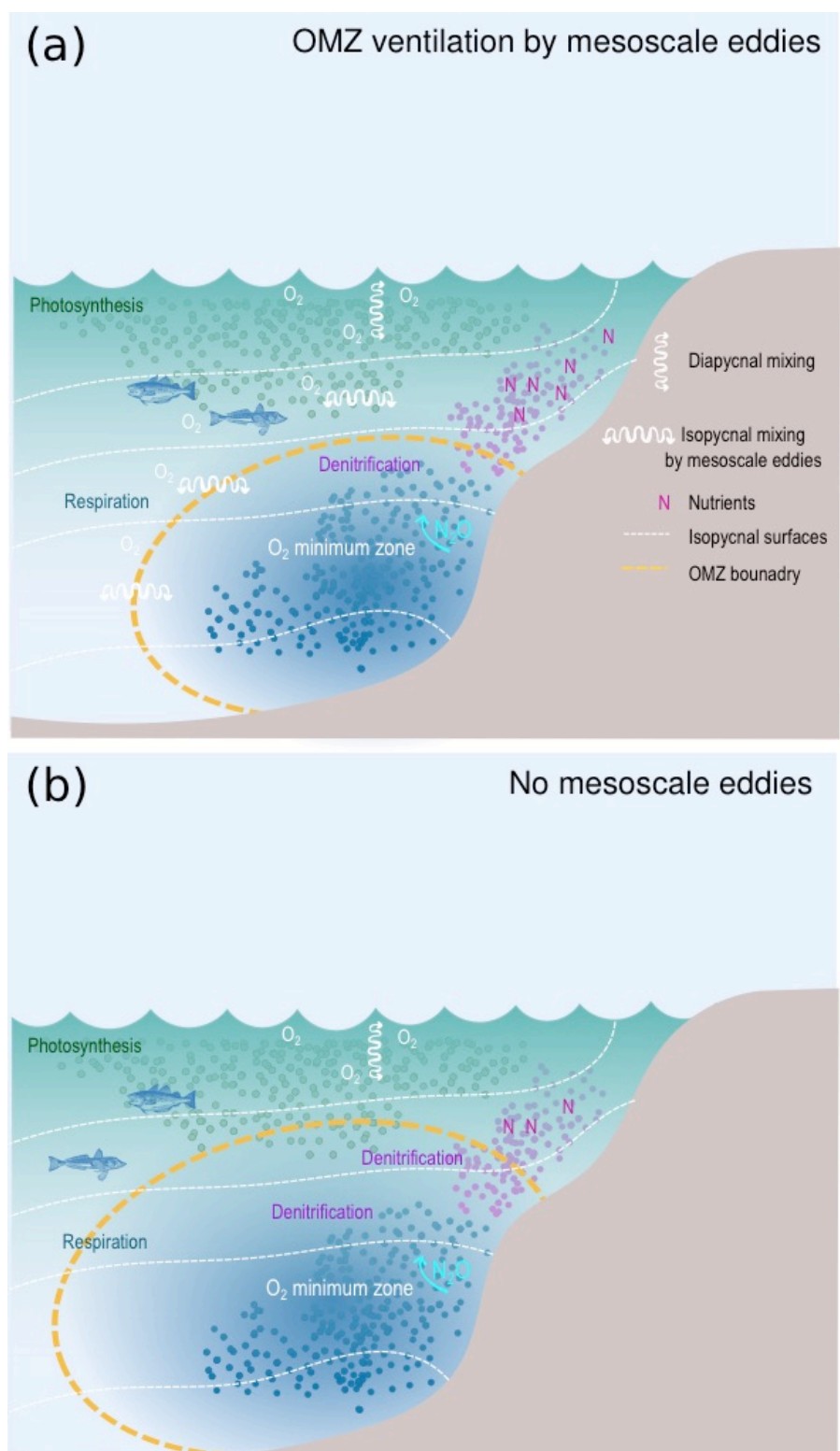

Figure 12

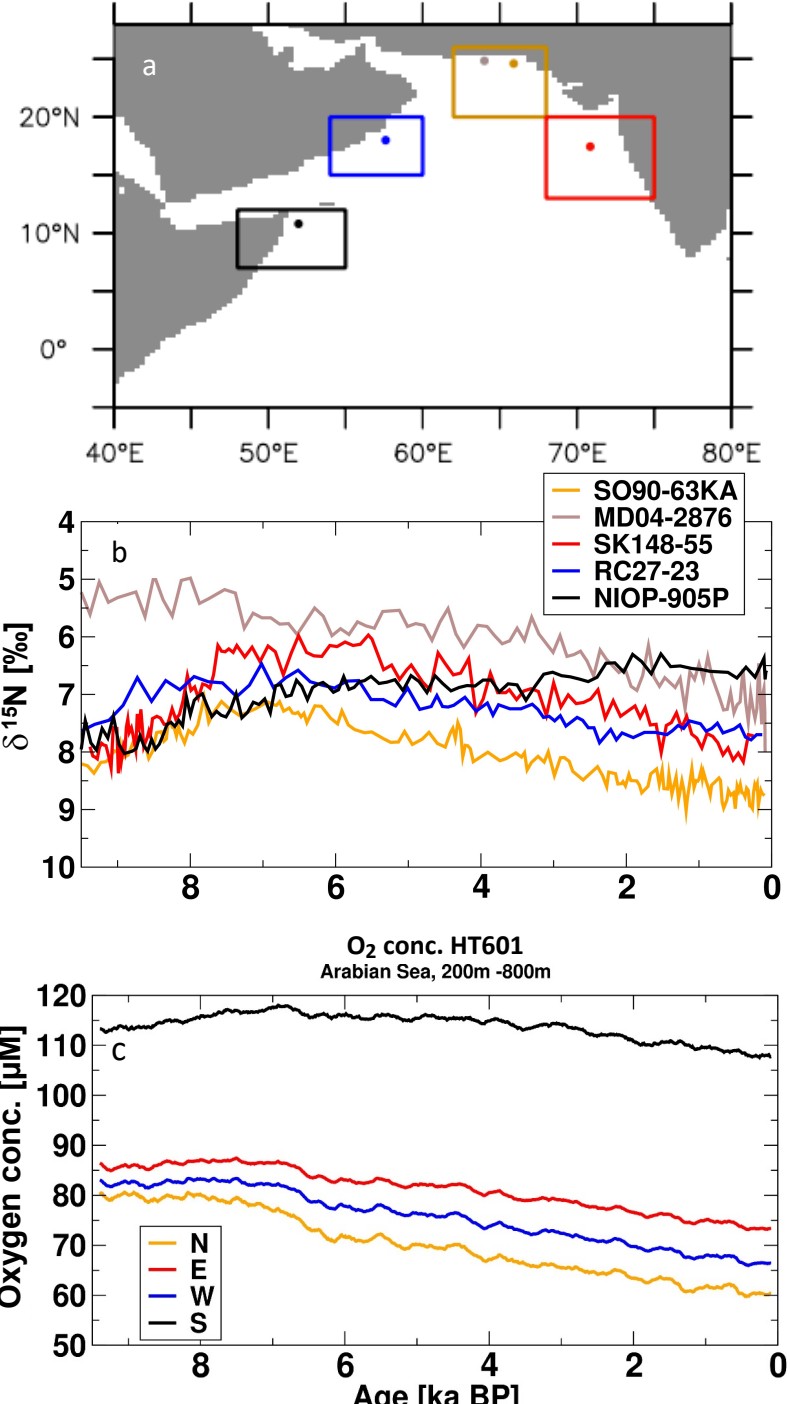

Figure 13

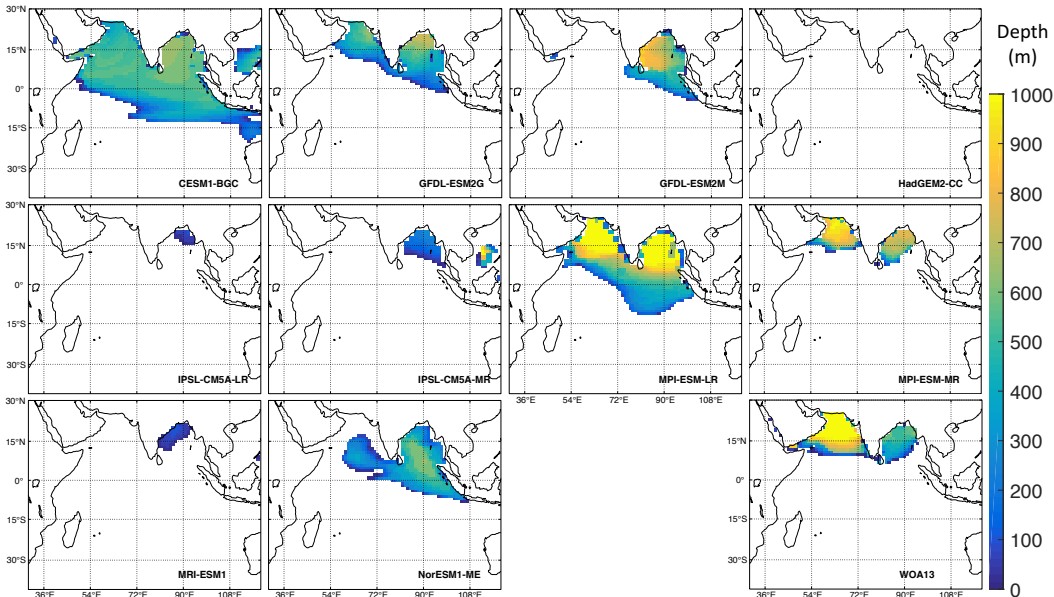

Figure 14

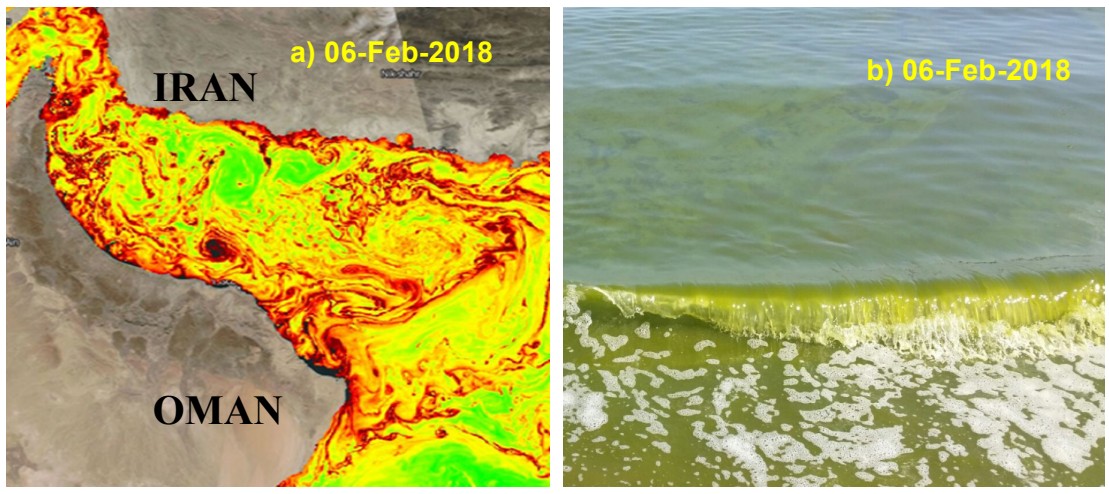

Figure 15

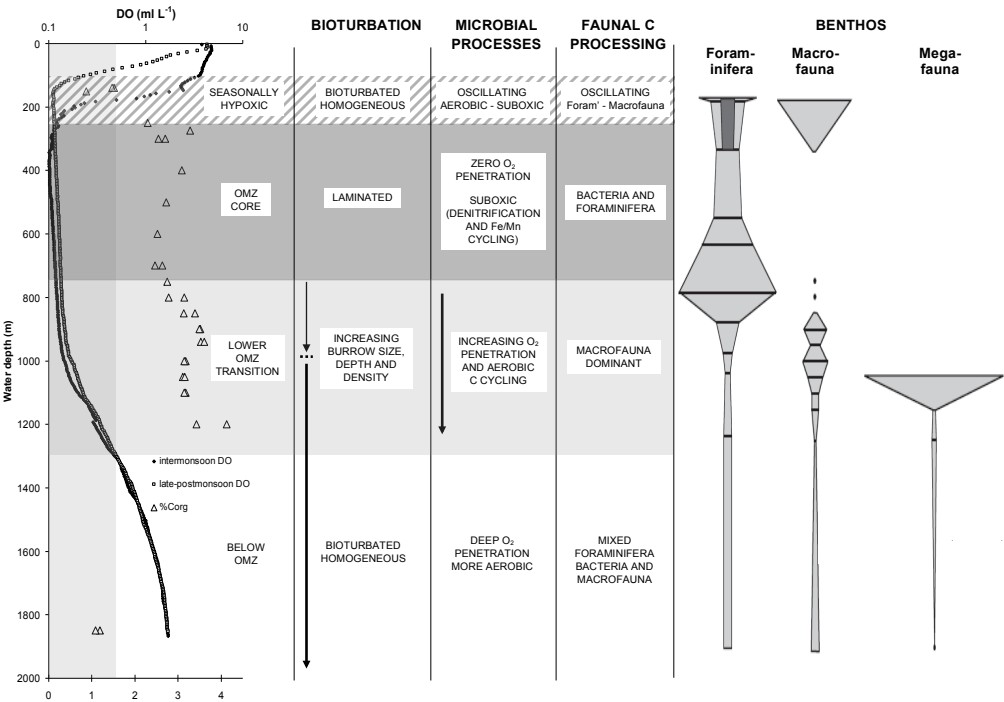