# Peer review of "Present, past, and future of the oxygen minimum zone in the northern Indian Ocean"

_Biogeosciences, 2020_

## Referee Comment (RC1) · Anonymous Referee #1 · 6 May 2020

Review of Rixen et al 'Present, past and future of the OMZ in the northern Indian Ocean'

Rixen et al provide a review on the comparison of the two oxygen minimum zones in the northern Indian Ocean, located in the Bay of Bengal (BoB) and the Arabian Sea (AS). The two basins are compared from a oceanographic and biogeochemical point of view. This is obviously a challenge and I acknowledge that it is never an easy task to synthesize results from different disciplines and authors into a coherent piece of writing. To me the manuscript is a valuable contribution, however, it needs some more streamlining and integration of the different sections.

That said, I have some very general remarks, which I hope will help to streamline the manuscript:

I am not sure what is meant by hypoxic, in order to be able to stick with it may be helpful to define a range of oxygen concentrations you are referring to.

I am not quite sure what the aim of the study is, and what it specifically contributes as a stand-alone publication. From the title, I would expect to learn about potential expansion and intensification patterns of the two OMZs based on an assessment of past developments on geological timescales. This can, to a certain extent, be distilled out of the paper but I could imagine that if the authors take some effort and work through the paper once more, it would be more obvious. I also expected to learn about why those two basins behave so differently- there are different reasons given, including stratification, which is credible for a certain part of the BoB but not visible anymore in the offshore OMZ as presented by Bristow, further a ballasting effect by riverine particles present in the BoB and absent in the AS. The latter doesn't convince me, because it seems to be a coastal phenomenon only.

Regarding the different oxygen concentrations in the two basins, a steady state between physical oxygen supply and biological oxygen consumption is also given. The slightly higher oxygen concentrations in the BoB is suggested to promote a feedback between nitrate reduction and nitrite oxidation. This is based on Bristow et al mainly, which is one study with 5 stations during one time of the year. In order to strengthen your case, it may be beneficial to also consider Canfield et al. (2019) and Löscher et al. (2020), both of which propose alternative feedbacks possibly stabilizing the BoB's remaining oxygen traces. I understand that those studies may have come out after the presented paper was submitted and may have not been visible enough. Other results on OMZ oxygen production as suggested for other regions (Garcia-Robledo et al., 2017) may be worthwhile considering given the abundance of small unicellular cyanobacteria as described for the BoB. As for the assessment of deep time changes, reference to the work of Orsi et al. (2017) could be helpful.

In addition, but this indeed may go beyond the scope of the manuscript, a discussion on possibly changing monsoon intensities and atmospheric dust inputs could be interesting for a future assessment.

Right now, it is obvious that the sections have been written by different authors, with sections 1-3 needing a native speaker to improve the language. I understand that the first author coordinated the writing and I know that this is an ungrateful job. However, there needs to be some more coherence regarding the writing style, the level on which the different topics are presented, and again, some more integration of the different sections to improve the reading flow.

I also have some specific comments and suggestions:

Title: there is a comma missing between present and past.

l. 2 'is' should be changed to 'are'; 'it favors' should be changed to 'they favor'. I also do not quite understand the use of the (admittedly modern) expression 'ecosystem services'

l. 4: change 'which' to 'and its'

l. 8/10 past tense is used- is this because it refers to the geological past?

l. 25: 600 mio years is a bit short. Canfield, Lenton and Lyons give different ranges, but they are about 2.3-3.2 billion of years for the rise of oxygen.

l. 27 ff I am not sure what this means? Are you suggesting those are the only habitats of anaerobic organisms? Because they are quite abundant throughout the marine water column on particles see e.g. (Ganesh et al., 2015;Ganesh et al., 2014). In addition, nitrate reduction to N2 can happen via anammox- in this case one could more or less claim those are anaerobic microbes. Denitrifiers are not anaerobic microbes, they are facultative and respire oxygen when possible.

l. 39 'of' is missing before 'oxygen'

l. 49 'expense'

l. 62 Here, a definition of hypoxic and anoxic would be helpful. Also, this way to abbreviate looks very awkward. Change to 'inhibit', 'prevent'

l. 66 'of' before 'anaerobic' is missing

l. 70 a reference to work by Schmidtko et al. (2017) and Keeling et al. (2010)is missing

l. 75'margins'

l. 77 Again, this needs a definition of hypoxia.

l. 85 ff a reference to Naqvi et al. (2010) is missing.

l. 166 How do the different primary producer communities look? How is the food web- wouldn't this also be important to make claims about export fluxes? Also, if we have a faster export, would the a more anoxic sediment or deeper water layer be expected?

This statement is also somewhat contradictory to the claim made based on Bristow et al, that a microbial feed back stabilizes the trace oxygen concentrations. I would suggest mentioning the reasons for the difference in OMZ intensity in a way that is less exclusive and so that they can complement each other. The way it is, it is confusing.

l. 174 this needs a reference

l. 174 ff The statement is unclear, I think you are talking about a sulfidic event when saying anoxia

l. 176 if this is an 'only report' why do you have three references?

l. 178 'don't seem to evolve every year'

l. 179 Who dies during those mass mortalities? Please replace ' in between' with 'occasionally'. How confident are we that those mass mortalities do not result from trace metal contaminations from the land?

L 183 'also' could be removed, sounds awkward.

l. 184 change 'were' to 'was'

l. 187 Actually, Bristow shows microaerobic processes to occur

l. 190 ff awkward sentence, please rephrase

l. 192 Please add an explanation what excess $N_2$ measurements are good for. I don't think a non-N cycle expert can possibly know that.

l. 199 'outcompetes'

l. 202 remove 'the', change 'rate implies' to 'rates may explain'

l. 204 ff, l. 206ff Please rephrase- awkward sentences.

l. 208 'Follow-up studies also reported'

l. 212 ff Schunck et al didn't report on periodic outbreaks but on a one-time event, another report from the same region would be Callbeck et al. (2018), both references combined may give some hint for a regular occurrence.

l. 215 this may also just be a result of the monitoring program. If no one went there to measure one wouldn't find it either.

l. 220 remove 'the' before 'biological'

l. 221 'This approach is based on'

l. 223 what does 'regarded' mean here? The water masses of interest?

l. 232 'approximately'

l. 238/ 239 I don't understand this statement.

l. 258 'isotope ratio'

l. 261 ff it would be helpful to explain which values are typical for denitrification and other processes

l. 262 change 'indicates to 'is located in'

l. 268 ff this statement doesn't make sense to me, the reference is also maybe not ideal.

l. 270 'SNM'

l 272 'suggests'

l. 273 ff What is the purpose of this statement?

l. 279 'within'

l. 280 'key factor'

l. 315 This could benefit from a reference

Generally, I was missing references to work on eddies in Atlantic OMZ waters and their relevance for oxygen budgets and biogeochemistry (Fiedler et al., 2016;Karstensen et al., 2017;Schütte et al., 2016), especially in lines 366 ff.

l. 391 remove 'the'

l. 392 what is meant by 'nitrogen'? $N_2$, organic or inorganic nitrogen species?

l. 392-405 this section would benefit from an explanation of what those values mean.

l. 397 remove 'the'

l. 407 the core has the lowest oxygen concentrations?

l. 419 this sentence seems to be missing something

l. 428 'an onset'

l. 422 ff, this part would benefit from observations by Orsi et al. (2017)

l. 429 ff I don't understand the purpose of this statement

l. 430, the abbreviation ICW is only explained in l. 439

l. 433 what does BP stad for?

l. 436'surfac- derived oxygen-rich water'

l. 448 Kiel Climate Model, introduce the abbreviation as you use it later on, also this needs a reference.

l. 454 explain what PISCES stands for

l 485 'the' before 'late'

l. 413 what does that mean that it is backward? Replace 'oxygen values' with 'oxygen concentrations or saturations' whatever is appropriate

l. 516 there are high resolution options including mesoscale dynamics in CMIP6

l. 519 Isn't it rather a general problem that there is no circulation model available?

L 579 ff this section is lengthy and could lead better to the point

L 612 'Arabian Sea'

L 721 what is and 'edge effect'?

References

Callbeck, C. M., Lavik, G., Ferdelman, T. G., Fuchs, B., Gruber-Vodicka, H. R., Hach, P. F., Littmann, S., Schoffelen, N. J., Kalvelage, T., Thomsen, S., Schunck, H., Löscher, C. R., Schmitz, R. A., and Kuypers, M. M. M.: Oxygen minimum zone cryptic sulfur cycling sustained by offshore transport of key sulfur oxidizing bacteria, Nature Communications, 9, 1729, 10.1038/s41467-018-04041-x, 2018.

Canfield, D. E., Kraft, B., Löscher, C. R., Boyle, R. A., Thamdrup, B., and Stewart, F. J.: The regulation of oxygen to low concentrations in marine oxygen-minimum zones, Journal of Marine Research, 77, 297-324, 10.1357/002224019828410548, 2019.

Fiedler, B., Grundle, D. S., Schütte, F., Karstensen, J., Löscher, C. R., Hauss, H., Wagner, H., Loginova, A., Kiko, R., Silva, P., Tanhua, T., and Körtzinger, A.: Oxygen utilization and downward carbon flux in an oxygen-depleted eddy in the eastern tropical North Atlantic, Biogeosciences, 13, 5633–5647, 10.5194/bg-13-5633-2016, 2016.

Ganesh, S., Parris, D. J., DeLong, E. F., and Stewart, F. J.: Metagenomic analysis of size-fractionated picoplankton in a marine oxygen minimum zone, ISME J, 8, 187-211, 2014.

Ganesh, S., Bristow, L. A., Larsen, M., Sarode, N., Thamdrup, B., and Stewart, F. J.: Size-fraction partitioning of community gene transcription and nitrogen metabolism in a marine oxygen minimum zone, ISME J, 1-15, doi:10.1038/ismej.2015.44, 2015.

Garcia-Robledo, E., Padilla, C. C., Aldunate, M., Stewart, F. J., Ulloa, O., Paulmier, A., Gregori, G., and Revsbech, N. P.: Cryptic oxygen cycling in anoxic marine zones, 114, 8319-8324, 10.1073/pnas.1619844114 %J Proceedings of the National Academy of Sciences, 2017.

Karstensen, J., Schütte, F., Pietri, A., Krahmann, G., Fiedler, B., Grundle, D., Hauss, H., Körtzinger, A., Löscher, C. R., Testor, P., Vieira, N., and Visbeck, M.: Upwelling and isolation in oxygen-depleted anticyclonic modewater eddies and implications for nitrate cycling, Biogeosciences, 14, 2167–2181, 10.5194/bg-14-2167-2017, 2017.

Keeling, R. F., Kortzinger, A., and Gruber, N.: Ocean Deoxygenation in a Warming World, in: Annual Review of Marine Science, Annual Review of Marine Science, Annual Reviews, Palo Alto, 199-229, 2010.

Löscher, C. R., Mohr, W., Bange, H. W., and Canfield, D. E.: No nitrogen fixation in the Bay of Bengal?, Biogeosciences 17, 851–864, 10.5194/bg-17-851-2020, 2020.

Naqvi, S. W. A., Naik, H., D'Souza, W., Narvekar, P. V., Paropkari, A. L., and Bange, H. W.: Carbon and nitrogen fluxes in the North Indian Ocean, in: Carbon and nutrient fluxes in continental margins: A global synthesis, edited by: Liu, K.-K., Atkinson, L., Quiñones, R., and Talaue-McManus, L., Springer-Verlag, New York, 180-191, 2010.

Orsi, W. D., Coolen, M. J. L., Wuchter, C., He, L., More, K. D., Irigoien, X., Chust, G., Johnson, C., Hemingway, J. D., Lee, M., Galy, V., and Giosan, L.: Climate oscillations reflected within the microbiome of Arabian Sea sediments, Scientific Reports, 7, 6040, 10.1038/s41598-017-05590-9, 2017.

Schmidtko, S., Stramma, L., and Visbeck, M.: Decline in global oceanic oxygen content during the past five decades, Nature, 542, 335-339, 10.1038/nature21399, 2017.

Schütte, F., Karstensen, J., Krahmann, G., Hauss, H., Fiedler, B., Brandt, P., Visbeck, M., and Körtzinger, A.: Characterization of "dead-zone" eddies in the eastern tropical North Atlantic, Biogeosciences, 13, 5865-5881, 10.5194/bg-13-5865-2016, 2016.

---

## Referee Comment (RC2) · Anonymous Referee #2 · 18 May 2020

**General comment**

The manuscript by Rixen et al. gives a good overview of the development of OMZs and recent trends in the Arabian Sea and the Bay of Bengal, and discusses impacts from ocean circulation, export production and mesoscale eddies. The paper also reviews past and potential future OMZ strength as inferred from the sediment $\delta^{15}N$ and model predictions and looks at the pelagic and benthic ecosystem responses.

The paper is generally sound and has the potential to present a much needed comprehensive review of the Arabian Sea OMZs. It is however, quite apparent that the different sections were written by different authors as the writing style and the transitions from one section to another are incongruent. The manuscript should be at least partially rewritten to improve the flow. Some sections clearly need editing by an native English speaker (in particular the abstract, introduction, and conclusion). Some sentences are unclear or repetitive. I found several typos through the manuscript. See my technical corrections below for some suggestions on how to improve these sections.

On another note, the adopted $O_2$ thresholds defining hypoxia and anoxia are confusing. Several papers use a much higher $O_2$ threshold to define hypoxia (e.g., >63 µM, Vaquer-Sunyer and Duarte, 2008 and reference therein). The presence of $H_2S$ should rather be referred to as sulfidic conditions, as sulfate reduction does not necessarily occur under anoxic conditions. A clear distinction between Oxygen Minimum Zones (OMZs) and Oxygen Deficient Zones (ODZs) should also be made.

Finally, some figures should be added or improved for clarity. For instance, a figure explaining the development of an OMZ in relation to ocean circulation and seasonal monsoons would be helpful (Background, section 2.2).

**Specific comments:**

**Abstract:**

Overall, the whole abstract ought to be rewritten to summarize the main points of the manuscript. The current version is confusing, and at times vague. Also, the abstract should follow a more logical order following the order of the different sections as presented in the text.

Line 4: Nitrate loss is only a problem if it is limiting (i.e., in a non-eutrophic system).

Lines 14-16: This sentence is confusing and needs clarification. It should be rephrased to emphasize that, based on previous studies (e.g., Aumont *et al.*, 2015), decreasing oxygen concentration slows down respiration and thus decreases oxygen demand. The following sentence is also unclear as it is.

Lines 19-21: This sentence is too vague. Effects on benthic and pelagic ecosystems should be better summarized.

**Introduction**

Lines 27-30: This sentence is ambiguously worded.

Lines 30-32: $N_2O$ is also produced as an intermediate during denitrification and is a by-product of nitrification. $N_2O$ is a greenhouse gas 300 times more potent than $CO_2$ and an ozone destructing substance and should also be included here.

Lines 46-48: The availability and quality (organic matter stoichiometry) of organic material is a key control on denitrification versus anammox (Babbin *et al.*, 2014).

Lines 60-61: Their definitiona of hypoxia and anoxia are a bit confusing since most studies define hypoxia at $O_2$ concentrations >63 μM (Vaquer-Sunyer and Duarte, 2008 and reference therein). Anoxia ($O_2$ concentrations close to zero or in the nmol range) can also occur without hydrogen sulfide production.

Lines 66-67: I don't quite understand this sentence. Marine ecosystem services need to be defined earlier in the text. I suggest removing this sentence as the next sentence (lines 67-70) articulates the same idea better.

Lines 80-82: The more recent estimates by Eugster and Gruber (2014) of 52 Tg N $yr^{-1}$ for water column denitrification and 93 Tg N $yr^{-1}$ for benthic denitrification should be referenced. The distinction between water column and benthic rates should be made more explicitly.

Lines 84-85: Considering a mean sedimentary denitrification rate by Eugster and Gruber (2014) of 93 Tg N $yr^{-1}$, the proportion of sedimentary denitrification at the Pakistan continental margin could be even higher.

**Main text**

Lines 124-151: A figure showing the impact of ocean circulation in relation to seasonal monsoon on OMZ expansion in the eastern and western Arabian Sea would be helpful.

Lines 144-146: Is this low areal extension associated with increased thickness of the ODZ, as shown in Figure 4? This should be clarified here.

Lines 166-168: Why this ballast-effect mostly occurring in the Bay of Bengal and not the Arabian Sea?

Lines 180-182: A distinction should be made between human-induced coastal eutrophication and coastal dead zone development due to the imbalance between higher $O_2$ consumption from primary productivity (upwelling) relative to $O_2$ supply from physical circulation.

Lines 198-199: A reference is needed to support this $O_2$ threshold.

Lines 191-193: How does the relatively low denitrification rate estimated by Bristow *et al.* (2017) compares to the denitrification rate (including anammox) measured in the Arabian Sea using [15]N-labeled incubations by Ward *et al.* (2009)?

Lines 215-217: This is an important point that should be described better in the abstract.

Line 229: The term (central Indian Ocean) is already defined in the previous section.

Lines 267-268: I don't quite understand this sentence either. Do they mean in contrast to the upper part of the SNM?

Lines 277-279: How does figure 3 support this point?

Lines 319-323: The roles of coastal mode water anticyclonic eddies as N-loss hotspot in the Peru upwelling system should also be referenced (Bourbonnais *et al.*, 2015; Altabet and Bourbonnais, 2019). The paper by Fassbender *et al.* (2018) also provides a good review of the effects of mesoscale and submesoscale features on ocean biogeochemistry.

Lines 339-342: On which timescale are these feedbacks expected to occur?

Lines 379-381: These two terms "eddy-driven isopycnal tracer mixing" and "isopycnal flattening" need to be explained.

Lines 419-420: The authors should be more specific about which results they are referring to (Bristow *et al.*, 2017).

Line 471: A reference is needed to support this $O_2$ threshold for denitrification. Dalsgaard *et al.* (2014) report an $O_2$ threshold in the nmol range for denitrification.

Lines 486-488: The model's results do not seems to support denitrification during the Holocene.

Lines 497-498: Was a relationship between orbital forcing (i.e., Milankovitch cycles) and the development of the OMZ ever investigated in the region? A reference should be added.

Lines 515-519: Submesoscale processes, which are ephemeral and take place over lengths of about 1-10 km lasting several days, are also poorly represented (see Fassbender *et al.*, 2018).

Lines 600-616: What is the effect of these larges blooms on OMZ expansion?

Lines 702-705 and 715-720: At which oxygen thresholds are these community composition and faunal abundance changes observed?

Lines 745-748: What is the N:P ratio in the overlying ODZ versus the sediments? Lower N:P ratios than expected based on $NO_3^-$ loss and biogenic $N_2$ production during denitrification are often observed in coastal ODZs due to the preferential release of $PO_4^{3-}$ following iron and manganese oxyhydroxide dissolution in anoxic sediments (Noffke *et al.*, 2012).

Line 750: Define "dark" carbon.

Lines 797-800: Higher oxygen concentrations are more likely the results of the development of a sharper pycnocline (from higher freshwater fluxes) and lower primary productivity in the Bay of Bengal.

**Conclusion:**

Lines 800-802: This sentence is unclear. Do they mean that mesoscale eddies sustain higher $O_2$ concentrations in the OMZ than expected in their absence?

**Figures:** AOU should be showed instead of $\Delta$oxygen ($\mu$mol kg$^{-1}$) in this figure since this is what is discussed in the text. Something must be wrong with the scale for $\Delta$oxygen ($\mu$mol kg$^{-1}$). The $\Delta$oxygen (deviation from $O_2$ concentrations at saturation) should be much higher than 10 $\mu$mol kg$^{-1}$ to cause hypoxic/anoxic conditions.
Why is the figure broken into two panels (a, b)? Another suggestion is the break the axis for depth >500 m.

Figure 4. It is unclear how to reconcile data in Figure 2 - showing that overall a decrease in the OMZ area seems to correspond to a decrease in the mean OMZ oxygen concentrations (at least during summer monsoon when POC flux is highest) and Figure 4 - showing a negative correlation between OMZ max thickness and the mean OMZ oxygen concentration.

Figure 5. This figure is difficult to read (white font on light blue background). Font size should be bigger. Isopycnal mixing by mesoscale eddies could be emphasized in a.

Figure 6. Make d as a symbol (y axis): "$\delta^{15}N$"

**Technical corrections:**

Line 4: Change "increases the loss nitrate" to "increases nitrate loss"

Lines 4-5: Change to "Nitrate is a macronutrient limiting primary productivity in most of the ocean."

Lines 7-10: This sentence seems to be out of context and repetitive considering the following sentence. I suggest rewriting:
"The main control on oxygen concentrations in the Arabian Sea and the Bay of Bengal is the balance between physical oxygen supply and biological oxygen consumption from respiration. Mesoscale eddies greatly enhance mixing and advection of $O_2$-rich waters, which compensate biological consumption and overall reduces ODZ expansion."

Lines 12-14: Change to: "However, due to slightly higher oxygen concentrations, aerobic nitrite oxidation outcompete anaerobic nitrite reduction and thus limits denitrification in the Bay of Bengal"

Line 39: Replace "At" with "Under" at the beginning of sentence.

Lines 62 and 64 and 806: Change "hyp-" for hypoxic here and everywhere else in the text.

Line 74: Remove "is": "..., with a much smaller proportion  located in the Bay of Bengal..."

Line 83: Replace "to this data" with "published data".

Line 90: Change to: "one of the least understood OMZs"

Line 183: Replace "conational" for "continental"

Line 190: Replace "nitrite oxidization" with "nitrite oxidation"

Line 201: Replace "this is with about 0.7 μM much higher" by "it is about 0.7 μM higher".

Line 202: Remove the at beginning of sentence: "However,  in comparison to the Arabian Sea..." and remove "as in the Arabian Sea" at the end of sentence.

Lines 208-209: Replace with: "Subsequent studies also reported decreasing oxygen concentrations in the western and northern Arabian Sea."

Line 255: Replace with: "in the upper part of the seasonal thermocline..."

Line 258: Replace "stabile" with "stable"

Line 270: Replace "SNN" with "SNM"

Lines 279-281: Therewith is used twice within the same sentence.

Line 286: Remove "in": "... is mostly remineralized within  the upper 300 m..."

Line 292: Remove "also"

Line 297: Replace "the hypothesis" by "this hypothesis"

Line 439: This term (ICW) is already defined earlier in the text.

Lines 534 and 539: Change for 80 μM $O_2$ and 50 μM $O_2$.

Line 573: Replace for: "... can survive at $O_2$ concentrations down to 4.5 μM"

Line 593: Remove one "waters": "... nutrient-enriched  subsurface waters..."

Line 617: Add a space after Gomes *et al.* (2014).

Line 630: Remove one "of": "... the capacity of the  endosymbionts..."

Line 685: Replace with: "... will  have implications for the cycling of nutrients and oxygen..."

Line 797: Change to:"... to a degree that  prevented denitrification..."

Lines 797: Start new sentence with "In": " In comparison to the..."

**Additional references:**

Altabet, M. A., & Bourbonnais, A. (2019). N-loss stoichiometry in a Peru ODZ eddy. *Journal of Marine Research*, *77*(2), 169-189.

Babbin, A. R., Keil, R. G., Devol, A. H., & Ward, B. B. (2014). Organic matter stoichiometry, flux, and oxygen control nitrogen loss in the ocean. *Science*, *344*(6182), 406-408.

Bourbonnais, A., Altabet, M. A., Charoenpong, C. N., Larkum, J., Hu, H., Bange, H. W., & Stramma, L. (2015). N-loss isotope effects in the Peru oxygen minimum zone studied using a mesoscale eddy as a natural tracer experiment. *Global Biogeochemical Cycles*, *29*(6), 793-811.

Eugster, O., & Gruber, N. (2012). A probabilistic estimate of global marine N-fixation and denitrification. *Global Biogeochemical Cycles*, *26*(4).

Fassbender, A. J., Bourbonnais, A., Clayton, S., Gaube, P., Omand, M., Franks, P. J. S., ... & McGillicuddy Jr, D. (2018). Interpreting mosaics of ocean biogeochemistry. *Eos*, *99*(10.1029).

Noffke, A., Hensen, C., Sommer, S., Scholz, F., Bohlen, L., Mosch, T., ... & Wallmann, K. (2012). Benthic iron and phosphorus fluxes across the Peruvian oxygen minimum zone. *Limnology and Oceanography*, *57*(3), 851-867.

Vaquer-Sunyer, R., & Duarte, C. M. (2008). Thresholds of hypoxia for marine biodiversity. *Proceedings of the National Academy of Sciences*, *105*(40), 15452-15457.

---

## Referee Comment (RC3) · Anonymous Referee #3 · 10 Jun 2020

Review of bg-2020-82 - Present past and future of the OMZ in the northern Indian Ocean

This review brings a timely update on the state of the knowledge on the OMZ in the Indian Ocean. The value of this review is to bring together a wide range of disciplines covering the influence of bio-physical coupling, insights from paleo-oceanography, pelagic and benthic ecosystems, leveraging present and paleo observations, and models (from early Holocene to future). This is a very valuable exercise for the community. The authors do need to address several issues before it is acceptable for publication.

Specifically, the authors need to clarify how they discuss the balance between biology and circulation throughout the text. The claim that large-scale circulation control long-term changes in the OMZ in the Arabian Sea rather than local changes in biological

demand is made several times in the paper (see comment #6, 11, 14 and 21). The authors show data and model suggesting a decline in oxygen during the Holocene. However, the claim that it is due to large scale circulation is a hypothesis. The authors do not show the contribution from physical and biological controls in the model. Options to address this include: i) show the simulated integrated biological production and/or export production (it is usually an output in models), ventilation age would be tremendous but might not be available in the model. If export and biological production do not change, this would substantiate the claim that ocean circulation is controlling the change; ii) use the results from Bopp et al 2017 which show ventilation changes in another model in the Indian Ocean (see comments # 14-15). In any case, the language must be changed throughout the text.

There are some misleading points that need to be addressed in the abstract and introduction. I also strongly encourage the authors to strengthen sections 3, 4 and 6 (see comments #4-11, 16, 17). Comments on section 2 and 7 are mostly on the form. Finally, please read the paper carefully and double check grammar and spelling. Introduction, Section 3 and conclusion need special attention.

Detailed major comments

1. Abstract and introduction. - L18-19: "OMZ in AS and BoB intensified and expanded.". This is misleading the readers in the abstract. The main text suggests a much more subtle response with regions of expansion and regions of reduction (section 2). Please clarify. - L 3 and L30. Mentioning methane is very misleading as this would apply to terrestrial ecosystem but not so much to oceanic systems, which are the focus of this review. Please remove. You might consider mentioning N2O instead, which is number 3 in the list of GHG.

2. Section 2.3- This section on trends in BoB and AS is key to the review and the community. It would benefit some streamlining, specifically rephrase and make clear when the text refers to observed trends vs. when it discusses implications and more

general concepts of these trends (e.g. threshold for nitrite oxidation etc.).

3. In Section 3.1, the text reads: "data . . . implies that the respiration . . . causes the low oxygen concentrations in the Arabian Sea . . . satellite-derived export production rates were much too low to sustain such a high biological oxygen consumption . . . The mismatch between oxygen deficits and the biological consumption reflects uncertainties caused by the poorly constrained physical oxygen supply and export production rates."

This section should discuss model results mentioned later in the manuscript (e.g. Resplandy et al 2011, 2012; Lachkar et al.), which managed to maintain the OMZ in the Arabian Sea at a quasi steady-state over decadal time-scales. Looking at their balance between biological demand and physical supply would inform how this balance is achieved. Comparing these numbers to the estimates mentioned by the authors would bring valuable information on the "mismatch" and how models manage to achieve the balance (it does not mean the models are right but it is still valuable). Do the models simulate higher productivity that satellite based estimates? Do they have higher ventilation? These papers include information that can be used for this discussion (PP, oxygen physical supply, biological consumption etc.).

4. Section 3.1 discusses the Arabian Sea. What about the Bay of Bengal?

5. Sections 3.2. The points of this section are not well presented I was struggling to guess the links the authors want to make between seasonal thermocline, SNM to ballast effect, zooplankton migration etc.. Please streamline and clarify the following points.

- L253: "In contrast to the BoB, nitrite accumulates in the seasonal thermocline of the Arabian Sea". Link to rest of paragraph is unclear. Clarify the link with export production. Again this sentence probably belongs to the next paragraph. - L267-271: this is not comprehensible. Please clarify grammar and meaning. - L285-290: facts about the colocation of remineralization, zooplankton migration and upwelling source waters but unclear what the implications are. Please explain and clarify how this relates

to the main point here.

6. Section 3.3 L307-310: "suggesting that physical supply rather than biological demand are drivers controlling the intensity of the OMZ." This sentence needs clarifying. Without biological demand there is no OMZ, and the OMZ can be considered at "quasi steady-state" BIO + PHY $\sim$ 0 [of course there ae small trends but the OMZ has been relatively stable on decadal and century time-scales]. Maybe what the authors mean is that temporal variations in the intensity of the OMZ are controlled by physical supply? Clarify and specify what time-scale you are talking about (seasonal only? Decadal, centennial etc?).

7. Section 4. L312-323. Intro on eddies. You mention the role of eddies on biological production and oxygen mixing but could add the influence on export. The literature has progressed a lot since Oschlies et al 1998 and "eddy pumping" is not considered as the only mechanisms at work anymore. Relevant publications for biological production are reviews by McGillicuddy 2016 (mesoscale eddies) and Mahadevan 2016 (submesoscale, includes Arabian Sea example) and refs therein. For eddy-driven export production: Omand et al 2015 and Resplandy et al 2019 (eddy-driven export), Boyd et al 2019 (all export pathways including eddy-driven). On the role of eddies in oxygen mixing, I would also consider adding Bahl et al 2019.

8. Section 4. L330: " due to the semiannual reversal of the mean circulation and a resulting reduced oxygen supply". It is not clear how this fits in the sentence. Eddies enhance the oxygen supply to the OMZ, while the mean circulation partly offsets this supply by eddies. Please clarify text.

9. Section 4. L336: note that both the work of McCreary and Resplandy suggest that "this mechanism strongly contributes to the eastward shift...". As the authors pointed out earlier in this paragraph eddy-driven ventilation supplies oxygen to the western Arabian Sea in both studies. Please rephrase so it is clear that both studies converge here.

10. Section 4. L374-376: Does the Chen et al paper mentions a decline in eddy activity? This should be clarified. If it is interannual variability and not a long term decline, then you would expect interannual variability in the OMZ ventilation and denitrification but not necessarily a deoxygenation.

11. Section 4. L385-389. The links here are not clear. The bio/eddy-driven ventilation balance identified in present day models does not suggest that remotely forced changes in physical supply cause long-term changes. The supply of oxygen to the OMZ has to be through mixing and is promoted by eddy-driven circulation, because there are no direct advective pathways into the OMZ shadow zone (by definition). The authors are right however that large scale circulation is important because it regulates the oxygen gradients at the OMZ edges. However, I don't see why Holocene changes could not be tied to changes in biological demand? I would remove these sentences here and keep this discussion for the Holocene section 5 (see comment #14).

12. Section 5 L425-430 should point to Figure 6 to help reader follow. I suggest the authors slightly reorganize the text between L425 and 449. Starting with early Holocene before 6000 BP (move L433-439 up), then transition with the increased in productivity and enhanced OMZ after 6000BP (combine L425-432 and L439-450).

13. Section 5.2 L488-490: "a data-model comparison . . . in both basins". I thought the data-model comparison was only for the Arabian Sea. Please clarify. Note that adding an insert map of core location and model regions on figure 6 or would help the reader locate things. At least provide lon/lat of cores.

14. Section 5.2 L495-498: I am not sure I follow how the match between model and data in oxygen suggests that it is due to oceanic circulation rather than local biological processes. The authors state "it is assumed that .." in L490 but it seems neither the authors nor prior work has actually showed that circulation controls the simulate change in oxygen in this region in the model. This is an important point because that claim is repeated several times in the manuscript (see L385 and comment #11, L307

and comment #6 and conclusion). The author should either look at the biological and/or circulation changes in the model they present here or use models from others such as Bopp et al 2017 to make the claim (see comment #15 below)

15. Section 5. Please consider adding the study of Bopp et al 2017, which compares simulations at the LGM and mid-holocene, linking to the changes from Pleistocene to Holocene mentioned by the authors. The paper includes a qualitative comparison of simulated O2 with O2 proxies (Fig 3) and shows model ventilation changes between LGM and mid-holocene (Fig S3) in the Indian Ocean. Note that this model is not a transient run

16. Section 6. Authors should discuss their Figure 7 here. It is only mention in passing in L512. Something like "as shown in Figure 7....". Indeed, most prior work on ESM's OMZ was not targeting the Indian Ocean. Figure 7 would be a good opportunity to present specifically the results in the Indian Ocean.

17. Section 6. Authors should consider folding in this section the following recent papers looking at global OMZ and oxygen in ESMs. Models agree on the sign of warming-driven (O2sat) and biological-circulation (AOU) changes, but uncertainties arise from the subtle balance between these two opposing terms (Bopp et al 2017, Resplandy 2018). Papers highlighting the influence of circulation changes and non-resolved processes such as eddy-driven circulation and mixing (Duteil and Oschlies, 2011, Duteil et al 2014, Lachkar et al 2016, Palter and Trossman 2018, Fu et al 2018, Busecke et al 2019, Bahl et al 2019, Couespel et al 2020).

18. Section 7.1.1 The text is well written but it is much more detailed than the rest of the sections in the review. Authors might consider summarizing/emphasizing the take home messages, the links with oxygen and the OMZ and the implications for trophic webs which are quickly metion at the end of section L640. If there are there other groups than the co-authors that worked on this topic (I am not a specialist of this subtopic), it might be worth including some of their work here.

19. Section 7.1.2. The OMZ control migration but please also consider adding the fact that zooplankton vertical migration influence the oxygen consumption vertical patterns. This effect is missing from most ocean bio models and from all ESMs. The following studies are global but include maps showing the impact in the Indian Ocean. Bianchi et al. 2013 (their Fig 3) Aumont et al 2018 (their Fig 9) show simulated oxygen decline due to DVM. Note that most references in this section are relatively old. The authors could consider checking for newer results on this topic, maybe including references from other OMZs to fuel their discussion if not available in Indian Ocean.

20. Section 7.1.3 in implications discusses zooplankton but not the DVM aspects. It might be missing because part of the section is missing (see unfinished sentences L 682).

21. Section 8. The conclusion is vague and speculative. It tries to blend mesoscale eddies to paleo-changes but this is a difficult task (see my comments #6, 11 and 14). "This was caused by .. changes in circulation". Again this has not been demonstrated by the Authors (note that the paper by Bopp et al 2017 which shows ventilation changes between LGM and mid-holocene might help the authors to make the case).

22. Figure 4: specify O2 threshold used to compute OMZ thickness and how this "maximum thickness" is evaluated. Also briefly describe the data used here: How many cruises or from a database? What are the years during which these data were taken? Label seasons on plot (e.g. change symbols, add labels or colors).

23. Figure 5 could be improved so the difference between the two panels with/without eddies is more obvious and consistent with the text, ie.e eddies influence oxygen supply, nutrient supply and production. Oxygen does not seem to change between the two. Production and nutrient supply do not seem to change either.

24. Figure 6: Nice and interesting figure. - It would be great to add the WOA present day oxygen concentration at 0 ka on the plot to compare to the model results on panel b. - Please specify the model depth range for the oxygen values in caption. - Please

provide an insert map with core locations and model regions and/or provide lon/lat of cores and model regions in caption. Please remove "sinking" from caption, this is confusing (It sounds like subduction of oxygen).

Other comments:

L27: check grammar

L112-114: move to next section? Unclear why it is here.

L150-151: Add other refs about filaments and eddies.

L183: typo on conational?

L195: drops? dropped? Check tense (past/present) in section 2.3.

L202: "the in comparison". Remove the?

L203: "less intense . . . than"

L223: should "since than" read "since then"?

L226-227: add ref for the mixing analyses.

L233: (to 75%). Is this up to 75%?

L239: last sentence of section should be clarified and better linked to the rest of the paragraph. Why are they linked if the seasonal thermocline is hypoxic? Do you mean if the oxygen content of the seasonal thermocline remains stable through seasonal changes? This sentence probably belongs to the next section which defines the seasonal thermocline and make the link between bio production and physical supply of oxygen.

L240: Note that you define seasonal thermocline in L 241 but use it already in L239.

L246: "the season thermoclines" > seasonal thermocline?

L255: upper part of the thermocline? Upper thermocline?

[Figure]

L266: remove "upper part" and "lower part" as depth are specified.

L267: "the base of the SNM is located. . .. In contrast to the SNM. . ." the base of the SNM is in the SNM. This sentence doesn't make sense. Please rewrite.

L272: which suggests

L306 "preventing the development of anoxic conditions". As noted by the authors in section 2.3 anoxic conditions already occur in the northern IO. Clarify the sentence.

L341-342: please rephrase sentence. "This leads" what leads? Clarify the links between oxygen change, denitrification, nutrient supply, production and feedback on oxygen change.

L345: Thus eddies "would/could" affect. . ... this model results present a very interesting hypothesis but it does not make the link to fish habitat. At least modulate the link to fish except if you have a reference that makes this link in this region.

L351: "Using YY It could be shown that XX (Lachkar et al )" replace by Lachkar et al () showed that XX using YY

L360 and 363: remove "and hence weaken the OMZ" and "weakening the OMZ". Here the authors discuss the mean state of the OMZ not a tremd. As mentioned above Bio + Phy $\sim$ 0 in OMZs, hence eddy ventilation does not weaken the OMZ, it contributes to the supply of oxygen that balance the biological demand. Note that your section 4.3 discusses how variations in eddy activity could indeed result in variations in the oxygen supply and OMZ volume.

L364-365: link to denitrification – cite paper(s) showing that denitrification inhibition occurs here. L385. Not clear why there is a "However" to start the sentence here. Is this sentence incompatible with the previous one?

L439: could you please clarify the link between enhanced upwelling and ventilation by ICW? Is it through reduced residence time?

L448: "matches results from model. . ." I don't think this model has been presented yet. Please provide reference here or reference to section 5.3 which comes after. Also add reference to Figure 6b here.

L524 please clarify that the increase in hypoxic waters is global scale not in the Indian Ocean.

L582: does the journal authorize "in review" citations?

L682: missing text?

L715: define OM.

L796-797: check sentence and grammar.

L798: "The in comparison"?

References;

Aumont, O., Maury, O., Lefort, S., Bopp, L., 2018. Evaluating the Potential Impacts of the Diurnal Vertical Migration by Marine Organisms on Marine Biogeochemistry. Global Biogeochemical Cycles 32, 1622–1643. https://doi.org/10.1029/2018GB005886

Bahl, A., Gnanadesikan, A., Pradal, M.-A., 2019. Variations in Ocean Deoxygenation Across Earth System Models: Isolating the Role of Parameterized Lateral Mixing. Global Biogeochemical Cycles 33, 703–724. https://doi.org/10.1029/2018GB006121

Bianchi, D., Galbraith, E.D., Carozza, D.A., Mislan, K. a. S., Stock, C.A., 2013. Intensification of open-ocean oxygen depletion by vertically migrating animals. Nature Geosci 6, 545–548. https://doi.org/10.1038/ngeo1837

Bopp, L., Resplandy, L., Untersee, A., Mezo, P.L., Kageyama, M., 2017. Ocean (de)oxygenation from the Last Glacial Maximum to the twenty-first century: insights from Earth System models. Phil. Trans. R. Soc. A 375, 20160323. https://doi.org/10.1098/rsta.2016.0323

Boyd, P.W., Claustre, H., Levy, M., Siegel, D.A., Weber, T., 2019. Multi-faceted particle pumps drive carbon sequestration in the ocean. Nature 568, 327. https://doi.org/10.1038/s41586-019-1098-2 Busecke, J.J.M., Resplandy, L., Dunne, J.P., 2019. The Equatorial Undercurrent and the Oxygen Minimum Zone in the Pacific. Geophysical Research Letters 46, 6716–6725. https://doi.org/10.1029/2019GL082692

Couespel, D., Lévy, M., Bopp, L., 2019. Major Contribution of Reduced Upper Ocean Oxygen Mixing to Global Ocean Deoxygenation in an Earth System Model. Geophys. Res. Lett. 46, 12239–12249. https://doi.org/10.1029/2019GL084162

Duteil, O., Oschlies, A., 2011. Sensitivity of simulated extent and future evolution of marine suboxia to mixing intensity. Geophys. Res. Lett. 38, L06607. https://doi.org/10.1029/2011GL046877

Duteil, O., Böning, C.W., Oschlies, A., 2014. Variability in subtropical-tropical cells drives oxygen levels in the tropical Pacific Ocean. Geophys. Res. Lett. 41, 2014GL061774.

Lachkar, Z., Smith, S., Lévy, M., Pauluis, O., 2016. Eddies reduce denitrification and compress habitats in the Arabian Sea. Geophysical Research Letters 43, 9148–9156.

McGillicuddy, D.J., 2016. Mechanisms of Physical-Biological-Biogeochemical Interaction at the Oceanic Mesoscale. Annual Review of Marine Science 8, 125–159. https://doi.org/10.1146/annurev-marine-010814-015606

Mahadevan, A., 2016. The Impact of Submesoscale Physics on Primary Productivity of Plankton. Annual Review of Marine Science 8, 161–184. https://doi.org/10.1146/annurev-marine-010814-015912 Omand, M.M., D'Asaro, E.A., Lee, C.M., Perry, M.J., Briggs, N., Cetinić, I., Mahadevan, A., 2015. Eddy-driven subduction exports particulate organic carbon from the spring bloom. Science 348, 222–225.

Palter Jaime B., Trossman David S., 2018. The Sensitivity of Future Ocean

Oxygen to Changes in Ocean Circulation. Global Biogeochemical Cycles 0. https://doi.org/10.1002/2017GB005777 Resplandy, L., 2018. Will ocean zones with low oxygen levels expand or shrink? Nature 557, 314–315.

Resplandy, L., Lévy, M., McGillicuddy, D.J., 2019. Effects of Eddy‐Driven Subduction on Ocean Biological Carbon Pump. Global Biogeochem. Cycles 2018GB006125. https://doi.org/10.1029/2018GB006125

---

## Author Comment (AC1) · 26 Jun 2020

The authors would like to thank anonymous reviewer #1 for the constructive and valuable comments, which will help to improve the manuscript. A point-by-point reply to the comments follows below. The author responses are marked in red.

Review of Rixen et al 'Present, past and future of the OMZ in the northern Indian Ocean'

Rixen et al provide a review on the comparison of the two oxygen minimum zones in the northern Indian Ocean, located in the Bay of Bengal (BoB) and the Arabian Sea (AS). The two basins are compared from an oceanographic and biogeochemical point of view. This is obviously a challenge and I acknowledge that it is never an easy task to synthesize results from different disciplines and authors into a coherent piece of writing. To me the manuscript is a valuable contribution, however, it needs some more streamlining and integration of the different sections.

That said, I have some very general remarks, which I hope will help to streamline the manuscript:

I am not sure what is meant by hypoxic, in order to be able to stick with it may be helpful to define a range of oxygen concentrations you are referring to.

Upper threshold concentrations of dissolved oxygen, which are used to define hypoxia vary. In fisheries and in ecology upper threshold concentrations of 60 – 63 µM are well accepted (Ekau et al., 2010; Vaquer-Sunyer et al., 2008). However, from a biogeochemical point of view, 20 µM appears to be a more suitable threshold concentration as it marks the concentration below which fixed nitrogen is transformed into N2. Furthermore, it was used to map the volume of OMZ in ocean (Acharya et al., 2016). Therefore, we considered 20 µM as the upper threshold of hypoxia and anoxia as the lower threshold of hypoxia. Because oxygen detection limits of classical Winkler titration (~1 µM), seabird sensors (0.09 µM) and the newly developed switchable trace oxygen sensors (STOX, 0.01 µM) is too high to prove anoxia (Thamdrup et al., 2012; Ulloa et al., 2012) the occurrence of hydrogen sulfide has been considered as an indicator of anoxia. This will be specified in the revised version.

I am not quite sure what the aim of the study is, and what it specifically contributes as a stand-alone publication. From the title, I would expect to learn about potential expansion and intensification patterns of the two OMZs based on an assessment of past developments on geological timescales. This can, to a certain extent, be distilled out of the paper but I could imagine that if the authors take some effort and work through the paper once more, it would be more obvious.

We agree that a restructuring and unified writing style would improve the flow and make our view and arguments more obvious. The referees provided extremely constructive and detailed suggestions of how to restructure the manuscript and the native speakers among the authors will unify the writing style.

I also expected to learn about why those two basins behave so differently- there are different reasons given, including stratification, which is credible for a certain part of the BoB but not visible anymore in the offshore OMZ as presented by Bristow, further a ballasting effect by riverine particles present in the BoB and absent in the AS. The latter doesn't convince me, because it seems to be a coastal phenomenon only.

Since in comparison to the Arabian Sea, the Bay of Bengal is poorly studied it is difficult to compare these two basins process by process. However, we agree that a restructuring and a unified writing style would make our current understanding of mechanisms causing the differences between these two basins much clearer.

Bristow et al. (2017) presented data from seven stations in northern Bay of Bengal, which were obtained from one cruise carried out in January 2014. We doubt that this data suffice to prove that the offshore OMZ in the Bay of Bengal is no more stratified. Furthermore, the oxygen profiles presented in Figures S1 by Bristow et al. (2017) show to our understanding a clear stratification within the upper water column. This is less evident from sigma-t due to scale used, which ranged from 0 and 40 while within the regarded T/S range sigma-t varies approximately between 21 and 26 (see e.g. Schott et al. 2001).

We also do not agree that the ballasting effect by riverine particles is only a coastal phenomenon in the Bay of Bengal. The lithogenic ballast effect operates everywhere where lithogenic matter is incorporated into sinking particles. It is stronger in the Bay of Bengal than in the Arabian Sea beyond shelves and slopes as indicated by a higher contribution of lithogenic matter to the total flux in the Bay of Bengal. This was shown by sediment trap studies carried in the Bay of Bengal and the Arabian Sea. Rixen et al. (2019) provides more detailed information on these sediment trap studies.

Regarding the different oxygen concentrations in the two basins, a steady state between physical oxygen supply and biological oxygen consumption is also given. The slightly higher oxygen concentrations in the BoB is suggested to promote a feedback between nitrate reduction and nitrite oxidation. This is based on Bristow et al mainly, which

is one study with 5 stations during one time of the year. In order to strengthen your case, it may be beneficial to also consider Canfield et al. (2019) and Löscher et al. (2020), both of which propose alternative feedbacks possibly stabilizing the BoB's remaining oxygen traces. I understand that those studies may have come out after the presented paper was submitted and may have not been visible enough.

Other results on OMZ oxygen production as suggested for other regions (Garcia-Robledo et al., 2017) may be worthwhile considering given the abundance of small unicellular cyanobacteria as described for the BoB. As for the assessment of deep time changes, reference to the work of Orsi et al. (2017) could be helpful.

Thank you very much for providing these references which will complement our review.

In addition, but this indeed may go beyond the scope of the manuscript, a discussion on possibly changing monsoon intensities and atmospheric dust inputs could be interesting for a future assessment.

This is a very interesting aspect but we agree with the reviewer that these topics are beyond the scope of our current work.

Right now, it is obvious that the sections have been written by different authors, with sections 1-3 needing a native speaker to improve the language. I understand that the first author coordinated the writing and I know that this is an ungrateful job. However, there needs to be some more coherence regarding the writing style, the level on which the different topics are presented, and again, some more integration of the different sections to improve the reading flow.

As mentioned before we agree with the reviewer and the native speakers among the authors will unify the writing style.

I also have some specific comments and suggestions:

In the following we respond to content issues and with 'ok' to comments and suggestions regarding grammar and spelling.

Title: there is a comma missing between present and past. Ok

The abstract will be rewritten

l. 2 'is' should be changed to 'are'; 'it favors' should be changed to 'they favor'. I also do not quite understand the use of the (admittedly modern) expression 'ecosystem services'

l. 4: change 'which' to 'and its'

l. 8/10 past tense is used- is this because it refers to the geological past?

Introduction

l. 25: 600 mio years is a bit short. Canfield, Lenton and Lyons give different ranges, but they are about 2.3-3.2 billion of years for the rise of oxygen.

The sentence refers to the rise of oxygen to the nearly present day level and the occurrence of algae and planktonic cyanobacteria. This will be clarified.

27 ff I am not sure what this means? Are you suggesting those are the only habitats of anaerobic organisms? Because they are quite abundant throughout the marine water column on particles see e.g. (Ganesh et al., 2015;Ganesh et al., 2014). In addition, nitrate reduction to N2 can happen via anammox- in this case one could more or less claim those are anaerobic microbes. Denitrifiers are not anaerobic microbes, they are facultative and respire oxygen when possible. This part will be deleted.

l. 39 'of' is missing before 'oxygen' Ok

l. 49 'expense' Ok

l. 62 Here, a definition of hypoxic and anoxic would be helpful. Also, this way to abbreviate looks very awkward. Change to 'inhibit', 'prevent' Ok – regarding the definition, please see the authors comment above.

l. 66 'of' before 'anaerobic' is missing Ok

l. 70 a reference to work by Schmidtko et al. (2017) and Keeling et al. (2010)is missing

Schmidtko et al. (2017) and Keeling et al. 2009 will be added (we assumed that 2009 instead of 2010 was meant)

l. 75 'margins' Ok
l. 77 Again, this needs a definition of hypoxia. see above

l. 85 ff a reference to Naqvi et al. (2010) is missing.
Line 845 ff describes sedimentary denitrification rates. Naqvi et al. 2010 is about iron-limitation and only mentioned that decreasing nitrate and nitrite concentrations indicated an intense denitrification.
However, this aspect will be mentioned in the revised ms.

l. 166 How do the different primary producer communities look? How is the food web- wouldn't this also be important to make claims about export fluxes?
The impact of ballast minerals on the export is well-studied whereas the influences of primary producers and food-web structures on the carbon export is not well-constrained. We are also not aware of any work describing the impact of primary producers on the carbon export in the Indian Ocean. On the other hand, there are papers quantifying the impact of ballast minerals on carbon export and we summarized their results (see the authors comment above).

Also, if we have a faster export, would the a more anoxic sediment or deeper water layer be expected?
In principle we agree but this depends on many more factors such as bottom water oxygen concentrations and the composition of sediments. The Bay of Bengal is a deep-sea fan where in addition to vertical, a lateral supply via deep-sea channels also matters (e.g. Galy et al.2007). However, organic carbon accumulation in the Bengal fan is high which supports the assumption that the ballast effect increases the export of organic matter and its preservation in sediments.

Galy, V., France-Lanord, C., Beyssac, O., Faure, P., Kudrass, H., Palhol, F., 2007. Efficient organic carbon burial in the Bengal fan sustained by the Himalayan erosional system. *Nature*, 450, 407-410.

This statement is also somewhat contradictory to the claim made based on Bristow et al, that a microbial feed back stabilizes the trace oxygen concentrations.
To our understanding Bristow et al. (2017) suggested first of all a microbial feedback that operates at low oxygen concentrations and prevents nitrite reduction from becoming significant.  However, this will be clarified.

I would suggest mentioning the reasons for the difference in OMZ intensity in a way that is less exclusive and so that they can complement each other. The way it is, it is confusing.
We will add an additional paragraph on steady states in a previous chapter, which includes the various processes controlling oxygen concentrations in the water column.

l. 174 this needs a reference ok
l. 174 ff The statement is unclear, I think you are talking about a sulfidic event when saying anoxia,
Ivanenkov , V.N., Rozanov, A.G., 1961  report the occurrence of H2S in the NE Arabian Sea within the secondary nitrite maximum. This implies that the H2S originated in the water column and was not produced in the underlying sediments. Since the occurrence of H2S indicates anoxia, we would say it was an anoxic event at which H2S was formed.

l. 176 if this is an 'only report' why do you have three references?
Because they all agree that it is the only report.

l. 178 'don't seem to evolve every year' Ok
l. 179 Who dies during those mass mortalities? Please replace ' in between' with 'occasionally'. How

confident are we that those mass mortalities do not result from trace metal contaminations from the land?
We went back to the original papers and found out that fish mass mortalities occurred in back waters only. Therefore we deleted the term 'mass mortalities'.

Nandan, S.B., Azis, P.K.A., 1995. Fish Mortality from Anoxia and Sulphide Pollutions. *Journal of Human Ecology*, 6, 97-104.

L 183 'also' could be removed, sounds awkward. Ok
l. 184 change 'were' to 'was' Ok
l. 187 Actually, Bristow shows microaerobic processes to occur Ok
l. 190 ff awkward sentence, please rephrase  Ok
l. 192 Please add an explanation what excess $N_2$ measurements are good for. I don't think a non-N cycle expert can possibly know that. This part will be deleted.
l. 199 'outcompetes' Ok
l. 202 remove 'the', change 'rate implies' to 'rates may explain' This part will be deleted.
l. 204 ff, l. 206ff Please rephrase- awkward sentences. Ok
l. 208 'Follow-up studies also reported' Ok
l. 212 ff Schunck et al didn't report on periodic outbreaks but on a one-time event, another report from the same region would be Callbeck et al. (2018), both references combined may give some hint for a regular occurrence. The term 'periodic' will be deleted.
l. 215 this may also just be a result of the monitoring program. If no one went there to measure one wouldn't find it either.  We agree!

l. 220 remove 'the' before 'biological' Ok
l. 221 'This approach is based on' Ok
l. 223 what does 'regarded' mean here? The water masses of interest? Ok
l. 232 'approximately' Ok

l. 238/ 239 I don't understand this statement. This statement will be deleted.

l. 258 'isotope ratio'  Ok
l. 261 ff it would be helpful to explain which values are typical for denitrification and other processes
We will add an additional figure to show this.

l. 262 change 'indicates to 'is located in' Ok
l. 268 ff this statement doesn't make sense to me, the reference is also maybe not ideal.
This will be changed.

l. 270 'SNM' Ok
l 272 'suggests'  Ok
l. 273 ff What is the purpose of this statement?
The statement will be deleted.

l. 279 'within'  Ok
l. 280 'key factor'  Ok

l. 315 This could benefit from a reference
Generally, I was missing references to work on eddies in Atlantic OMZ waters and their relevance for oxygen budgets and biogeochemistry (Fiedler et al., 2016;Karstensen et al., 2017;Schütte et al., 2016), especially in lines 366 ff.
These references will be included. We will also add the following two references to support the statement

in line #315 (McWilliams, J.C., 2008, *Ocean modeling in an eddying regime*, and 2) McGillicuddy Jr, D.J., 2016. *Mechanisms of physical-biological-biogeochemical interaction at the oceanic mesoscale*.)

l. 391 remove 'the' Ok
l. 392 what is meant by 'nitrogen'? $N_2$, organic or inorganic nitrogen species?
Organic nitrogen in sediments is meant.

l. 392-405 this section would benefit from an explanation of what those values mean. This will be included
l. 397 remove 'the' Ok
l. 407 the core has the lowest oxygen concentrations?
It is not the core but the water column above the site at which the core was taken. This will be clarified.

l. 419 this sentence seems to be missing something. This will be clarified.
l. 428 'an onset' Ok
l. 422 ff, this part would benefit from observations by Orsi et al. (2017) This reference will be included.
l. 429 ff I don't understand the purpose of this statement  This will be clarified.
l. 430, the abbreviation ICW is only explained in l. 439.  ICW was already defined in the chapter 'background'.
l. 433 what does BP stad for? (before present – will be clarified)
l. 436'surfac- derived oxygen-rich water' Ok
l. 448 Kiel Climate Model, introduce the abbreviation as you use it later on, also this needs a reference. This will be added.
l. 454 explain what PISCES stands for
PISCES stands for 'Pelagic Interactions Scheme for Carbon and Ecosystem Studies) and is a biogeochemical model which simulates the lower trophic levels of marine and the biogeochemical cycles of carbon and of the main nutrients

Aumont, O., Ethé, C., Tagliabue, A., Bopp, L., Gehlen, M., 2015. PISCES-v2: an ocean biogeochemical model for carbon and ecosystem studies. *Geosci. Model Dev.*, 8, 2465-2513.

l 485 'the' before 'late' Ok
l. 513 what does that mean that it is backward? Replace 'oxygen values' with 'oxygen concentrations or saturations' whatever is appropriate
We rewrote this sentence to make it clearer:
"In most ESMs the east – west contrast between the Arabian Sea and Bay of Bengal is opposing to what observations show, with most global models producing lower oxygen concentrations in the Bay of Bengal than in the Arabian Sea."

l. 516 there are high resolution options including mesoscale dynamics in CMIP6
Analyzing the performance of the CMIP6 models in the northern Indian Ocean would exceed the scope of this paper. And as far as we know there hasn't been a publication on that so far that we could refer to. As we do not want to speculate here, we only could include the outlook that the future generation of ESMs is targeting that problem.

l. 519 Isn't it rather a general problem that there is no circulation model available?
Here we are not quite sure of what is meant: Do you mean that the large-scale circulation in the Indian Ocean is not well represented in the ESMs or that the parameterization of the mesoscale processes in the Indian Ocean is insufficient?

L 579 ff this section is lengthy and could lead better to the point
L 612 'Arabian Sea' ok
L 721 what is and 'edge effect'?

The "edge effect" concept is described previously, on L 702-705.

References

Callbeck, C. M., Lavik, G., Ferdelman, T. G., Fuchs, B., Gruber-Vodicka, H. R., Hach, P. F., Littmann, S., Schoffelen, N. J., Kalvelage, T., Thomsen, S., Schunck, H., Löscher, C. R., Schmitz, R. A., and Kuypers, M. M. M.: Oxygen minimum zone cryptic sulfur cycling sustained by offshore transport of key sulfur oxidizing bacteria, Nature Communications, 9, 1729, 10.1038/s41467-018-04041-x, 2018.

Canfield, D. E., Kraft, B., Löscher, C. R., Boyle, R. A., Thamdrup, B., and Stewart, F. J.: The regulation of oxygen to low concentrations in marine oxygen-minimum zones, Journal of Marine Research, 77, 297-324, 10.1357/002224019828410548, 2019.

Fiedler, B., Grundle, D. S., Schütte, F., Karstensen, J., Löscher, C. R., Hauss, H., Wagner, H., Loginova, A., Kiko, R., Silva, P., Tanhua, T., and Körtzinger, A.: Oxygen utilization and downward carbon flux in an oxygen-depleted eddy in the eastern tropical North Atlantic, Biogeosciences, 13, 5633–5647, 10.5194/bg- 13-5633-2016, 2016.

Ganesh, S., Parris, D. J., DeLong, E. F., and Stewart, F. J.: Metagenomic analysis of size-fractionated picoplankton in a marine oxygen minimum zone, ISME J, 8, 187-211, 2014.

Ganesh, S., Bristow, L. A., Larsen, M., Sarode, N., Thamdrup, B., and Stewart, F. J.: Size-fraction partitioning of community gene transcription and nitrogen metabolism in a marine oxygen minimum zone, ISME J, 1-15, doi:10.1038/ismej.2015.44, 2015.

Garcia-Robledo, E., Padilla, C. C., Aldunate, M., Stewart, F. J., Ulloa, O., Paulmier, A., Gregori, G., and Revsbech, N. P.: Cryptic oxygen cycling in anoxic marine zones, 114, 8319-8324, 10.1073/pnas.1619844114
%J Proceedings of the National Academy of Sciences, 2017.

Karstensen, J., Schütte, F., Pietri, A., Krahmann, G., Fiedler, B., Grundle, D., Hauss, H., Körtzinger, A., Löscher, C. R., Testor, P., Vieira, N., and Visbeck, M.: Upwelling and isolation in oxygen-depleted anticyclonic modewater eddies and implications for nitrate cycling, Biogeosciences, 14, 2167–2181, 10.5194/bg-14-2167-2017, 2017.

Keeling, R. F., Kortzinger, A., and Gruber, N.: Ocean Deoxygenation in a Warming World, in: Annual Review of Marine Science, Annual Review of Marine Science, Annual Reviews, Palo Alto, 199-229, 2010.

Löscher, C. R., Mohr, W., Bange, H. W., and Canfield, D. E.: No nitrogen fixation in the Bay of Bengal?, Biogeosciences 17, 851–864, 10.5194/bg-17-851-2020, 2020.

Naqvi, S. W. A., Naik, H., D'Souza, W., Narvekar, P. V., Paropkari, A. L., and Bange, H. W.: Carbon and nitrogen fluxes in the North Indian Ocean, in: Carbon and nutrient fluxes in continental margins: A global synthesis, edited by: Liu, K.-K., Atkinson, L., Quiñones, R., and Talaue-McManus, L., Springer-Verlag, New York, 180-191, 2010.

Orsi, W. D., Coolen, M. J. L., Wuchter, C., He, L., More, K. D., Irigoien, X., Chust, G., Johnson, C., Hemingway,
J. D., Lee, M., Galy, V., and Giosan, L.: Climate oscillations reflected within the microbiome of Arabian Sea sediments, Scientific Reports, 7, 6040, 10.1038/s41598-017-05590-9, 2017.

Schmidtko, S., Stramma, L., and Visbeck, M.: Decline in global oceanic oxygen content during the past five decades, Nature, 542, 335-339, 10.1038/nature21399, 2017.

Schütte, F., Karstensen, J., Krahmann, G., Hauss, H., Fiedler, B., Brandt, P., Visbeck, M., and Körtzinger, A.: Characterization of "dead-zone" eddies in the eastern tropical North Atlantic, Biogeosciences, 13, 5865- 5881, 10.5194/bg-13-5865-2016, 2016.

Acharya, S.S., Panigrahi, M.K., 2016. Eastward shift and maintenance of Arabian Sea oxygen minimum zone: Understanding the paradox. *Deep Sea Research Part I: Oceanographic Research Papers*, 115, 240-252.
Ekau, W., Auel, H., Pörtner, H.O., Gilbert, D., 2010. Impacts of hypoxia on the structure and processes in pelagic communities (zooplankton, macro-invertebrates and fish). *Biogeosciences*, 7, 1669-1699.
Vaquer-Sunyer, R., Duarte, C.M., 2008. Thresholds of hypoxia for marine biodiversity. *Proceedings of the National Academy of Sciences*, 105, 15452.

---

## Author Comment (AC2) · 26 Jun 2020

The authors would like to thank anonymous reviewer #2 for the constructive and valuable comments, which will help to improve the manuscript. A point-by-point reply to the comments follows below. The author responses are marked in red.

Review of Rixen et al.

**General comment**

The manuscript by Rixen et al. gives a good overview of the development of OMZs and recent trends in the Arabian Sea and the Bay of Bengal, and discusses impacts from ocean circulation, export production and mesoscale eddies.

The authors would like to thank anonymous reviewer #2 for this comment

The paper also reviews past and potential future OMZ strength as inferred from the sediment $d^{15}N$ and model predictions and looks at the pelagic and benthic ecosystem responses.

The paper is generally sound and has the potential to present a much needed comprehensive review of the Arabian Sea OMZs. It is however, quite apparent that the different sections were written by different authors as the writing style and the transitions from one section to another are incongruent. The manuscript should be at least partially rewritten to improve the flow. Some sections clearly need editing by an native English speaker (in particular the abstract, introduction, and conclusion). Some sentences are unclear or repetitive. I found several typos through the manuscript. See my technical corrections below for some suggestions on how to improve these sections.

We agree that a restructuring and unified writing style would improve the flow and make our view and arguments more obvious. The referees provided extremely constructive and detail suggestions of how to restructure the manuscript and the native speakers among the authors will unify the writing style.

On another note, the adopted $O_2$ thresholds defining hypoxia and anoxia are confusing. Several papers use a much higher $O_2$ threshold to define hypoxia (e.g., >63 µM, Vaquer-Sunyer and Duarte, 2008 and reference therein). The presence of $H_2S$ should rather be referred to as sulfidic conditions, as sulfate reduction does not necessarily occur under anoxic conditions. A clear distinction between Oxygen Minimum Zones (OMZs) and Oxygen Deficient Zones (ODZs) should also be made.

We are aware that in fisheries and in ecology an upper threshold concentration of 60 – 63 µM for hypoxia is well accepted (Ekau et al., 2010; Vaquer-Sunyer et al., 2008). However, from a biogeochemical point of view a 20 µM threshold concentration appears to be more suitable as it marks the concentration below which fixed nitrogen is transformed into $N_2$. Furthermore, it was used to map the volume of OMZ in the ocean (Acharya et al., 2016). Therefore, we considered 20 µM as upper threshold and anoxia as the lower threshold of hypoxia (zero oxygen). Because oxygen detection limits of classical Winkler titration (~1 µM), seabird sensors (0.09 µM) and the newly developed switchable trace oxygen sensors (STOX, 0.01 µM) is too high to prove anoxia (Thamdrup et al., 2012; Ulloa et al., 2012) the occurrence of hydrogen sulfide was considered as an indicator of anoxia. According to our understanding sulfate reduction is a strict anoxic process. However, we would be very grateful if the reviwer#2 could provide us a reference so that we can integrate this aspect.

Finally, some figures should be added or improved for clarity. For instance, a figure explaining the development of an OMZ in relation to ocean circulation and seasonal monsoons would be helpful (Background, section 2.2).

This will be done.

**Specific comments:**

In the following we respond to content issues and with short statements to comments and suggestions

regarding grammar, spelling, and unclear explanations.

**Abstract:** The abstract will be rewritten.

Overall, the whole abstract ought to be rewritten to summarize the main points of the manuscript. The current version is confusing, and at times vague. Also, the abstract should follow a more logical order following the order of the different sections as presented in the text.

Line 4: Nitrate loss is only a problem if it is limiting (i.e., in a non-eutrophic system).

Lines 14-16: This sentence is confusing and needs clarification. It should be rephrased to emphasize that, based on previous studies (e.g., Aumont *et al.*, 2015), decreasing oxygen concentration slows down respiration and thus decreases oxygen demand. The following sentence is also unclear as it is.

Lines 19-21: This sentence is too vague. Effects on benthic and pelagic ecosystems should be better summarized.

**Introduction**

Lines 27-30: This sentence is ambiguously worded.
This will be clarified

Lines 30-32: $N_2O$ is also produced as an intermediate during denitrification and is a by-product of nitrification. $N_2O$ is a greenhouse gas 300 times more potent than $CO_2$ and an ozone destructing substance and should also be included here.
This will be changed.

Lines 46-48: The availability and quality (organic matter stoichiometry) of organic material is a key control on denitrification versus anammox (Babbin *et al.*, 2014).
The reference will be included.

Lines 60-61: Their definitiona of hypoxia and anoxia are a bit confusing since most studies define hypoxia at $O_2$ concentrations >63 µM (Vaquer-Sunyer and Duarte, 2008 and reference therein). Anoxia ($O_2$ concentrations close to zero or in the nmol range) can also occur without hydrogen sulfide production.
See authors comment above

Lines 66-67: I don't quite understand this sentence. Marine ecosystem services need to be defined earlier in the text. I suggest removing this sentence as the next sentence (lines 67-70) articulates the same idea better.
We will rephrase this sentence.

Lines 80-82: The more recent estimates by Eugster and Gruber (2014) of 52 Tg N $yr^{-1}$ for water column denitrification and 93 Tg N $yr^{-1}$ for benthic denitrification should be referenced. The distinction between water column and benthic rates should be made more explicitly.
Lines 84-85: Considering a mean sedimentary denitrification rate by Eugster and Gruber (2014) of 93 Tg N $yr^{-1}$, the proportion of sedimentary denitrification at the Pakistan continental margin could be even higher.
This reference will be included.

**Main text**

Lines 124-151: A figure showing the impact of ocean circulation in relation to seasonal monsoon on OMZ expansion in the eastern and western Arabian Sea would be helpful.

Lines 144-146: Is this low areal extension associated with increased thickness of the ODZ, as shown in Figure 4? This should be clarified here.
In summer the low areal extension is associated with deepening of the OMZ. This is caused by the enhanced carbon export and reflects also the deepening the mixed layer due to downwelling, which occurs in the central Arabian Sea during the summer. This will be clarified.

Lines 166-168: Why this ballast-effect mostly occurring in the Bay of Bengal and not the Arabian Sea?
It operates in both basins but since the lithogenic matter content in sinking particles is higher in the Bay of Bengal than in the Arabian Sea, the ballast effect is stronger in the Bay of Bengal. We will change 'The ballast-effect' into 'A stronger ballast effect'.

Lines 180-182: A distinction should be made between human-induced coastal eutrophication and coastal dead zone development due to the imbalance between higher $O_2$ consumption from primary productivity (upwelling) relative to $O_2$ supply from physical circulation.
This will be change to:
The spatial expansion of hypoxia, which seems to be an increasingly common feature in coastal waters, is called the "spreading of dead zones" (Altieri et al., 2017; Diaz et al., 2008) which expresses the threat of oxygen-depletion to oxygen-breathing organism, and more specifically to fisheries. This global phenomenon is a consequence of eutrophication and global warming whereas eutrophication increases oxygen consumption by enhancing the production of organic matter and global warming decreases the oxygen supply due to a reduced solubility of oxygen in warmer waters.

Lines 198-199: A reference is needed to support this $O_2$ threshold.
We will delete this part of the text.

Lines 191-193: How does the relatively low denitrification rate estimated by Bristow *et al.* (2017) compares to the denitrification rate (including anammox) measured in the Arabian Sea using [15]N-labeled incubations by Ward *et al.* (2009)?
Bristow *et al.* (2017) measured a rate of 0.9 nmol $L^{-1}$ $day^{-1}$ at one site while Ward et al. (2009) measured rates of > 20 nmol $L^{-1}$ $day^{-1}$ (denitrification) and up to approximately 5 nmol $L^{-1}$ $day^{-1}$ (anammox). These numbers will be included into the ms.

Lines 215-217: This is an important point that should be described better in the abstract.  Ok

Line 229: The term (central Indian Ocean) is already defined in the previous section. Ok

Lines 267-268: I don't quite understand this sentence either. Do they mean in contrast to the upper part of the SNM?
This sentence will be rephrased.

Lines 277-279: How does figure 3 support this point?
It supports this point in so far as it illustrates fluxes of free and protected organic carbon as calculated according the equations introduced by Armstrong et al. 2002 and our sediment trap data. However, this will be clarified.

Lines 319-323: The roles of coastal mode water anticyclonic eddies as N-loss hotspot in the Peru upwelling system should also be referenced (Bourbonnais *et al.*, 2015; Altabet and Bourbonnais, 2019). The paper by Fassbender *et al.* (2018) also provides a good review of the effects of mesoscale and submesoscale features on ocean biogeochemistry.

These references will be included.

Lines 339-342: On which timescale are these feedbacks expected to occur?

These feedbacks occur on a relatively short timescale (i.e., years) as the reduction in the suboxic core volume and denitrification happens mostly in the upper 200-400m.

Lines 379-381: These two terms "eddy-driven isopycnal tracer mixing" and "isopycnal flattening" need to be explained.

The isopycnal tracer mixing refers to the mixing of tracers by eddies along isopycnal (or neutral) surfaces. This is the diffusive part of eddy mixing that is typically parameterized in coarse-resolution models following the Redi scheme (Redi, 1982). On the other hand, the isopycnal flattening refers to the advective effect of eddies that acts to adiabatically flatten the slope of isopycnal surfaces. The latter is typically represented in models following Gent and McWiliiams parameterization (1990). The explanation of the difference between the two components will be added in the revised manuscript.

Lines 419-420: The authors should be more specific about which results they are referring to (Bristow *et al.*, 2017).

Ok , this will be clarified.

Line 471: A reference is needed to support this $O_2$ threshold for denitrification. Dalsgaard *et al.* (2014) report an $O_2$ threshold in the nmol range for denitrification.

The mentioned $O_2$ threshold for denitrification refers simply to the value used in the PISCES model (erroneously put at 5 rather than 6 µM – this will be corrected). However, a discussion about thresholds will be included into the background section. It will be pointed out that denitrification sets in at oxygen concentrations > 6 µM but becomes significant for the N-cycle only at oxygen concentrations of about 0.05 µM. A discussion of how models deal with these thresholds will be included.

Lines 486-488: The model's results do not seems to support denitrification during the Holocene.

As stated in lines 467-469 (see also Segschneider et al. 2018 in references) the Kiel Climate Model/PISCES, as all GCMs, simulates higher oxygen concentrations than observed at present in the Arabian Sea. Despite this difference in total oxygen concentrations, the Holocene trends (oxygen decreases) and the spatial differences can be reproduced by KCM/PISCES. We therefore surmise, that the model results can give us hints at the driving mechanism of oxygen decline (increasing age of OMZ water mass).

Lines 497-498: Was a relationship between orbital forcing (i.e., Milankovitch cycles) and the development of the OMZ ever investigated in the region? A reference should be added.

Bopp et al. 2017 investigated this for LGM – 6k time slice experiments (relation to our experiment discussed in Segschneider et al., 2018), but to our knowledge there are no further transient experiments but ours that are investigating this over the Holocene for the Indian Ocean. We will add Bopp et al. 2017 to the references.

Lines 515-519: Submesoscale processes, which are ephemeral and take place over lengths of about 1-10 km lasting several days, are also poorly represented (see Fassbender *et al.*, 2018).

This is true. Processes that are even smaller and more short-lived than mesoscale processes are also poorly represented in such coarse resolution models. We will address them too, as they are connected to

eddy transport and play a crucial role in distributing nutrients in frontal zones.

Lines 600-616: What is the effect of these larges blooms on OMZ expansion?
This is an interesting aspect, which will be integrated into the discussion but feedbacks of these blooms on declining oxygen concentrations have so far not been studied.

Lines 702-705 and 715-720: At which oxygen thresholds are these community composition and faunal abundance changes observed?
Clear changes in community composition and abundance have been reported across oxygen gradients both on OMZ margins (in the Indian Ocean and elsewhere, e.g., Gooday et al, 2009; Levin et al, 1991, 2000, 2009a,b), as well as in hypoxic basins such as the Baltic Sea, and at sites impacted by excess organic matter input (e.g. Rosenberg, 2001). Systematic changes occur in the lifestyles and feeding modes of the benthos across these gradients, sometimes showing depth ranges/areas with exceptional abundances of as few as one species (e.g., Jeffreys et al, 2012).  This has been attributed to the combined effects of oxygen (differing tolerance to slight changes in low oxygen levels) food availability, and predation avoidance (e.g. Levin et al). Consequently, and because of common lack of *in situ* bottom-water oxygen measurement, or differences in the sensitivity/precision of oxygen measurement across studies, specific oxygen thresholds have been difficult to constrain. Further, while similar general patterns are observed (e.g., on different margins of the Arabian Sea), local differences occur in the individual species that show abundance peaks across oxygen gradients. However, the threshold observed for a change from foraminifera to metazoan macrofauna in the uptake and processing of organic matter, based on isotope tracer studies across the Pakistan margin of Arabian Sea, occurred at an oxygen concentration of 5-7 $\mu$M (Woulds et al 2007).

Rosenberg, R (2001) Marine benthic faunal successional stages and related sediment activity. *Scientia Marina*, 66, 107-119. Other cited references are in the manuscript bibliography.

Lines 745-748: What is the N:P ratio in the overlying ODZ versus the sediments? Lower N:P ratios than expected based on $NO_3^-$ loss and biogenic $N_2$ production during denitrification are often observed in coastal ODZs due to the preferential release of $PO_4^{3-}$ following iron and manganese oxyhydroxide dissolution in anoxic sediments (Noffke *et al.*, 2012).
It is not clear what information is being sought, or how it would be relevant to the statement made in the text (which refers to the N:P stoichiometry of benthic nutrient fluxes). The N:P ratios in these fluxes differ due to the net effects of multiple benthic processes occurring across the cross-margin gradients in bottom-water redox conditions (as per the referee's comment).  To what ODZ sample type, and from what depth, would one compare sediment N:P ratios (in solids or porewaters?); water or suspended or sinking particles? While water-column nutrient analyses have been commonplace in both the Arabian Sea and Bay of Bengal, this is less true for particulates. There have been various sediment trap deployment programmes in the Arabian Sea over the decades, and also some in the Bay of Bengal, and some of these have included deployments within the OMZ. However, these have generally been at different (single) OMZ depths. There has been even less systematic sampling of suspended particles, and neither sample type has had routine determinations of both N and P.

Line 750: Define "dark" carbon.

Dark carbon refers to the carbon fixed in aphotic zones by autotrophic organisms. This will be defined in the manuscript.

Lines 797-800: Higher oxygen concentrations are more likely the results of the development of a sharper pycnocline (from higher freshwater fluxes) and lower primary productivity in the Bay of Bengal.

**Conclusion:** The conclusion will be rewritten

Lines 800-802: This sentence is unclear. Do they mean that mesoscale eddies sustain higher $O_2$ concentrations in the OMZ than expected in their absence?

**Figures:** AOU should be showed instead of Δoxygen ($\propto$mol kg$^{-1}$) in this figure since this is what is discussed in the text. Something must be wrong with the scale for Δoxygen ($\propto$mol kg$^{-1}$). The Δoxygen (deviation from $O_2$ concentrations at saturation) should be much higher than 10 $\propto$mol kg$^{-1}$ to cause hypoxic/anoxic conditions.
Why is the figure broken into two panels (a, b)? Another suggestion is the break the axis for depth >500 m.
Doxygen is not the AOU, it is the change of oxygen concentrations with depth. It will be removed from the figure and the axis will be broken.

Figure 4. It is unclear how to reconcile data in Figure 2 - showing that overall a decrease in the OMZ area seems to correspond to a decrease in the mean OMZ oxygen concentrations (at least during summer monsoon when POC flux is highest) and Figure 4 - showing a negative correlation between OMZ max thickness and the mean OMZ oxygen concentration.

The circulation mainly affects the areal extension while the carbon export into the deep sea and the oxygen concentration seem to exert the main influence on the thickness. Due to the different drivers the thickness does not correlate with areal extension. This will be clarified in the text and the figures will be changed accordingly.

Figure 5. This figure is difficult to read (white font on light blue background). Font size should be bigger. Isopycnal mixing by mesoscale eddies could be emphasized in a.
This will be changed

Figure 6. Make d as a symbol (y axis): "$\delta^{15}N$"
Figure will be changed.

**Technical corrections:**
The abstract will be rewritten

Line 4: Change "increases the loss nitrate" to "increases nitrate loss"

Lines 4-5: Change to "Nitrate is a macronutrient limiting primary productivity in most of the ocean."

Lines 7-10: This sentence seems to be out of context and repetitive considering the following sentence. I suggest rewriting:
"The main control on oxygen concentrations in the Arabian Sea and the Bay of Bengal is the balance between physical oxygen supply and biological oxygen consumption from respiration. Mesoscale eddies greatly enhance mixing and advection of $O_2$-rich waters, which compensate biological consumption and overall reduces ODZ expansion."

Lines 12-14: Change to: "However, due to slightly higher oxygen concentrations, aerobic nitrite oxidation outcompete anaerobic nitrite reduction and thus limits denitrification in the Bay of Bengal"

Line 39: Replace "At" with "Under" at the beginning of sentence. Ok

Lines 62 and 64 and 806: Change "hyp-" for hypoxic here and everywhere else in the text.Ok

Line 74: Remove "is": "..., with a much smaller proportion  located in the Bay of

Bengal..."Ok

Line 83: Replace "to this data" with "published data". Ok

Line 90: Change to: "one of the least understood

OMZs"Ok

Line 183: Replace "conational" for "continental" Ok

Line 190: Replace "nitrite oxidization" with "nitrite oxidation" Ok

Line 201: Replace "this is with about 0.7 $\propto$M much higher" by "it is about 0.7 $\propto$M higher". Ok

Line 202: Remove the at beginning of sentence: "However,  in comparison to the Arabian Sea..." and remove "as in the Arabian Sea" at the end of sentence. Ok

Lines 208-209: Replace with: "Subsequent studies also reported decreasing oxygen concentrations in the western and northern Arabian Sea." Ok

Line 255: Replace with: "in the upper part of the seasonal thermocline..." Ok

Line 258: Replace "stabile" with "stable" Ok

Line 270: Replace "SNN" with "SNM"

Ok

Lines 279-281: Therewith is used twice within the same sentence. Ok

Line 286: Remove "in": "... is mostly remineralized within  the upper 300 m..." Ok

Line 292: Remove "also" Ok

Line 297: Replace "the hypothesis" by "this hypothesis" Ok

Line 439: This term (ICW) is already defined earlier in the text.

Ok

Lines 534 and 539: Change for 80 $\propto$M $\underline{O_2}$ and 50 $\propto$M $\underline{O_2}$. Ok

Line 573: Replace for: "... can survive at $O_2$ concentrations down to 4.5 ∝M" Ok

Line 593: Remove one "waters": "... nutrient-enriched  subsurface waters..." Ok

Line 617: Add a space after Gomes *et al.* (2014).
Ok

Line 630: Remove one "of": "... the capacity of the  endosymbionts..."

Line 685: Replace with: "... will have implications for the cycling of nutrients and oxygen..."

Line 797: Change to:"... to a degree that  prevented denitrification..."

Lines 797: Start new sentence with "In": " In comparison to the..."

**Additional references:**

Altabet, M. A., & Bourbonnais, A. (2019). N-loss stoichiometry in a Peru ODZ eddy. *Journal of Marine Research*, *77*(2), 169-189.

Babbin, A. R., Keil, R. G., Devol, A. H., & Ward, B. B. (2014). Organic matter stoichiometry, flux, and oxygen control nitrogen loss in the ocean. *Science*, *344*(6182), 406-408.

Bourbonnais, A., Altabet, M. A., Charoenpong, C. N., Larkum, J., Hu, H., Bange, H. W., & Stramma, L. (2015). N-loss isotope effects in the Peru oxygen minimum zone studied using a mesoscale eddy as a natural tracer experiment. *Global Biogeochemical Cycles*, *29*(6), 793-811.

Eugster, O., & Gruber, N. (2012). A probabilistic estimate of global marine N-fixation and denitrification. *Global Biogeochemical Cycles*, *26*(4).

Fassbender, A. J., Bourbonnais, A., Clayton, S., Gaube, P., Omand, M., Franks, P. J. S., ... & McGillicuddy Jr, D. (2018). Interpreting mosaics of ocean biogeochemistry. *Eos*, *99*(10.1029).

Noffke, A., Hensen, C., Sommer, S., Scholz, F., Bohlen, L., Mosch, T., ... & Wallmann, K. (2012). Benthic iron and phosphorus fluxes across the Peruvian oxygen minimum zone. *Limnology and Oceanography*, *57*(3), 851-867.

Vaquer-Sunyer, R., & Duarte, C. M. (2008). Thresholds of hypoxia for marine biodiversity. *Proceedings of the National Academy of Sciences*, *105*(40), 15452-15457.

---

## Author Comment (AC4) · 26 Jun 2020

The authors would like to thank anonymous reviewer #3 for the constructive and valuable comments, which will help to improve the manuscript. A point-by-point reply to the comments follows below. The author responses are marked in red.

Review of bg-2020-82 - Present past and future of the OMZ in the northern
Indian Ocean

This review brings a timely update on the state of the knowledge on the OMZ in the Indian Ocean. The value of this review is to bring together a wide range of disciplines covering the influence of bio-physical coupling, insights from paleo-oceanography, pelagic and benthic ecosystems, leveraging present and paleo observations, and models (from early Holocene to future). This is a very valuable exercise for the community. The authors do need to address several issues before it is acceptable for publication.
The authors would like to thank anonymous reviewer #3 for this comment

Specifically, the authors need to clarify how they discuss the balance between biology and circulation throughout the text. The claim that large-scale circulation control long-term changes in the OMZ in the Arabian Sea rather than local changes in biological demand is made several times in the paper (see comment #6, 11, 14 and 21). The authors show data and model suggesting a decline in oxygen during the Holocene. However, the claim that it is due to large scale circulation is a hypothesis. The authors do not show the contribution from physical and biological controls in the model. Options to address this include: i) show the simulated integrated biological production and/or export production (it is usually an output in models), ventilation age would be tremendous but might not be available in the model.  If export and biological production do not change, this would substantiate the claim that ocean circulation is controlling the change; ii) use the results from Bopp et al 2017 which show ventilation changes in another model in the Indian Ocean (see comments # 14-15). In any case, the language must be changed throughout the text.
Simulated export production which are constant throughout the Holocene and water mass age in the Arabian Sea have been discussed for an earlier model experiment with the same model setup (but accelerated forcing) in Gaye et al. (2017) and in more detail for the global OMZs including the Indian Ocean for the model experiment analyzed here in Segschneider et al. 2018 (which also discusses the results of Bopp et al. 2017). We will point more prominently to the respective sections and figures in Gaye et al. 2017 and Segschneider et al. 2018.

There are some misleading points that need to be addressed in the abstract and introduction. I also strongly encourage the authors to strengthen sections 3, 4 and 6 (see comments #4-11, 16, 17). Comments on section 2 and 7 are mostly on the form. Finally, please read the paper carefully and double check grammar and spelling. Introduction, Section 3 and conclusion need special attention.
We agree that a restructuring and unified writing style would improve the flow and make our view and arguments more obvious. The referees provided extremely constructive and detailed suggestions of how to restructure the manuscript and the native speakers among the authors will unify the writing style.

**Detailed major comments**
In the following we respond to content issues and with short statements to comments and suggestions regarding grammar, spelling, and unclear explanations.

1. Abstract and introduction. Abstract will be rewritten
   - **L18-19**: "OMZ in AS and BoB intensified and expanded.". This is misleading the readers in the abstract. The main text suggests a much more subtle response with regions of expansion and regions of reduction (section 2). Please clarify.
   - **L 3 and L30**. Mentioning methane is very misleading as this would apply to terrestrial ecosystem but not so much to oceanic systems, which are the focus of this review. Please remove. You might consider mentioning N2O instead, which is number 3 in the list of GHG.

2. **Section 2.3-** This section on trends in BoB and AS is key to the review and the community. It would benefit some streamlining, specifically rephrase and make clear when the text refers to observed trends vs. when it discusses implications and more general concepts of these trends (e.g. threshold for nitrite oxidation etc.).
   This will be done.

3. **In Section 3.1, t**he text reads: "data … implies that the respiration … causes the low oxygen concentrations in the Arabian Sea … satellite-derived export production rates were much too low to sustain such a high biological oxygen consumption … The mismatch between oxygen deficits and the biological consumption reflects uncertainties caused by the poorly constrained physical oxygen supply and export production rates."
   This section will be restructured.

This section should discuss model results mentioned later in the manuscript (e.g. Resplandy et al 2011, 2012; Lachkar et al.), which managed to maintain the OMZ in the Arabian Sea at a quasi steady-state over decadal time-scales. Looking at their balance between biological demand and physical supply would inform how this balance is achieved. Comparing these numbers to the estimates mentioned by the authors would bring valuable information on the "mismatch" and how models manage to achieve the balance (it does not mean the models are right but it is still valuable). Do the models simulate higher productivity that satellite based estimates? Do they have higher ventilation? These papers include information that can be used for this discussion (PP, oxygen physical supply, biological consumption etc.).
This is a very interesting aspect, which will be included.

4. **Section 3.1** discusses the Arabian Sea. What about the Bay of Bengal?
   A discussion about the Bay of Bengal will be added.

5. **Sections 3.2.** The points of this section are not well presented I was struggling to guess the links the authors want to make between seasonal thermocline, SNM to ballast effect, zooplankton migration etc.. Please streamline and clarify the following points.
   This section will be restructured.

   - L253: "In contrast to the BoB, nitrite accumulates in the seasonal thermocline of the Arabian Sea". Link to rest of paragraph is unclear. Clarify the link with export production. Again this sentence probably belongs to the next paragraph.
   - L267-271: this is not comprehensible. Please clarify grammar and meaning.
   - L285-290: facts about the colocation of remineralization, zooplankton migration and upwelling source waters but unclear what the implications are. Please explain and clarify how this relates to the main point here.

6. **Section 3.3 L307-310**: "suggesting that physical supply rather than biological demand are drivers controlling the intensity of the OMZ." This sentence needs clarifying. Without biological demand there is no OMZ, and the OMZ can be considered at "quasi steady-state" BIO + PHY ~ 0 [of course there ae small trends but the OMZ has been relatively stable on decadal and century time-scales]. Maybe what the authors mean is that temporal variations in the intensity of the OMZ are controlled by physical supply? Clarify and specify what time-scale you are talking about (seasonal only? Decadal, centennial etc?).
   Yes, we will clarify this point.

7. **Section 4. L312-323. Intro on eddies.** You mention the role of eddies on biological production and oxygen mixing but could add the influence on export. The literature has progressed a lot since Oschlies et al 1998 and "eddy pumping" is not considered as the only mechanisms at work anymore. Relevant publications for biological production are reviews by McGillicuddy 2016 (mesoscale eddies) and Mahadevan 2016 (submesoscale, includes Arabian Sea example) and refs therein. For eddy-driven export production: Omand et al 2015 and Resplandy et al 2019 (eddy-driven export), Boyd et al 2019 (all export pathways including eddy-driven). On the role of eddies in oxygen mixing, I would also consider adding Bahl et al 2019.
   These references will be included into the discussion.

8. **Section 4. L330**: " due to the semiannual reversal of the mean circulation and a resulting reduced oxygen supply". It is not clear how this fits in the sentence. Eddies enhance the oxygen supply to the OMZ, while the mean circulation partly offsets this supply by eddies. Please clarify text.
   This statement refers to the fact that according to Resplandy et al (2012) the Ekman pumping contribution to O2 supply is nearly of comparable magnitude but of opposite sign between the summer and winter monsoon seasons, which hence leads to these effects compensating each other on an annual timescale, resulting in a weak contribution of the mean circulation to vertical oxygen supply, thus dominated by mesoscale eddies (on an annual timescale).

   For more clarity, we will change that statement to: "Furthermore, mesoscale eddies and filaments were shown to dominate, on an annual timescale, the vertical supply of oxygen to the OMZ in the Arabian Sea (Resplandy et al., 2012). This study also showed that eddy-driven horizontal advection substantially contributes to the lateral transport of ventilated waters into the central and northern Arabian Sea."

9. **Section 4. L336:** note that both the work of McCreary and Resplandy suggest that "this mechanism strongly contributes to the eastward shift…". As the authors pointed out earlier in this paragraph eddy-driven ventilation supplies oxygen to the western Arabian Sea in both studies. Please rephrase so it is clear that both studies converge here.
   We agree with the referee. For more clarity we will change that statement to:
   "In a process study aiming to explore the dynamics of the Indian Ocean OMZs, McCreary et al (2013) also highlighted the important role of vertical eddy mixing in the ventilation of the western Arabian Sea in addition to the inflow of ICW. Their work further stresses the importance of this mechanism in the eastward shift of the upper OMZ relative to the region of highest productivity located along the western part of the Arabian Sea."

10. **Section 4. L374-376:** Does the Chen et al paper mentions a decline in eddy activity? This should be clarified. If it is interannual variability and not a long term decline, then you would expect interannual variability in the OMZ ventilation and denitrification but not necessarily a deoxygenation.
We agree with the referee. We will change this statement to:
"In the Bay of Bengal, strong interannual variations in the intensity of the eddy activity have been reported (Chen et al., 2012). These are expected to cause strong variations in the subsurface ventilation that may eventually lead to episodic suboxia and onset of denitrification at the core of the OMZ (Johnson et al., 2019)."

11. **Section 4. L385-389.** The links here are not clear. The bio/eddy-driven ventilation balance identified in present day models does not suggest that remotely forced changes in physical supply cause long-term changes. The supply of oxygen to the OMZ has to be through mixing and is promoted by eddy-driven circulation, because there are no direct advective pathways into the OMZ shadow zone (by definition). The authors are right however that large scale circulation is important because it regulates the oxygen gradients at the OMZ edges. However, I don't see why Holocene changes could not be tied to changes in biological demand? I would remove these sentences here and keep this discussion for the Holocene section 5 (see comment #14).
Following the referee suggestion, this statement will be removed.

12. **Section 5 L425-430** should point to Figure 6 to help reader follow. I suggest the authors slightly reorganize the text between L425 and 449. Starting with early Holocene before 6000 BP (move L433-439 up), then transition with the increased in productivity and enhanced OMZ after 6000BP (combine L425-432 and L439-450).
We will rephrase this section to make the line of events more clear.

13. **Section 5.2 L488-490:** "a data-model comparison … in both basins". I thought the data-model comparison was only for the Arabian Sea. Please clarify. Note that adding an insert map of core location and model regions on figure 6 or would help the reader locate things. At least provide lon/lat of cores.
The wording comes from an earlier version of the ms. which included a section on the BoB in Section 5 which was removed due to excessive length of the ms. We will rephrase this section.

14. **Section 5.2 L495-498:** I am not sure I follow how the match between model and data in oxygen suggests that it is due to oceanic circulation rather than local biological processes. The authors state "it is assumed that .." in L490 but it seems neither the authors nor prior work has actually showed that circulation controls the simulate change in oxygen in this region in the model. This is an important point because that claim is repeated several times in the manuscript (see L385 and comment #11, L307 and comment #6 and conclusion). The author should either look at the biological and/or circulation changes in the model they present here or use models from others such as Bopp et al 2017 to make the claim (see comment #15 below).
See authors' response to second paragraph of review with ref to Segschneider et al., 2018, where all these points are addressed.

15. **Section 5.** Please consider adding the study of Bopp et al 2017, which compares simulations at the LGM and mid-holocene, linking to the changes from Pleistocene to Holocene mentioned by the authors. The paper includes a qualitative comparison of simulated O2 with O2 proxies (Fig 3) and

shows model ventilation changes between LGM and mid-holocene (Fig S3) in the Indian Ocean. Note that this model is not a transient run.
See response to #14.

16. **Section 6.** Authors should discuss their Figure 7 here. It is only mention in passing in L512. Something like "as shown in Figure 7….". Indeed, most prior work on ESM's OMZ was not targeting the Indian Ocean. Figure 7 would be a good opportunity to present specifically the results in the Indian Ocean.
That is correct. We will discuss Figure 7 and problems associated with ESMs in more detail.

17. **Section 6.** Authors should consider folding in this section the following recent papers looking at global OMZ and oxygen in ESMs. Models agree on the sign of warming-driven (O2sat) and biological-circulation (AOU) changes, but uncertainties arise from the subtle balance between these two opposing terms (Bopp et al 2017, Resplandy 2018). Papers highlighting the influence of circulation changes and non-resolved processes such as eddy-driven circulation and mixing (Duteil and Oschlies, 2011, Duteil et al 2014, Lachkar et al 2016, Palter and Trossman 2018, Fu et al 2018, Busecke et al 2019, Bahl et al 2019, Couespel et al 2020).
It is true that we missed out some of the recent model results in this section and we thank the reviewer for pointing them out. We will include them in section 6 by discussing more in detail, why the future predictions have such great uncertainties.

18. **Section 7.1.1** The text is well written but it is much more detailed than the rest of the sections in the review. Authors might consider summarizing/emphasizing the take home messages, the links with oxygen and the OMZ and the implications for trophic webs which are quickly metion at the end of section L640. If there are there other groups than the co-authors that worked on this topic (I am not a specialist of this subtopic), it might be worth including some of their work here.
We will shorten this section by deleting some of the details and make a better effort to summarize our work especially the implications from and for the OMZ. Very few researchers have looked at the reasons for the recent and sudden appearance of mixotrophic blooms in winter in the Arabian Sea and none has delved into its connection to the OMZ. However their results will be discussed.

19. **Section 7.1.2.** The OMZ control migration but please also consider adding the fact that zooplankton vertical migration influence the oxygen consumption vertical patterns. This effect is missing from most ocean bio models and from all ESMs. The following studies are global but include maps showing the impact in the Indian Ocean. Bianchi et al. 2013 (their Fig 3) Aumont et al 2018 (their Fig 9) show simulated oxygen decline due to DVM. Note that most references in this section are relatively old. The authors could consider checking for newer results on this topic, maybe including references from other OMZs to fuel their discussion if not available in Indian Ocean.
Ok

20. **Section 7.1.3** in implications discusses zooplankton but not the DVM aspects. It might be missing because part of the section is missing (see unfinished sentences L 682).
This will be clarified

21. **Section 8.** The conclusion is too vague and too speculative. It tries to blend mesoscale eddies to paleo-changes but this is a difficult task (see my comments #6, 11 and 14). "This was caused by .. changes in circulation". Again this has not been demonstrated by the Authors (note that the paper

by Bopp et al 2017 which shows ventilation changes between LGM and mid-holocene might help the authors to make the case).
This will be clarified.

22. **Figure 4**: specify O2 threshold used to compute OMZ thickness and how this "maximum thickness" is evaluated. Also briefly describe the data used here: How many cruises or from a database? What are the years during which these data were taken? Label seasons on plot (e.g. change symbols, add labels or colors).
The depth of the OMZ and mean oxygen concentration were obtained by Acharya and Panigrahi (2016) from the Word Ocean Atlas 2013 and the Global Ocean Data Analysis Project. These authors applied the 20 µM threshold to define the OMZ. We will clarify this in the revised version of the ms.

23. **Figure 5** could be improved so the difference between the two panels with/without eddies is more obvious and consistent with the text, ie.e eddies influence oxygen supply, nutrient supply and production. Oxygen does not seem to change between the two. Production and nutrient supply do not seem to change either.
Ok, this will be improved.

24. **Figure 6:** Nice and interesting figure.
  - It would be great to add the WOA present day oxygen concentration at 0 ka to compare to the model results on panel b. done in Segschneider et al., 2018
  - Please specify the model depth range for the oxygen values in caption.
  - 200 -800m, specified in Fig
  - Please provide an insert map with core locations and model regions and/or provide lon/lat of cores and model regions in caption. Please remove "sinking" from caption, this is confusing (It sounds like subduction of oxygen).
  Ok

**Other comments:**

L27: check grammar Ok
L112-114: move to next section? Unclear why it is here. Ok, this will be removed.
L150-151: Add other refs about filaments and eddies.
The given reference refers to organic carbon fluxes measured along a filament that developed during the JGOFS expeditions. This will be clarified and filament and eddies are discussed in a different chapter.
L183: typo on conational? Ok
L195: drops? dropped? Check tense (past/present) in section 2.3. Ok
L202: "the in comparison". Remove the? Ok
L203: "less intense … than" Ok
L223: should "since than" read "since then"? Ok
L226-227: add ref for the mixing  analyses. Ok
L233: (to 75%). Is this up to 75%?  It is the average – It is the average. The 'to' will be deleted.
L239: last sentence of section should be clarified and better linked to the rest of the paragraph. Why are they linked if the seasonal thermocline is hypoxic? Do you mean if the oxygen content of the seasonal thermocline remains stable through seasonal changes? This sentence probably belongs to the next section which defines the seasonal thermocline and make the link between bio production and physical supply of oxygen. The sentence will be deleted.

L240: Note that you define seasonal thermocline in L 241 but use it already in L239. This part will be changed.

L246: "the season thermocline**s**" > seasonal thermocline? Ok

L255: upper part **of** the thermocline? Upper thermocline? Ok

L266: remove "upper part" and "lower part" as depth are specified. Ok

L267: "the base of the SNM is located…. In contrast to the SNM…" the base of the SNM is in the SNM. This sentence doesn't make sense. Please rewrite. Ok

L272: which suggests. Ok

L306 "preventing the development of anoxic conditions". As noted by the authors in section 2.3 anoxic conditions already occur in the northern IO. Clarify the sentence. This will be changed into 'persistent anoxic conditions'.

L341-342: please rephrase sentence. "This leads" what leads? Clarify the links between oxygen change, denitrification, nutrient supply, production and feedback on oxygen change. Ok

L345: Thus eddies "would/could" affect….. this model results present a very interesting hypothesis but it does not make the link to fish habitat. At least modulate the link to fish except if you have a reference that makes this link in this region. Ok

L351: "Using YY It could be shown that XX (Lachkar et al )" replace by Lachkar et al () showed that XX using YY. This will be clarified.

L360 and 363: remove "and hence weaken the OMZ" and "weakening the OMZ". Here the authors discuss the mean state of the OMZ not a tremd. As mentioned above Bio + Phy ~ 0 in OMZs, hence eddy ventilation does not weaken the OMZ, it contributes to the supply of oxygen that balance the biological demand. Note that your section 4.3 discusses how variations in eddy activity could indeed result in variations in the oxygen supply and OMZ volume. Ok, this will be clarified.

L364-365: link to denitrification – cite paper(s) showing that denitrification inhibition occurs here. Ok

L385. Not clear why there is a "However" to start the sentence here. Is this sentence incompatible with the previous one? 'However' will be deleted.

L439: could you please clarify the link between enhanced upwelling and ventilation by ICW? Is it through reduced residence time? The monsoon drives both upwelling and the inflow of ICW. The ICW improves the ventilation by supplying oxygen to the OMZ and reducing the residence time of water within the OMZ. This will be clarified.

L448: "matches results from model…" I don't think this model has been presented yet. Please provide reference here or reference to section 5.3 which comes after. Also add reference to Figure 6b here. Ok

L524 please clarify that the increase in hypoxic waters is global scale not in the Indian Ocean. Ok

L582: does the journal authorize "in review" citations? This will be changed.

L682: missing text? Ok

L715: define OM. (organic matter, Ok)

L796-797: check sentence and grammar. Ok

L798: "The in comparison"? Ok

References;

Aumont, O., Maury, O., Lefort, S., Bopp, L., 2018. Evaluating the Potential Impacts of the Diurnal Vertical Migration by Marine Organisms on Marine Biogeochemistry. Global Biogeochemical Cycles 32, 1622–1643. https://doi.org/10.1029/2018GB005886

Bahl, A., Gnanadesikan, A., Pradal, M.-A., 2019. Variations in Ocean Deoxygenation Across Earth System Models: Isolating the Role of Parameterized Lateral Mixing. Global Biogeochemical Cycles 33, 703–724. https://doi.org/10.1029/2018GB006121

Bianchi, D., Galbraith, E.D., Carozza, D.A., Mislan, K. a. S., Stock, C.A., 2013. Intensification of open-ocean oxygen depletion by vertically migrating animals. Nature Geosci 6, 545–548. https://doi.org/10.1038/ngeo1837

Bopp, L., Resplandy, L., Untersee, A., Mezo, P.L., Kageyama, M., 2017. Ocean (de)oxygenation from the Last Glacial Maximum to the twenty-first century: insights from Earth System models. Phil. Trans. R. Soc. A 375, 20160323. https://doi.org/10.1098/rsta.2016.0323

Boyd, P.W., Claustre, H., Levy, M., Siegel, D.A., Weber, T., 2019. Multi-faceted particle pumps drive carbon sequestration in the ocean. Nature 568, 327. https://doi.org/10.1038/s41586-019-1098-2

Busecke, J.J.M., Resplandy, L., Dunne, J.P., 2019. The Equatorial Undercurrent and the Oxygen Minimum Zone in the Pacific. Geophysical Research Letters 46, 6716–6725. https://doi.org/10.1029/2019GL082692

Couespel, D., Lévy, M., Bopp, L., 2019. Major Contribution of Reduced Upper Ocean Oxygen Mixing to Global Ocean Deoxygenation in an Earth System Model. Geophys. Res. Lett. 46, 12239–12249. https://doi.org/10.1029/2019GL084162

Duteil, O., Oschlies, A., 2011. Sensitivity of simulated extent and future evolution of marine suboxia to mixing intensity. Geophys. Res. Lett. 38, L06607. https://doi.org/10.1029/2011GL046877

Duteil, O., Böning, C.W., Oschlies, A., 2014. Variability in subtropical-tropical cells drives oxygen levels in the tropical Pacific Ocean. Geophys. Res. Lett. 41, 2014GL061774. https://doi.org/10.1002/2014GL061774

Lachkar, Z., Smith, S., Lévy, M., Pauluis, O., 2016. Eddies reduce denitrification and compress habitats in the Arabian Sea. Geophysical Research Letters 43, 9148–9156. https://doi.org/10.1002/2016GL069876

McGillicuddy, D.J., 2016. Mechanisms of Physical-Biological-Biogeochemical Interaction at the Oceanic Mesoscale. Annual Review of Marine Science 8, 125–159. https://doi.org/10.1146/annurev-marine-010814-015606

Mahadevan, A., 2016. The Impact of Submesoscale Physics on Primary Productivity of Plankton. Annual Review of Marine Science 8, 161–184. https://doi.org/10.1146/annurev-marine-010814-015912

Omand, M.M., D'Asaro, E.A., Lee, C.M., Perry, M.J., Briggs, N., Cetinić, I., Mahadevan, A., 2015. Eddy-driven subduction exports particulate organic carbon from the spring bloom. Science 348, 222–225. https://doi.org/10.1126/science.1260062

Palter Jaime B., Trossman David S., 2018. The Sensitivity of Future Ocean Oxygen to Changes in Ocean Circulation. Global Biogeochemical Cycles 0. https://doi.org/10.1002/2017GB005777

Resplandy, L., 2018. Will ocean zones with low oxygen levels expand or shrink? Nature 557, 314–315. https://doi.org/10.1038/d41586-018-05034-y

Resplandy, L., Lévy, M., McGillicuddy, D.J., 2019. Effects of Eddy-Driven Subduction on Ocean Biological Carbon Pump. Global Biogeochem. Cycles 2018GB006125. https://doi.org/10.1029/2018GB006125

---

## Author Response (AR1)

Dear Dr. Viviane V. Menezes,

thank you very much for your constructive and helpful comments as well as for extending the deadline which allowed us to wok jointly with all co-authors on the revised manuscript.

Since the reviewer mainly criticized the readability and writing style almost the entire ms was rewritten and sections 1 - 3 were also restructured. The entire ms was checked by the co-authors among which are also native speakers. The changes were so intensive that a marked-up manuscript would have been entirely red except section 7.2 . This section was hardly criticized by the reviewer. Since the point-to-point responses to the reviewer comments have already been submitted earlier I am sending you here a non-marked-up version of the revised ms. Hope that is ok.

With best regards, Tim Rixen

---

## Author Response (AR2)

Referee # 2/1: Annie Bourbonnais

Dear Annie Bourbonnais,

thank you very much for your constructive and valuable comments, which helped us a lot to further improve the manuscript. The point-by-point reply to your latest comments follows below.

With best regards,
Tim Rixen, also on behalf of all coauthors.

General comment:

In their revised manuscript, Rixen et al. addressed most of my concerns. The revised manuscript is greatly improved, and was completely re-structured. Although quite dense and not always concise, this manuscript will represents a great contribution to the special issue "Understanding the Indian Ocean system: past, present and future" in Biogeosciences. I recommend minor revisions after addressing the few comments below.

Overall, some confusion still persists regarding their $O_2$ threshold used to define hypoxia. In their response, they explain that they considered 20 µM as a upper threshold below which denitrification and anammox occurs, i.e., fixed nitrogen is transformed to N2. However, such a high $O_2$ threshold is not supported by recent studies (including Dalsgaard et al. (2014) and Bristow et al. (2016)). For instance, Dalsgaard et al. (2014) reports $O_2$ threshold in the nmol range for conversion of nitrate to N2. Other recent studies (e.g., Frey et al. 2020) report a much higher $O_2$ threshold (up to 10 µmol) for nitrate reduction to N2O, yet lower than the suggested value of 20 µmol. Furthermore, later on (response to my comment, line 471 of the original manuscript), they admit using a $O_2$ threshold of 6 µM for denitrification in the PISCES model, which makes it even more confusing. I recommend the authors to lower their upper $O_2$ threshold for denitrification (and hypoxia) to at most 10 µM, which would be more in line with the value suggested by most recent studies as well as the value used in the PISCES model.

In principle, we see here no contradiction to the text in our manuscript, but maybe we did not express ourselves clear enough. What we wrote was:

'According to experiments and *in situ* observations, anammox sets in when oxygen concentrations drop below ~20 µM, while denitrification occurs at oxygen concentrations of approximately < 6 µM (Fig. 2, Bristow et al., 2016; Dalsgaard et al., 2014; Kalvelage et al., 2011).'

Since we defined microbial hypoxia as the range within which anoxic microbial processes can occur, and anammox is one of these, we set the threshold for microbial hypoxia to an oxygen concentration of ~20 µM. However, we also pointed out that anoxic processes gain importance and slowly outcompete oxic processes at lower oxygen concentrations. As shown e.g. in Fig. 3 by Dalsgaard et

al. (2014), a decreasing oxygen inhibition seems to cause this, which includes also a stepwise onset of further anoxic processes such as denitrification. We tried to clarify this in the revised ms.

PISCES, which is an often-used biogeochemical ocean model, considers the impact of denitrification on the marine nitrogen cycle starting at oxygen concentrations below a threshold of 6 µM. Full denitrification sets in at 0.05 µM. This is state of the art. However, considering the latest results, one might even suggest to use 0.05 µM as an upper oxygen threshold in future studies because only at such low oxygen concentration the reduction of fixed nitrogen to N2 becomes significant. However, this opinion is not yet commonly accepted and we see our ms as a contribution to change this.

Additionally, sulfate reduction could theoretically only occur after the development of anoxic conditions, hence anoxic waters are not necessarily sulfidic (for instance, if nitrate concentrations are high, nitrate will first be used as the terminal electron acceptor during respiration). Defining anoxia as purely sulfidic conditions is consequently misleading.

We agree and wrote: 'the appearance of hydrogen sulfide is generally considered as an indicator of anoxia'

Abstract:

Lines 20-22: This sentence is confusing. What do they mean by "... which includes negative feedback mechanisms reducing the oxygen consumption at decreasing oxygen concentrations."? I assume denitrification/anammox and reduced respiration are the negative feedback mechanisms? I would change for: "which includes negative feedback mechanisms reducing oxygen consumption at decreasing oxygen concentrations (e.g., reduced respiration).

To clarify the text we added 'which includes negative feedback mechanisms reducing oxygen consumption at decreasing oxygen concentrations (e.g., reduced respiration)'

Text:

Lines 123-135: I would consider a lower O2 threshold for microbial hypoxia of 10 µM (see my comment above).

As stated above, we defined microbial hypoxia as the range within which anoxic microbial processes can occur and anammox is one of these, and occurs apparently already at higher oxygen concentrations.

Lines 415-416: Why is the reliability of the older data set questioned?

As far as we understood because there have been no other reports on H2S in the open Arabian Sea and the Bay of Bengal thereafter. Our statement is based on the

work of  Sen Gupta, R., Naqvi, S.W.A., 1984. Chemical Oceanography of the Indian Ocean, North of the Equator. Deep Sea Research, 31, 671 - 706.

Lines 575-578: I would cite Fassbender et al. (2018), which offers a concise reviews of mesoscale and sub-mesoscale circulation in the ocean.
Thanks , was done

Lines 618-623: I don't think this mechanism is well represented in Figure 11. Export production seems to be equal in both panels (with and without eddies).
Following the suggestion from Reviewer #4, this and two more figures have been removed from the ms.

Technical corrections:

Line 254: change to " in combination with the a strong inflow…"
was changed

Lines 924-925: This sentence needs to be revised: "from the lack of oxygen the tolerance to decreasing oxygen, critical concentrations…"
Figure 6: change to: "meridional overturning circulation in the Indian Ocean according to…"

Following also the suggestion from Reviewer #4 this paragraph and figure 6 have been deleted.

Additional references:

Fassbender, A. J., Bourbonnais, A., Clayton, S., Gaube, P., Omand, M., Franks, P. J. S., … & McGillicuddy Jr, D. (2018). Interpreting mosaics of ocean biogeochemistry. Eos, 99(10.1029).

Frey, C., Bange, H. W., Achterberg, E. P., Jayakumar, A., Löscher, C. R., Arévalo-Martínez, D. L., … & Oleynik, S. (2020). Regulation of nitrous oxide production in low-oxygen waters off the coast of Peru. Biogeosciences, 17(8), 2263-2287.

**Referee # 1**

We thank the reviewer for the suggestions, which helped to streamline the ms.

Rixen et al. assemble a comprehensive and useful overview of the state of knowledge of the OMZ in the northern Indian Ocean, specifically the Arabian Sea (AS) and Bay of Bengal (BoB). They address a wide range of topics related to biological and physical dynamics, paleo records, and model results. The authors addressed most of the 3 anonymous reviewers' comments on the original manuscript. Overall, the revised manuscript is more comprehensible, however the revisions did not unify the different sections of the manuscript - the overall structure and content of each section remained the same, which results in repetition and disjointedness.

Some examples:
Sections 2.2 and 3.1 both discuss the link between OMZs and upwelling and parts could be combined;
Sections 2.3 and 7.1 both discuss impacts of OMZ expansion and intensification on biology;
Sections 2.1 and 4 both discuss the physical processes that ventilate the OMZ – from large scale circulation to mesoscale eddies – and these could be tied together somehow.
Combining some of these sections and eliminating repetition would streamline this manuscript and help the reader understand the points the authors' are trying to make.

We agree and
1) eliminated repetition in 7.1,
2) merged section 2.1 and 4, and
3) selected more suitable heading for section 2.2, which was divided in two parts:
      2.2 Spatial and temporal variability of the Arabian Sea OMZ
      2.3 The Bay of Bengal OMZ.
4) Section 3 was also renamed and includes now both the biological and physical drivers.
All over these changes reduced the length and streamlined the ms.

Specific comments:

Lines 58-60. "The transition from anaerobic…hydrogen sulfide." From the first part of this sentence I expected to read about the steps from anaerobic to aerobic. The second part of this sentence seems to be a non sequitur. Additionally, sulfate reduction can be coupled to the oxidation of multiple types of reduced carbon, not only methane.
This sentence was deleted

Section 2.1. This entire section is confusing and needs work. The authors first state that both primary production and flux from the atmosphere supply oxygen to the surface (lines 171-172), but then make no mention of the role of primary productivity in affecting the gradient/flux between the OMZ and the surface (lines 194-205). Then it seems like the authors are arguing that vertical mixing ventilates the OMZ (lines 194-205), but then they discuss the importance of lateral mixing (lines 210-211).
Line 172. After "sources of dissolved oxygen", insert "to the surface".

In a first step toward clarification we followed the suggestion to insert 'in surface waters', and secondly the section was shortened and, as suggested, merged with section 4.

Line 253. "Fig. 4c" should be Figs. 3a and 3b.
ok thanks!

Lines 503 and 506, Fig. 7a. "deepening" and "OMZ depth" Do the authors mean "thickening" and "thickness"?
Yes, we changed it accordingly

Line 540. "low oxygen consumption" Should this be "high oxygen consumption"?
Yes, was changed

Lines 622-623. "The consequence is an expansion of the volume of the OMZ." Fig. 11 shows the opposite – eddies decrease the size of the OMZ. I understand that the effect of eddies can be to both increase and decrease the size of the OMZ but Fig. 11 only shows eddies decreasing the OMZ by increasing ventilation and the juxtaposition with the text is confusing.
Figures 4 and 6. These figures do not add much information beyond what is stated in the text. I suggest removing them to help shorten this manuscript.
Figures 11, 5 and 6 were deleted.

Line 728. "Fig. 6a" should be 12b and 12c.
ok

Section 7.1. There is no mention of the BoB.
We added the following sentence:
little is known of its effect on zooplankton distribution and vertical migration and this also holds true for the Bay of Bengal OMZ. ' to the summary of section 7.1 which is now section 6.2.

Line 1291. "section 7.3" should be 7.2.
Great, we changed it

Figure 7a and b. Please provide maximum and minimum ranges for these mean values.
Figure 7b. "mean OMZ oxygen concentration" How is this value calculated?

Volume-weighted average of water <20 micromolar oxygen?
Figure 8a and b. Same as with Fig. 7 – please provide ranges

The data on the aerial extend, OMZ thickness and oxygen concentrations were obtained from Table 5 in Acharya, S.S., Panigrahi, M.K., (2016) and not calculated by us. These authors provide only the standard deviations for the mean oxygen concentrations, which in addition to those derived from the primary production rates were included into the respective figures.

---

## Author Response (AR3)

Dear Dr. Menezes,
please find attached the corrected ms.
Thank you very much for your support,
which I appreciate very much.
With best regards, Tim Rixen